# Off-Policy Actor-Critic for Adversarial Observation Robustness: Virtual Alternative Training via Symmetric Policy Evaluation

**Kosuke Nakanishi** [1][2]  **Akihiro Kubo** [1][3][4]  **Yuji Yasui** [2]  **Shin Ishii** [1][3][4]

## Abstract

Recently, robust reinforcement learning (RL) methods designed to handle adversarial input observations have received significant attention, motivated by RL's inherent vulnerabilities. While existing approaches have demonstrated reasonable success, addressing worst-case scenarios over long time horizons requires both minimizing the agent's cumulative rewards for adversaries and training agents to counteract them through alternating learning. However, this process introduces mutual dependencies between the agent and the adversary, making interactions with the environment inefficient and hindering the development of off-policy methods. In this work, we propose a novel off-policy method that eliminates the need for additional environmental interactions by reformulating adversarial learning as a soft-constrained optimization problem. Our approach is theoretically supported by the symmetric property of policy evaluation between the agent and the adversary. The implementation is available at https://github.com/nakanakakosuke/VALT_SAC.

## 1. Introduction

In recent years, advancements in computational technology, coupled with the practical successes of deep neural networks (DNNs) (Krizhevsky et al., 2012; Simonyan & Zisserman, 2015; He et al., 2016), have fueled expectations for automated decision-making and control in increasingly complex environments (Kober et al., 2013; Levine et al., 2016; Kiran et al., 2021). Deep reinforcement learning (DRL) is a promising framework for such applications, demonstrating performance that surpasses human capabilities by acquiring high-dimensional representational power through function approximation (Mnih et al., 2015; Silver et al., 2017; Vinyals et al., 2019; Wurman et al., 2022). However, the concerns are raised that only for the small input perturbations, such as adversarial attacks, significant performance degradation can occur, not only in supervised learning tasks (Szegedy et al., 2014; Goodfellow et al., 2014; Kurakin et al., 2016; Papernot et al., 2016), but also in decision making and control tasks (Huang et al., 2017; Lin et al., 2017).

Robust decision-making under uncertainty has long been studied by formulating worst-case scenarios to ensure stable performance in adverse conditions. Classical approaches include game-theoretic models such as the two-player Markovian game (Shapley, 1953), and the Robust Markov Decision Process (Robust-MDP) framework (Nilim & El Ghaoui, 2005; Iyengar, 2005; Tamar et al., 2014), which seeks policies that perform well under the most pessimistic transition dynamics. However, these methods are typically restricted to tabular or linear settings and do not scale well to complex environments (Tessler et al., 2019). More recently, Pinto et al. (2017) proposed an adversarial RL framework in which an agent is jointly trained with an adversary to approximate a max-min Nash equilibrium (Nash, 1951), yielding policies that are empirically robust, though lacking theoretical guarantees.

These early studies primarily addressed robustness by explicitly modeling uncertainty in environment dynamics or introducing adversarial agents to simulate worst-case behaviors. In contrast, recent work has highlighted the need to distinguish between different sources of uncertainty and to formulate robustness at the level of specific failure modes. In this work, we focus on observation uncertainty, which has emerged as a critical yet underexplored challenge in DRL.

Recent research (Zhang et al., 2021; Sun et al., 2022; Liang et al., 2022) identifies two main types of vulnerabilities for observation robustness in DRL. The first arises from the function approximation properties of DNNs, often manifesting as sensitivity to small perturbations due to lack of smoothness in the learned policy. The second vulnerability

---

[1]Department of Information Science, Kyoto University, Kyoto, Japan [2]Honda R&D Co., Ltd., Tokyo, Japan [3]ATR Neural Information Processing Laboratories, Kyoto, Japan [4]International Research Center for Neurointelligence, The University of Tokyo, Tokyo, Japan. Correspondence to: Kosuke Nakanishi <kosuke_nakanishi@jp.honda>, Shin Ishii <ishii@i.kyoto-u.ac.jp>.

*Proceedings of the 42nd International Conference on Machine Learning*, Vancouver, Canada. PMLR 267, 2025. Copyright 2025 by the author(s).

stems from the dynamics of the environment and is considered within the framework of MDPs.

To illustrate this, consider a scenario where an agent must cross a deep valley using one of two bridges. One bridge is short but narrow, while the other is longer but wider. With clear vision, the agent may prefer the shorter path; however, under noisy or partially observed conditions—such as heavy fog—the wider, safer bridge may be the more reliable choice. Such decision-making requires incorporating long-term rewards considerations into robustness design, rather than focusing solely on local smoothness or output consistency as is common in supervised learning settings.

To address the observation vulnerability stemming from the MDPs, Zhang et al. (2021) propose an approach where the (victim) agent and the corresponding (optimal) adversary are trained alternately, resulting in a robust agent. They refer to this framework as *Alternating Training with Learned Adversaries* (ATLA).

Although this alternating approach is theoretically robust and has shown practical success, it has several limitations and unresolved challenges.

First, the mutual dependency between the victim agent and the adversary effectively doubles the required sample size for training,[1] resulting in inefficiencies and increased computational cost. Moreover, to the best of our knowledge, no existing off-policy actor-critic method can address such interactions within the standard MDP framework.

In this study, we propose a novel framework that eliminates the need for additional interactions with the environment and is compatible with off-policy actor-critic algorithms. Our contributions are summarized as follows: **(1)** We theoretically prove that it is possible to construct an alternative adversarial framework for observation robustness without requiring an explicit RL process for the adversary. **(2)** We propose two practical algorithms that demonstrate both sample efficiency and robustness, while simultaneously providing insights into underexplored regions in off-policy RL.

Our research advances the fields of RL and off-policy frameworks by addressing key limitations and introducing a practical approach to enhance performance and generalization.

## 2. Related Work

**Adversarial Attack and Defense on State Observations.** Building on a seminal work by Goodfellow et al. (2014),

there has been a surge of research activity in the field of supervised learning, particularly focusing on various adversarial attacks and corresponding defense methods (Kurakin et al., 2016; Papernot et al., 2016; 2017; Carlini & Wagner, 2017; Ilyas et al., 2018). In the context of DRL, Huang et al. (2017) demonstrated that similar challenges could arise from small perturbations, such as the Fast Gradient Sign Method (FGSM). This study led to the early proposal of various attacks on observations and corresponding robust RL methods (Kos & Song, 2017; Behzadan & Munir, 2017; Mandlekar et al., 2017; Pattanaik et al., 2018).

Recently, a line of research has focused on maintaining the consistency (smoothness) of the agent's policy to acquire robustness against observation perturbations (Zhang et al., 2020b; Shen et al., 2020; Oikarinen et al., 2021; Sun et al., 2024). Zhang et al. (2020b) first defined the adversarial observation problem as *State Adversarial*-MDPs (SA-MDPs). They proved and demonstrated that the loss in performance due to adversarial perturbations could be bounded by the consistency of the policy. However, in this study, they did not show practical methods to create the adversary that could estimate the long-term rewards assumed in the SA-MDPs. Due to this gap, the proposed robust methods were not robust enough against stronger attacks.

To address this issue, Zhang et al. (2021) proposed that the optimal (worst-case) adversary for the policy could be learned as a DRL agent. The policy, learned alternatively with such an adversary, can become robust against strong attacks (ATLA). Building on the ATLA framework, Sun et al. (2022) suggested dividing the adversary into two parts: searching for the mislead direction in the action space for the policy and calculating the actual perturbations in the state space through numerical approximation (PA-AD). This makes the adversary capable of handling high-dimensional state problems (such as Atari), which are difficult to address in the ATLA framework. These frameworks provide practical methods for on-policy algorithms (PPO, A2C), but applications for off-policy algorithms were not shown. This is because the adversarial attacker learns from trajectories generated by the (current) fixed agent, but the agent policy is constantly updated during training.

Liang et al. (2022) introduced an additional worst-case action-value function to improve the policy's robustness by utilizing a mixture with the original value function, referred to as *Worst-case-aware Robust* RL (WocaR-RL). This approach computes the worst-case action-value through convex relaxation and heuristic gradient iterations, thereby omitting additional environment steps, unlike ATLA-based methods (ATLA, PA-ATLA). WocaR-RL demonstrated effectiveness in high-dimensional discrete action domains using off-policy algorithms (e.g., DQN). However, applications for off-policy algorithms in continuous action domains

---

[1]This assumes that neither an environment model (true or approximated) nor sufficient prior data is available. Such models or data entail challenges like model errors, data collection, and OOD issues. These issues are actively studied in the contexts of model-based and offline RL, but are considered out of scope in this work in order to maintain focus.

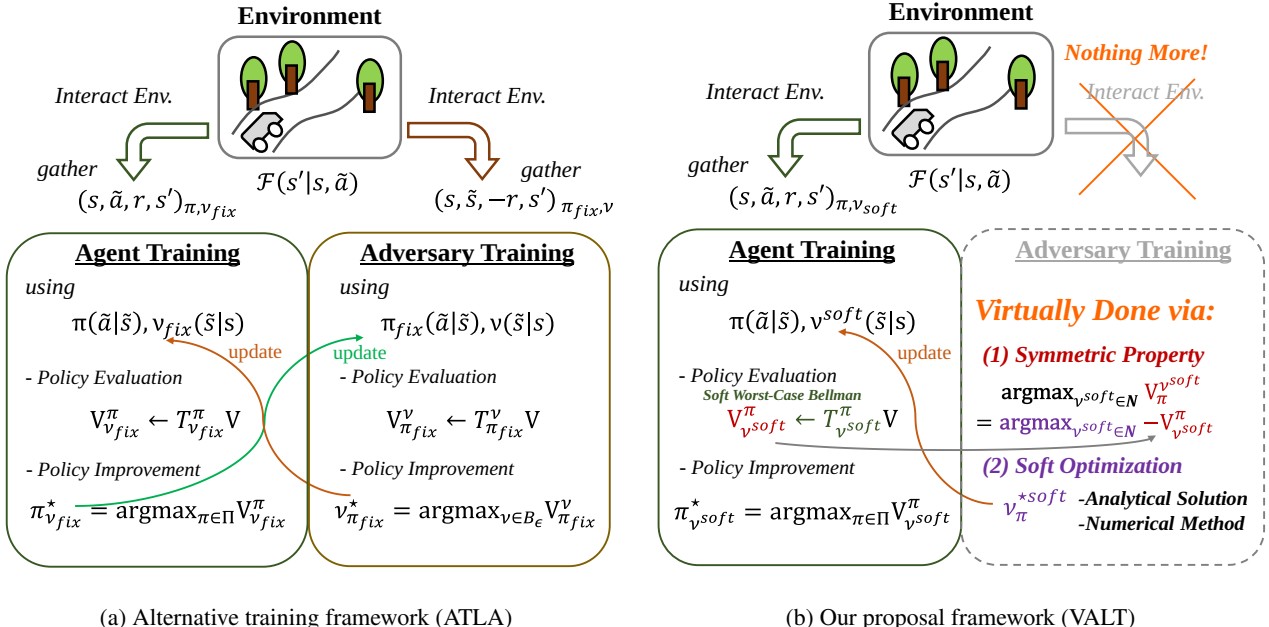

(a) Alternative training framework (ATLA)    (b) Our proposal framework (VALT)

*Figure 1.* Overview of two learning frameworks for creating a robust agent by countering a corresponding adversary. Figure 1a illustrates the conventional approach, which alternates between training the agent via RL and training the adversary via another RL. This method requires twice as many environmental interactions due to the mutual dependency between the agent and the adversary. In contrast, our proposed approach, shown in Figure 1b, eliminates this inefficiency while still accounting for the corresponding adversary.

were not shown and remain unknown.

As a separate line of research, there are studies that address the ratio and temporal strategies of attacks as a multi-agent problem (Lin et al., 2017; Gleave et al., 2020; Sun et al., 2020; Liu et al., 2024; Franzmeyer et al., 2024). However, these studies focus on the objectives of attack efficiency or the stealthiness of attacks. This motivation differs from our approach, which limits the attacker based on constraints from an assumed distribution.

**Adversarial Attacks on Other Components.** Robustness against dynamics uncertainty has been extensively studied, starting from the classical Robust MDP formulation (Nilim & El Ghaoui, 2005; Iyengar, 2005; Tamar et al., 2014), which considers a set of uncertain transition models. Subsequent works have extended this line of research by incorporating additive perturbations to the dynamics (Morimoto & Doya, 2005; Pinto et al., 2017; Reddi et al., 2024; Dong et al., 2025), handling dynamics parameters (episodic variations) (Peng et al., 2018; Vinitsky et al., 2020; Tanabe et al., 2022), or modeling time-varying dynamics based on contextual information (Zhang et al., 2023).

Other studies focus on perturbations in the agent's action output or its stochastic components, often assuming noise-like corruptions during execution (Tessler et al., 2019; Klima et al., 2019; Lee et al., 2020; Tan et al., 2020). Robustness

against reward poisoning during training has also been investigated (Ma et al., 2019; Rakhsha et al., 2020), though such attacks are typically considered as training-time data perturbations rather than test-time vulnerabilities, since the reward function itself does not directly affect policy performance at evaluation time (Schott et al., 2024).

In connection with off-policy RL, recent works in offline RL have proposed methods that incorporate adversarial dynamics to avoid out-of-distribution (OOD) actions via max-min formulation (Rigter et al., 2022), as well as techniques that handle perturbations in dynamics or inputs under offline settings (Panaganti et al., 2022; Tang et al., 2024).

Although many of these studies are related in spirit, they are based on different problem settings and assumptions. Therefore, we do not conduct direct empirical comparisons in this work.

Additional related work and details are provided in Appendix A.

## 3. Preliminaries and Background

### 3.1. State Adversarial Markov Decesion Process

We assume that the agent receives perturbed inputs from the environment, as proposed in SA-MDP (Zhang et al., 2020b), which deviates from a standard MDP. The SA-MDP can

be characterized by the parameters $\langle \mathcal{S}, \tilde{\mathcal{S}}, \tilde{\mathcal{A}}, \mathcal{F}, r, \gamma, \mathcal{S}_0 \rangle$, where: $\mathcal{S}$ denotes the state space, $\tilde{\mathcal{S}}$ represents the observation space available to the agent, $\tilde{\mathcal{A}}$ is the action space, $\mathcal{F} : \mathcal{S} \times \tilde{\mathcal{A}} \to \Delta(\mathcal{S})$ defines the environment's transition probabilities, with $\Delta(\mathcal{S})$ denoting the set of probability distributions over the state space $\mathcal{S}$, $r$ is the reward function, $\gamma \in (0.0, 1.0)$ is the discount factor, and $\mathcal{S}_0$ is the set of initial states. In the observation-robust RL framework, we assume the agent observation is perturbed by some environment model (adversary), $\nu(\tilde{s} \mid s) \in \mathcal{N} : \mathcal{S} \to \Delta(\tilde{\mathcal{S}})$, then the agent policy, $\pi(\tilde{a} \mid \tilde{s}) \in \Pi : \tilde{\mathcal{S}} \to \Delta(\tilde{\mathcal{A}})$, acts based on the perturbed observations. Here, $\mathcal{N}$ is the set of all possible perturbation models, and $\Pi$ is the set of all possible policies.

The goal of robust RL is to derive a policy that maximizes the cumulative discounted reward along a trajectory, even under perturbed observations: $J[\nu, \pi] = \mathbb{E}_{\mathcal{S}_0, \nu, \pi, \mathcal{F}} \left[ \sum_{t=0}^{\infty} \gamma^t r(s_t, \tilde{a}_t) \right]$. To mitigate the high variance associated with trajectory-dependent estimation, most research focuses on estimating the expected action-value and state-value functions, defined respectively as $Q_{\nu}^{\pi}(s, a) = \mathbb{E}_{\nu, \pi, \mathcal{F}} \left[ \sum_{t=0}^{\infty} \gamma^t r(s_t, \tilde{a}_t) \mid s_0 = s, \tilde{a}_0 = a \right]$ and/or $V_{\nu}^{\pi}(s) = \mathbb{E}_{\nu, \pi, \mathcal{F}} \left[ \sum_{t=0}^{\infty} \gamma^t r(s_t, \tilde{a}_t) \mid s_0 = s \right]$, instead of directly maximizing $J[\nu, \pi]$.

To provide meaningful scenarios, we assume that the perturbation is constrained within an $L_{\infty}$-norm ball of radius $\epsilon$ around the true state as the most related research (Pattanaik et al., 2018; Zhang et al., 2020b; 2021; Sun et al., 2022; Oikarinen et al., 2021; Liang et al., 2022). It can be represented a restricted perturbation subset,

$$
\mathcal{B}_{\epsilon} := \left\{ \nu_{\epsilon} \in \mathcal{N} : \begin{array}{l} \nu_{\epsilon}(\tilde{s} \mid s) = 0 \text{ if } |\tilde{s} - s|_{\infty} > \epsilon, \\ \text{otherwise } \nu_{\epsilon}(\tilde{s} \mid s) \geq 0, \forall s \in \mathcal{S} \end{array} \right\}. \quad (1)
$$

### 3.2. Max-Min Alternative Training for Robust RL

To learn a robust agent policy that performs effectively under worst-case scenarios induced by an adversary, the solution satisfies the following Nash-equilibrium condition:

$$
\begin{aligned}
\nu_{\pi^{\star}}^{\star}, \pi_{\nu^{\star}}^{\star} &= \arg \max_{\pi \in \Pi} \arg \min_{\nu \in \mathcal{B}_{\epsilon}} J[\nu, \pi] \\
&= \arg \min_{\nu \in \mathcal{B}_{\epsilon}} \arg \max_{\pi \in \Pi} J[\nu, \pi].
\end{aligned} \quad (2)
$$

Zhang et al. (2021) adopt an alternating training approach to approximate this equilibrium. In this method, the agent and adversary are alternately optimized while holding the other fixed. The process alternates between the following objectives:

$$
\nu_{\pi_{\text{fix}}}^{\star} = \arg \max_{\nu \in \mathcal{B}_{\epsilon}} \mathbb{E}_{s \sim \rho_{\nu, \pi_{\text{fix}}}} \left[ V_{\pi_{\text{fix}}}^{\nu}(s) \right] \simeq -J[\nu, \pi_{\text{fix}}], \quad (3)
$$

$$
\pi_{\nu_{\text{fix}}}^{\star} = \arg \max_{\pi \in \Pi} \mathbb{E}_{s \sim \rho_{\nu_{\text{fix}}, \pi}} \left[ V_{\nu_{\text{fix}}}^{\pi}(s) \right] \simeq J[\nu_{\text{fix}}, \pi], \quad (4)
$$

where $\rho_{\nu, \pi}$ represents the discounted state visitation probability under $\nu$ and $\pi$, $V_{\pi_{\text{fix}}}^{\pi}(s)$ represents the expected value

for the agent with a fixed adversary, and $V_{\pi_{\text{fix}}}^{\nu}(s)$ represents the expected value for the adversary with a fixed agent. Due to the two optimization schemes, (3) and (4), previous works (Zhang et al., 2021; Sun et al., 2022) have been limited to on-policy methods (e.g., TRPO, PPO, A2C), and use twice the number of training steps. We summarize the procedures for this framework in Fig. 1a.

## 4. Methodology

To address the issue of sample inefficiency, we propose a new framework that does not require additional interactions with the environment while achieving optimization comparable to alternative training, as depicted in Fig. 1b. The core strategy consists of two key steps:

- **(1) Symmetric Property:** Updating the agent through policy evaluation with a symmetric property, which enables deriving the optimal adversary's value function directly from the agent's perspective, without requiring an explicit training process for the adversary.

- **(2) Soft Optimization:** Applying soft optimization scheme with utilizing the agent's value estimation to obtain the (soft) optimal adversary.

These procedures are naturally realized by introducing the soft-constrained adversary, which enables efficient optimization under the proposed framework. We refer to this framework as Virtual ALternative Training (VALT), as it lacks an explicit RL process for the adversary.

In this study, we adopt the *Soft Actor-Critic (SAC)* (Haarnoja et al., 2018a;b) framework as the base off-policy actor-critic method, but the discussion remains applicable even without the entropy term $\mathcal{H}$.

### 4.1. Soft Constrained Adversary

The optimization problem for the adversary in Eq. (3) and Eq. (4) is a hard-constrained problem due to the constraint defined in Eq. (1). To address this, we relax the problem by introducing a general $f$-divergence term, modifying the total objective function as follows:

**Definition 4.1** (Soft Constrained Optimal Adversary)**.**

$$
\nu_{\pi}^{\star \text{soft}} = \arg \max_{\nu \in \mathcal{N}} (-\tilde{J}[\nu, \pi]), \quad (5)
$$

$$
\tilde{J}[\nu, \pi] \triangleq J[\nu, \pi] + \alpha_{ent} \mathcal{H}(\pi \circ \nu) + \alpha_{attk} D_f(\nu \parallel p). \quad (6)
$$

Here, $D_f(\nu \parallel p)$ represents the $f$-divergence with respect to the prior distribution $p(\tilde{s} \mid s)$, and $\alpha_{attk}$ is the corresponding regularization coefficient. The term $\mathcal{H}(\pi \circ \nu)$ represents the Shannon entropy of the joint distribution $\pi \circ \nu$, with $\alpha_{ent}$ as

its coefficient. In this introduction, we make the following two assumptions:

**Assumption 4.2** (Sufficient Support Assumption of the Prior). For all $s, \tilde{s} \in \mathcal{S}$, if $\nu(\tilde{s} \mid s) > 0$, then $p(\tilde{s} \mid s) > 0$.

**Assumption 4.3** (Properties of the $f$ Function). The function $f$ used in the $f$-divergence term, $f\left(\frac{\nu}{p}\right)$, is assumed to be continuously differentiable and convex with respect to $\nu$.

Assumption 4.2 is easily satisfied in this work by setting $p$ as the uniform distribution such that $\text{dom}(p) = \text{dom}(\mathcal{B}_\epsilon)$. For Assumption 4.3, we select appropriate functions for $f$, as described in a later subsection.

### 4.2. Symmetric Policy Evaluation

To approximate $\tilde{J}[\nu, \pi]$ using the value function for adversary learning, we first consider the formulation of the RL process. In the following, we describe a soft worst-case (optimal) policy evaluation for both the agent and the adversary, assuming a fixed agent policy.

**Proposition 4.4** (Soft Worst Policy Evaluation for Agent).

$$(\mathcal{T}_{\nu^{soft}}^\pi Q_\nu^\pi)(s_t, \tilde{a}_t) \triangleq r(s_t, \tilde{a}_t) + \gamma \mathbb{E}_{\mathcal{F}} \left[ \min_{\nu \in \mathcal{N}} V_{\nu^{soft}}^\pi(s_{t+1}) \right],$$
$$V_{\nu^{soft}}^\pi(s_t) \triangleq \mathbb{E}_\nu \left[ \mathbb{E}_\pi \left[ Q_\nu^\pi(s_t, \tilde{a}_t) \right] \right] \\ + \alpha_{ent} \mathcal{H}(\pi \circ \nu) + \alpha_{attk} D_f(\nu \parallel p), \tag{7}$$

**Proposition 4.5** (Soft Optimal Policy Evaluation for Adv.).

$$(\mathcal{T}_\pi^{\nu^{soft}} Q_\pi^\nu)(s_t, \tilde{s}_t) \triangleq c(s_t, \tilde{s}_t) + \gamma \mathbb{E}_{\mathcal{F} \circ \pi} \left[ \max_{\nu \in \mathcal{N}} V_\pi^{\nu^{soft}}(s_{t+1}) \right],$$
$$V_\pi^{\nu^{soft}}(s_t) \triangleq \mathbb{E}_\nu \left[ Q_\pi^\nu(s_t, \tilde{s}_t) \right] \\ - \alpha_{ent} \mathcal{H}(\pi \circ \nu) - \alpha_{attk} D_f(\nu \parallel p), \tag{8}$$

where $c(s, \tilde{s}) \triangleq \mathbb{E}_\pi[-r(s, \tilde{a})]$ is reward function for the adversary, $\mathcal{T}_{\nu^{soft}}^\pi$ is a Bellman Operator for the agent, and $\mathcal{T}_\pi^{\nu^{soft}}$ is a Bellman Operator for the adversary, both in the soft worst adversary case as in Eq. (5). Due to a symmetric structure of the two propositions, we call the important theorem as:

**Theorem 4.6** (Symmetry of $\gamma$-Contraction Properties). *For a fixed agent policy $\pi$, if there exists a bounded function $Q_\nu'^\pi(s, \tilde{a})$ such that $Q_\pi^\nu(s, \tilde{s}) = \mathbb{E}_\pi[-Q_\nu'^\pi(s, \tilde{a})]$, and if $\mathcal{T}_{\nu^{soft}}^\pi$ is a $\gamma$-contraction operator, then $\mathcal{T}_\pi^{\nu^{soft}}$ is also a $\gamma$-contraction operator. Moreover, $\mathcal{T}_{\nu^{soft}}^\pi$ and $\mathcal{T}_\pi^{\nu^{soft}}$ share fixed points with opposite signs, $V_\pi^{\nu^{\star soft}} = \max_\nu V_\pi^{\nu^{soft}} = -\min_\nu V_{\nu^{soft}}^\pi = -V_{\nu^{\star soft}}^\pi$.*

*Proof Overview.* Substitute $Q_\pi^\nu(s, \tilde{s}) = \mathbb{E}_\pi[-Q_\nu'^\pi(s, \tilde{a})]$ and $c(s, \tilde{s}) = \mathbb{E}_\pi[-r(s, \tilde{a})]$ into Eq. (8). By multiplying both sides of Eq. (8) by $-1$, and flipping $\max$ into $\min$, Eq. (8) can be transformed into a form similar to Eq. (7). See Appendix B.1 for more details. $\square$

This theorem demonstrates that the adversary's value function can be avoided by leveraging the agent's action-value function, as the fixed point satisfies $V_\pi^{\nu^{\star soft}} = -V_{\nu^{\star soft}}^\pi$.

To exploit this property, we ensure that $\mathcal{T}_{\nu^{soft}}^\pi$ satisfies the contraction property by solving Eq. (5) from the agent's perspective. In the next subsection, we propose two approaches to achieve this.

### 4.3. Soft Optimization Approach

We aim to obtain the (soft) optimal adversary in Eq. (5) by utilizing the agent's value function:

$$\nu_\pi^{\star soft} = \arg \max_{\nu \in \mathcal{N}} -\tilde{J}[\nu, \pi] \simeq \arg \max_{\nu \in \mathcal{N}} \mathbb{E}_{s \sim \rho} \left[ -V_{\nu^{soft}}^\pi(s) \right].$$

This approach eliminates the need for direct adversary policy improvement, which is conventionally expressed as:

$$\nu_\pi^{\star soft} = \arg \max_{\nu \in \mathcal{N}} \mathbb{E}_{s \sim \rho} \left[ V_\pi^{\nu^{soft}}(s) \right].$$

**Approach 1: Analytical Solution-Based Modeling.** In this first approach, we utilize the case where an analytical solution can be derived.

**Lemma 4.7** (KL-divergence Analytical Solution). *If we set $f(x) = x \log(x)$ in Eq. (7), the analytical solution of Eq. (5) is given by:*

$$\nu_\pi^{\star soft}(\tilde{s} \mid s) = \frac{p(\tilde{s} \mid s) \exp\left(-V^\pi(s, \tilde{s})/\alpha_{attk}\right)}{Z}, \tag{9}$$

*where $V^\pi(s, \tilde{s}) = \mathbb{E}_\pi \left[ Q_\nu^\pi(s, \tilde{a}) \right] + \alpha_{ent} \mathcal{H}(\pi)$ and $Z$ is the partition function that normalizes the distribution.*

Moreover, regarding the policy evaluation for the agent:

**Theorem 4.8** (Contraction Property in the KL Case). *In the case of Eq. (9), the Bellman operator $\mathcal{T}_{\nu^{soft}}^\pi$ possesses the $\gamma$-contraction property.*

The full derivation of the Lemma 4.7 and the Theorem 4.8 is provided in Appendix B.2.

For the practical implementation, similar to the policy improvement in SAC (Haarnoja et al., 2018a;b), we prepare variational model, $\nu_{\text{model}}$, then update this model so as to be similar distribution as $\nu_\pi^{\star soft}$:

$$L(\nu_{\text{model}}) = \mathbb{E}_{s \sim \rho_R} \left[ D_{KL} \left( \nu_{\text{model}} \parallel \frac{p \exp\left(-V^\pi/\alpha_{attk}\right)}{Z} \right) \right]$$
$$\propto \mathbb{E}_{\rho_R} \left[ \mathbb{E}_{\nu_{\text{model}}} \left[ \alpha_{attk} \log \nu_{\text{model}}(\tilde{s} \mid s) + V^\pi + const. \right] \right]. \tag{10}$$

We refer to this strategy as Virtual Alternative Training with the Soft-Worst method (VALT-SOFT). See Appendix C.2 for more details about the implementation.

**Approach 2: Approximation-Based Numerical Solution.**
Inspired by the work (Belousov & Peters, 2017; 2019), we can transform Eq. (5) into non-minimization form as:

**Lemma 4.9.** *(Dual Form of Soft Constrained Adversary)*

$$\nu_\pi^{\star soft}(\tilde{s} \mid s) = p(\tilde{s} \mid s)(f^*)' \left( \frac{-V^\pi - \lambda^\star I_{\tilde{s}} + \kappa(\tilde{s} \mid s)}{\alpha_{attk}} \right),$$
(11)

where $(f^*)'$ represents a differential function of the conjugated function for $f$, $\lambda^\star$ means Lagrangian variable, $I_{\tilde{s}}$ is an unit vector for the observation space, and $\kappa$ is a complemental slackness variable.

This result shows that the solution distribution of Eq. (5) is shaped based on the prior $p$, and that $f$, $\alpha_{attk}$, and $V_{\pi_{fix}}^\nu$ determine how the probability mass is distributed over the space $\tilde{S}$. Here, we focus on $\alpha$-divergent cases, which are a subclass of $f$-divergences, allowing us to control the tendency of the distribution's shape. When $\alpha \ll 1$, the distribution concentrates strongly on the worst-case scenarios while remaining flat elsewhere. We can approximate this distribution by using the mode probability, denoted as $\kappa_{worst}$, and the flat probability for others as follows:

$$\nu_\pi^{\star soft}(\tilde{s}|s) \simeq \begin{cases} \kappa_{worst} + \frac{1-\kappa_{worst}}{|\tilde{S}_\epsilon|}, & \text{if } \tilde{s} = \underset{\tilde{s}' \in \mathcal{B}_\epsilon}{\arg\min} \, V^\pi(s, \tilde{s}'), \\ \frac{1-\kappa_{worst}}{|\tilde{S}_\epsilon|}, & \text{otherwise,} \end{cases}$$
(12)

where $|\tilde{S}_\epsilon|$ represents the measure of the observation state space within the $\epsilon$-bounded domain. Also this case satisfies the required condition for the Theorem 4.6, otherwise

**Theorem 4.10** (Contraction Property in the Epsilon Case). *In the case of Eq. (12), the Bellman operator $\mathcal{T}_{\nu^{soft}}^\pi$ possesses the $\gamma$-contraction property.*

A derivation of the Lemma 4.9 and a proof of the Theorem 4.10 are provided in Appendix B.3. In the practical use case, Eq. (12) can be approximated by combining the uniform distribution with a numerical gradient approach, similar to the *Projected Gradient Decent (PGD)* attack (Madry et al., 2018; Pattanaik et al., 2018; Zhang et al., 2020b). We refer to this strategy as Virtual Alternative Training with the Epsilon-Worst method (VALT-EPS). See Appendix C.1 for more details about the implementation.

### 4.4. Policy Improvement with a Fixed Adversary

Based on the preceding discussion, we demonstrate that the soft (worst-case) adversary can be determined without requiring additional interactions or extra value estimations. In the final phase, we fix the adversary and enhance the agent's policy using a modified version of the maximum-entropy policy improvement approach.

**Proposition 4.11** (Policy Improvement with a Fixed Adversary)**.**

$$\pi_{new} = \underset{\pi \in \Pi}{\arg\min} \, D_{KL}(\pi \circ \nu_{fix} \parallel \pi_{old}^\star \circ \nu_{fix}),$$

$$s.t. \; \pi_{old}^\star = \frac{\exp\left(Q_{\nu_{fix}}^{\pi_{old}}(s, \tilde{a})/\alpha_{ent}\right)}{Z}, \forall s \in \mathcal{S}.$$
(13)

To confirm the improvement achieved by this operation, we present the following theorem:

**Theorem 4.12** (Policy Improvement Theorem with a Fixed Adversary)**.** *Given a fixed adversary $\nu$ and a policy $\pi$, define a new policy $\pi_{new}$ such that for all states $s$ as Eq. (13). assuming that $Q_\nu^\pi$ and $\sum_{a \in \tilde{A}} \exp(Q_\nu^\pi(s, a))$ are bounded for all states $s$, it follows that*

$$Q_\nu^{\pi_{new}}(s, \tilde{a}) \geq Q_\nu^\pi(s, \tilde{a})$$
(14)

*for all actions $a$ and states $s$.*

*Proof.* See Appendix B.4. □

**Considerations for Behavior Policy.** To calculate Eq. (13), we use data from a replay buffer $R$ as: $\pi_{new} = \arg\min_{\pi \in \Pi} \mathbb{E}_{\rho_R}[D_{KL}]$. For collecting $R$, we use a uniform distribution $p$ as a proxy due to the high computational cost of PGD in VALT-EPS. In the HalfCheetah environment, we rely entirely on states with adversarial noise, whereas for the high-dimensional input task, Ant, training VALT-SOFT with entirely adversarial state inputs leads to corruption due to over-pessimism. We found that a 50:50 mix of $\pi(\cdot \mid s)$ and $\pi \circ \nu(\cdot \mid s)$ generally works well. A key challenge in off-policy robust learning remains: the mismatch between the data distribution $\rho_R$ and the distribution induced by the current behavior policy. Future work involves addressing this mismatch by refining adversarial intensity and improving methods for handling off-policy data more effectively, while accounting for the progress of learning. Appendix E.3 analyzes how the adversarial ratio in the behavior policy affects performance.

## 5. Experiments

In this section, we set up experiments to confirm our algorithm achieve a good sample efficiency and robustness especially against the adversaries. We use four OpenAI Gym MuJoCo environments (Brockman et al., 2016; Todorov et al., 2012): Hopper, HalfCheetah, Walker2d, and Ant, as utilized in most prior works (Zhang et al., 2020b; 2021; Oikarinen et al., 2021; Liang et al., 2022).

### 5.1. Baselines and Implementations.

We select SAC (Haarnoja et al., 2018a;b) as our base off-policy method. Since existing studies have not reported

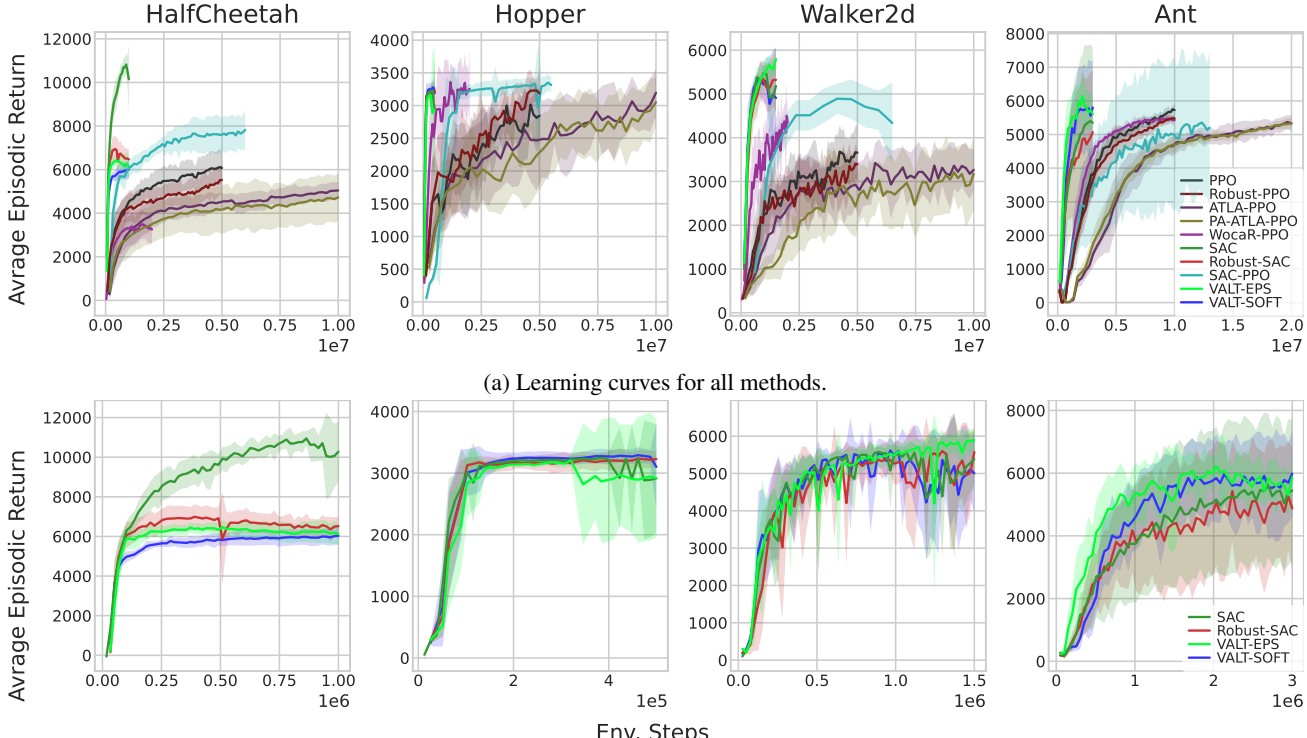

(a) Learning curves for all methods.

(b) Learning curves focusing only on sample-efficient SAC variants.

*Figure 2.* Learning curves for four MuJoCo control tasks evaluated under no-attack settings. Subfigure 2a shows results for all algorithms, while Subfigure 2b focuses on sample-efficient off-policy methods. Solid lines indicate average evaluation scores, and shaded areas represent standard deviations across different random seeds. For simplicity, we abbreviate +*reg.*, +*wreg.*, and "PA-", and omit "-SAC" unless explicitly stated. Note that these figures illustrate training performance and sample efficiency under clean conditions, while robustness against attacks is evaluated separately in Table 1.

the application of the latest off-policy Actor-Critic methods[2], particularly SAC, we incorporated these methods into our SAC implementation and tuned the hyper-parameters appropriately. Additionally, for comparison, we included a representative robustness-enhancing method based on the on-policy *Proximal Policy Optimization (PPO)* algorithm (Schulman et al., 2017) in our experiments.

**Comparison with On-Policy (PPO) Settings.** We evaluate five representative methods: (1) Original PPO (**PPO**) (Schulman et al., 2017); (2) **Robust-PPO**, which incorporates a robust regularization term into PPO, as proposed by Zhang et al. (2020b; 2021); (3) **ATLA-PPO**, an adversarial training approach based on Eq. (3), introduced by Zhang et al. (2021); (4) **PA-ATLA-PPO**, a variant of ATLA-PPO designed to operate in a lower-dimensional space than the observation space (Sun et al., 2022); (5) **WocaR-PPO**, which integrates the worst-case-aware action-value function method proposed by Liang et al. (2022).

For fairness, we use the LSTM-based PPO when available in the provided code.

**Comparison with Off-Policy (SAC) Settings.** We prepare four baseline algorithms for off-policy settings: (1) the original SAC (**SAC**); (2) **Robust-SAC**, a SAC version of the robust regularizer proposed in Zhang et al. (2020b); and (3) **SAC-PPO**, an off-policy max-min alternative training method similar to ATLA-PPO, but using SAC as the agent and PPO as the adversary.

SAC-PPO is a novel method we introduced specifically for this comparison, offering a theoretically sound alternative training for off-policy algorithms, although it suffers from sample inefficiency. In the Ant environment, SAC-PPO is combined with the PA-ATLA technique (Sun et al., 2022) to address the large observation space; we refer to this variant as PA-SAC-PPO.

We also attempted to adapt the WocaR technique to the off-policy setting (WocaR-SAC), but it failed to perform consistently across the four benchmark tasks. The corresponding experiments and analysis are described in the paragraph

---

[2]While DDPG is implemented in Zhang et al. (2020b), it appears that DDPG does not perform adequately on the four MuJoCo benchmarks.

*Table 1.* Average episodic rewards (± standard deviation) for the median seed models of our proposed methods (VALT-EPS, VALT-SOFT) and other baselines across four MuJoCo tasks. For learning-based attacks, attack parameters are consistently optimized, and the minimum scores are reported. All evaluations are conducted over twenty episodes with different random seeds. The highest scores within each on-policy or off-policy category are **highlighted in bold**, and among the most sample-efficient methods, the highest-scoring ones are shaded in gray . As a robustness metric, we summarize the lowest score observed across all attacks in the rightmost column.

| Environment | Method | Env. Steps | Natural Reward | Uniform | MAD | PGD (minV) | PGD (minQ) | RS | SA-RL (SAC) | PA-AD (SAC) | SA-RL (PPO) | PA-AD (PPO) | Worst score across attacks |
|---|---|---|---|---|---|---|---|---|---|---|---|---|---|
| **HalfCheetah** state-dim: 17 action-dim: 6 $\epsilon = 0.15$ | PPO | 5M | **5917±136** | **5674±139** | 3821±248 | 3447±143 | - | 4448±134 | -128±31 | -65±58 | -61±86 | -21±77 | -128±31 |
| | Robust-PPO | 5M | 5613±146 | 5510±141 | **4886±125** | **4792±63** | - | 3313±1197 | 3715±122 | 3408±903 | 2839±1492 | 3734±625 | 2839±1492 |
| | ATLA-PPO +*reg.* | 10M | 5082±106 | 5103±133 | 4813±110 | 4320±979 | - | **4520±136** | **5232±957** | **4048±323** | **3504±720** | 3795±873 | **3504±720** |
| | PA-ATLA-PPO +*reg.* | 10M | 5137±65 | 5022±99 | 4759±112 | 4719±80 | - | 4424±210 | 3235±1058 | 3901±206 | 3306±611 | **4063±941** | 3235±1058 |
| | WocaR-PPO +*wreg.* | 2M | 4388±99 | 4198±465 | 3245±1254 | 4127±747 | - | 3875±106 | 1895±1097 | 1566±1081 | 2630±1289 | 2643±1214 | 1566±1081 |
| | SAC | 1M | **10985±117** | 6411±273 | 3288±154 | - | 3424±171 | 4215±247 | -51±267 | 1757±409 | 936±465 | 1171±214 | -51±267 |
| | Robust-SAC | 1M | 6350±68 | 6127±84 | 5837±112 | - | 2601±311 | 3183±375 | 2026±417 | 3481±383 | 3197±532 | 3367±468 | 2026±417 |
| | SAC-PPO +*reg.* | 6M | 7487±147 | **7154±141** | **6030±1084** | - | 1823±871 | 5836±138 | 3206±112 | 2052±835 | 2849±726 | 3008±303 | 1823±871 |
| | **VALT-EPS-SAC** +*reg.* | 1M | 6027±112 | 5900±84 | 5850±98 | - | **4647±180** | 5834±107 | 5216±159 | 5396±113 | 5707±100 | 5695±114 | **4647±180** |
| | **VALT-SOFT-SAC** +*reg.* | 1M | 5881±45 | 5798±55 | 5448±93 | - | 4480±83 | 5434±80 | 3584±61 | 4449±69 | 4722±83 | 4868±87 | 3584±61 |
| **Hopper** state-dim: 11 action-dim: 3 $\epsilon = 0.075$ | PPO | 5M | 3018±879 | 3122±743 | 2463±742 | 3569±40 | - | 1360±227 | 2828±1047 | 2683±1050 | 871±146 | 1460±510 | 871±146 |
| | Robust-PPO | 5M | 3617±71 | 3620±66 | 3236±688 | 3559±117 | - | **3514±493** | **3686±14** | **3278±677** | 1528±353 | 2188±909 | 1528±353 |
| | ATLA-PPO +*reg.* | 10M | 3484±333 | 3581±52 | **3445±414** | 3453±64 | - | 2158±431 | 2602±1311 | 2508±1334 | **1806±665** | 2050±477 | **1806±665** |
| | PA-ATLA-PPO +*reg.* | 10M | 2522±919 | 2666±819 | 2683±811 | 3417±1043 | - | 1301±322 | 2739±1043 | 3088±1162 | 1020±134 | 1530±511 | 1020±134 |
| | WocaR-PPO +*wreg.* | 2M | **3689±9** | **3643±243** | 2701±811 | **3688±13** | - | 1575±410 | 2477±849 | 2786±704 | 1615±541 | **3039±987** | 1575±410 |
| | SAC | 0.5M | 3199±1 | 3213±6 | 3194±391 | - | 3281±12 | 3337±5 | 2979±1 | 3166±11 | 3034±1 | 3070±2 | 2979±1 |
| | Robust-SAC | 0.5M | 3237±5 | 3234±8 | 3237±9 | - | 2947±6 | 3290±2 | 2990±10 | 3114±26 | 2952±4 | 3015±3 | 2947±6 |
| | SAC-PPO | 5.5M | 3359±1 | 3368±4 | **3479±17** | - | **3528±12** | 3527±6 | 3203±1 | **3286±12** | 3198±2 | 3236±6 | 3198±2 |
| | **VALT-EPS-SAC** +*reg.* | 0.5M | 3297±2 | 3297±3 | 3293±7 | - | 3264±17 | 3358±2 | **3267±2** | 3255±9 | 3212±4 | 3264±6 | 3212±4 |
| | **VALT-SOFT-SAC** +*reg.* | 0.5M | 3228±2 | 3230±10 | 3241±15 | - | 2986±3 | 3462±3 | 3034±3 | 3102±9 | 3059±7 | 3031±4 | 2986±3 |
| **Walker2d** state-dim: 17 action-dim: 6 $\epsilon = 0.05$ | PPO | 5M | 3895±802 | 3953±968 | 3921±724 | 3847±857 | - | 2140±1256 | 540±1203 | 624±1265 | 1961±1087 | 2098±1057 | 540±1203 |
| | Robust-PPO | 5M | 4323±6 | 4320±8 | 4305±15 | 4162±12 | - | **4097±7** | 293±900 | 281±898 | 4101±19 | 3967±889 | 281±898 |
| | ATLA-PPO | 10M | 3948±84 | 3977±103 | 4003±66 | 3813±344 | - | 3425±1062 | 2359±1721 | 2406±1683 | 3416±941 | 3284±1332 | 2359±1721 |
| | PA-ATLA-PPO +*reg.* | 10M | 3370±1169 | 3534±1145 | 3590±1117 | 4156±156 | - | 2341±919 | 2574±1801 | 1625±1791 | 1765±995 | 1834±909 | 1625±1791 |
| | WocaR-PPO +*wreg.* | 2M | **4579±127** | **4340±835** | **4630±106** | **4391±831** | - | 3791±1045 | **4238±523** | 4127±319 | 4441±723 | 4376±82 | **3791±1045** |
| | SAC | 1.5M | 5866±47 | 5601±1086 | 4288±2280 | - | 3550±2427 | 2732±2710 | 2143±2452 | 337±567 | 1911±2525 | 3252±2773 | 337±567 |
| | Robust-SAC | 1.5M | 5840±21 | 5824±41 | **5784±63** | - | 5180±808 | **5766±20** | 5432±842 | 5552±26 | 5645±50 | 5644±67 | **5180±808** |
| | SAC-PPO | 6.5M | 3493±1473 | 4004±1423 | 4365±1501 | - | 2423±1536 | 3266±1145 | 1812±153 | 1343±178 | 1609±528 | 1695±1666 | 1343±178 |
| | **VALT-EPS-SAC** | 1.5M | 5901±56 | 5904±50 | 5587±1158 | - | 4777±1932 | 5486±1146 | 5820±913 | 3442±2428 | 4883±2066 | 3993±2371 | 3442±2428 |
| | **VALT-SOFT-SAC** | 1.5M | 5604±24 | 5609±27 | 5579±55 | - | 5327±925 | 5490±71 | 5249±18 | 4981±1567 | 5485±19 | 5468±20 | 4981±1567 |
| **Ant** state-dim: 111 action-dim: 8 $\epsilon = 0.15$ | PPO | 10M | **5628±440** | **5586±305** | 2968±1391 | 3813±147 | - | 1975±1788 | 588±735 | -404±139 | 511±30 | 143±79 | -404±139 |
| | Robust-PPO | 10M | 5543±89 | 5463±116 | **5333±117** | **5358±223** | - | 4249±1976 | 4965±558 | **3301±142** | **3196±83** | **3273±107** | **3196±83** |
| | ATLA-PPO +*reg.* | 20M | 5433±97 | 5374±100 | 5298±153 | 5193±101 | - | 3825±1553 | **5247±142** | 3054±166 | 2726±58 | 2408±98 | 2408±98 |
| | PA-ATLA-PPO +*reg.* | 20M | 5350±104 | 5318±122 | 5298±110 | 5034±903 | - | **5089±683** | 4950±1044 | 3231±241 | 2700±133 | 2726±145 | 2700±133 |
| | WocaR-PPO +*wreg.* | 10M | 5501±182 | 5480±174 | 4496±1035 | 5296±140 | - | 4821±949 | 2847±1729 | -637±195 | 1526±123 | 896±162 | -637±195 |
| | SAC | 3M | 6583±1852 | 996±1076 | -27±74 | - | -23±53 | -106±370 | -1953±1213 | -1047±700 | -725±322 | -7±102 | -1953±1213 |
| | Robust-SAC | 3M | 6790±1548 | 6117±2226 | 4847±2578 | - | 1687±1125 | 937±899 | 3171±2529 | 68±1016 | 1045±1135 | 3351±2085 | 68±1016 |
| | PA-SAC-PPO +*reg.* | 13M | 6351±1730 | 6308±1518 | **5680±1404** | - | 1778±693 | 2760±1198 | 4169±2144 | -354±468 | 3754±691 | 4077±2110 | -354±468 |
| | **VALT-EPS-SAC** +*reg.* | 3M | 6332±748 | 5129±2397 | 5019±1923 | - | 1487±1176 | **3356±2408** | **4710±2319** | 4944±1891 | 1738±1466 | **4585±2022** | 1487±1176 |
| | **VALT-SOFT-SAC** +*reg.* | 3M | 6719±66 | **6552±260** | 5287±2035 | - | **2757±1297** | 3088±1521 | 3808±1718 | **5152±1653** | 4281±167 | 3684±2590 | **2757±1297** |

"WocaR-SAC Settings" in Appendix D.2.

**Notation for Regularization Settings.** Apart from the original PPO and SAC, robust regularization techniques are applicable and are used in most of the proposed methods. We denote methods that use the robust regularizer from Zhang et al. (2020b) with the suffix (+*reg.*), and those that utilize the robust regularizer from Liang et al. (2022) with the suffix (+*wreg.*).

Additional details on the implementations and hyperparameters are provided in Appendix D.1 and Appendix D.2.

### 5.2. Attacker Settings

We adopt the same attack scales commonly used in previous studies (Zhang et al., 2020b; 2021; Oikarinen et al., 2021; Sun et al., 2022; Liang et al., 2022): $\epsilon = 0.15, 0.075, 0.05, 0.15$ for HalfCheetah, Hopper, Walker2d, and Ant, respectively.

**Heuristic Attacks.** For robustness evaluation, we employ several heuristic attack methods: Random (**Uniform**), *Max-*

*ActionDiff* (**MAD**) (Zhang et al., 2020b), *PGD* (Madry et al., 2018; Pattanaik et al., 2018; Zhang et al., 2020b), and *RobustSarsa* (**RS**) (Zhang et al., 2020b). For *PGD*, we use distinct notations to specify the target: **PGD (minQ)** for SAC and **PGD (minV)** for PPO.

**Learning-Based Optimal Adversary Attacks.** We further incorporate more advanced attackers, including **SA-RL (PPO)** (Zhang et al., 2021) and **PA-AD (PPO)** (Sun et al., 2022). These attackers are also evaluated with the SAC algorithm, where they are denoted as **SA-RL (SAC)** and **PA-AD (SAC)**, respectively.

Those details of the implementations and hyper-parameters are provided in Appendix D.3.

### 5.3. Result

**Training Sample Efficiency.** Fig. 2a presents the training curves for our proposed methods (VALT-EPS, VALT-SOFT) and other baselines. The x-axis represents the number of environment interaction steps, while the y-axis shows the average episodic return during unperturbed evaluation.

Most PPO-based algorithms require at least 5M steps for HalfCheetah, Hopper, and Walker2d, and 10M steps for Ant. Alternative training-based methods (ATLA-PPO, PA-ATLA-PPO) need approximately twice as many steps as other PPO-based approaches due to their iterative training processes. Although WocaR-PPO is highly efficient due to the absence of adversary training, it still requires 2M, 2M, 2M, and 10M steps for HalfCheetah, Hopper, Walker2d, and Ant, respectively.

In contrast, Fig. 2b focuses on sample-efficient off-policy methods, excluding SAC-PPO. SAC-based algorithms demonstrate superior efficiency, achieving training within 1M, 0.5M, 1.5M, and 3M steps for HalfCheetah, Hopper, Walker2d, and Ant, respectively. This corresponds to a 3x-10x improvement over standard PPO-based methods and a 1.5x-4x improvement over WocaR-PPO, the most efficient on-policy method known to us. Furthermore, VALT-EPS and VALT-SOFT achieve comparable or even more stable training across different seeds, particularly for Ant.

**Robustness Evaluation.** Table 1 summarizes the evaluation results of the four benchmarks across all algorithms, with the environment interaction steps provided in the third column from the left. To facilitate an easy comparison of robustness among the methods, the worst attacked scores are presented in the rightmost column. Across all four benchmarks, our proposed methods, VALT-EPS and VALT-SOFT, exhibit exceptional robustness, particularly in the more challenging tasks of HalfCheetah and Ant.

SAC-PPO, which can be seen as a natural formulation of the alternative framework, demonstrates more robust performance compared to the original SAC. However, its robustness remains moderate when compared to Robust-SAC. Interestingly, our proposed methods (VALT-EPS and VALT-SOFT) outperform not only the simple regularization-based method (Robust-SAC), but also SAC-PPO, despite the latter's potential to naturally implement the alternative formulation.

We hypothesize that although the natural alternative approach aims to reach a Nash equilibrium, its practical implementation is hindered by the difficulty of finding a sharp saddle point, which requires carefully balancing the learning dynamics of both the agent and the adversary (Reddi et al., 2024). In contrast, our objective function, defined in Eq. (6), not only addresses the sample inefficiency of existing methods, but also reformulates the problem as a Quantal Response Equilibrium (QRE) (McKelvey & Palfrey, 1995; Reddi et al., 2024), leading to a more stable learning process for discovering robust equilibrium solutions.

SAC-based variants exhibit three notable characteristics that distinguish them from PPO-based variants.

First, SAC-based methods demonstrate superior sample efficiency. Under clean (unperturbed) environments, they achieve higher episodic rewards with fewer environment interactions than PPO-based methods, primarily due to the data efficiency of off-policy learning.

Second, SAC-based variants are more sensitive to the PGD (minQ) heuristic attack. This effect is especially pronounced for methods based on alternating training, such as SAC-PPO, VALT-EPS, and VALT-SOFT. This sensitivity arises because SAC maintains an explicit action-value function that estimates long-term returns under the agent's policy. When these Q-functions are adversarially degraded (e.g., by minimizing the predicted value), performance drops significantly—often approximating that of an optimal adversary. We further analyze this phenomenon in Appendix E.2 by ablating the adversarial effect during policy evaluation.

Third, SAC-based variants—including vanilla SAC and Robust-SAC—tend to exhibit strong robustness on tasks such as Hopper and Walker2d. Eysenbach & Levine (2022) showed that the maximum entropy framework naturally induces robustness, with theoretical justification for resilience against dynamics and reward variations. While our study focuses on observation robustness, we note that observation perturbations can be interpreted as modifying the agent's perceived dynamics, which can be formalized as $\mathcal{F}' = (\nu \circ \mathcal{F})(\tilde{s}' \mid s, \tilde{a})$, where the true dynamics remain unchanged, but the agent cannot distinguish between perturbed and unperturbed states due to limited observability.

The task dependency of robustness and the suitability of specific methods across tasks could be further systematized theoretically. This represents an important direction for future research, particularly in practical applications.

## 6. Conclusion and Future Work

We propose a novel approach that optimizes the corresponding adversary without requiring additional interactions with the environment, making it suitable for off-policy settings. Our algorithm is grounded in a solid theoretical foundation, and evaluation results demonstrate both excellent sample efficiency and consistent success.

For future work, we highlight three directions. First, the algorithm can be improved by integrating recent advances in functional smoothness to enhance robustness. Second, the framework may be extended to other domains, such as domain randomization. Third, a promising direction is to explore adaptive correction methods that leverage past replay data and adapt to current environments, enabling more efficient learning in settings like offline RL, curriculum learning, and online domain adaptation.

## Acknowledgements

We sincerely appreciate the cooperation of the conference committee and the constructive comments and suggestions from the reviewers, which have significantly improved the quality of this paper. We also acknowledge the open-source software and libraries used in this research, including PyTorch, OpenAI Gym, MuJoCo, Stable-Baselines3, auto_LiRPA, as well as foundational work on robust reinforcement learning.

We would like to thank Satoshi Yamamori and Sotetsu Koyamada for their valuable advice on this research. We also extend our sincere appreciation to all members of the Computer Science Domain at Honda R&D Co., Ltd., especially Ken Iinuma, Akira Kanahara, Kenji Goto, Umiaki Matsubara, Ryoji Wakayama, Keisuke Oka, and Koki Aizawa, for their cooperation and support.

This research was supported by the authors' affiliation, Honda R&D Co., Ltd., and Kyoto University, Japan. It was partially funded by JSPS KAKENHI (No. 22H04998) and NEDO (No. JPNP20006) and made use of the supercomputer system provided by the Institution for Information Management and Communication (IIMC) of Kyoto University.

## Impact Statement

Our research primarily focuses on proposing robust RL methods, which we believe will significantly advance the practical applications of RL. However, it is important to note that robustness methods, including those proposed in our methods, require additional computational resources compared to the original RL algorithms. Consequently, utilizing these methods in scenarios where robustness is not a critical requirement may result in increased computational costs. Furthermore, as with RL in general, there exists a potential risk of misuse in applications involving automation and optimization. Nonetheless, we believe that the positive utility of assisting humanity in creating a better society far outweighs these concerns.

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

# APPENDIX

We describe additional related works, provide more detailed theoretical derivations, offer additional evaluation results, and deeper discussions on the future research direction to help the reader gain a deeper understanding of our work. In the following, we temporarily set aside strict notation and represent expressions like "$\int_{a \in \mathcal{A}} \cdot, da$" as "$\sum_{a \in \mathcal{A}} \cdot$" when discussing continuous action space.

## A. Additional Related Work

### A.1. f-Divergence Constrained Methods.

The use of optimization methods constrained by f-divergence has spanned various contexts from historical applications to recent advancements. Historically, these methods were utilized for relative entropy maximization in inverse RL (Boularias et al., 2011). In offline RL and imitation learning (IL), they primarily serve to constrain divergence from the probability density distribution of state-action pairs within the dataset (Xu et al., 2023; Kostrikov et al., 2020; Nachum & Dai, 2020; Kim et al., 2021; 2022; Ma et al., 2022). Moreover, methods that limit update intervals by distribution constraints (old policy, prior) during policy updates (Peters et al., 2010; Fox et al., 2016; Schulman et al., 2015; 2017; Belousov & Peters, 2017; 2019) are closely related to our proposed soft-constrained adversary approach.

### A.2. Additional Details on Robust RL for Adversarial Dynamics

We additionally discuss robustness in dynamics, particularly focusing on adversarial training due to its diversity and importance.

Broadly, robust RL can be categorized along two axes: (1) the *target of uncertainty or perturbation* (e.g., observation, dynamics, action, reward), which determines the appropriate modeling framework and optimization strategy; and (2) the *treatment of adversariality*, which varies depending on theoretical assumptions.

For the second axis, some approaches seek formal guarantees via Nash equilibria or Stackelberg games with fixed update orderings, while others—including our work—adopt *alternating training* as a practical approximation. Beyond hard max-min formulations, recent studies have also explored *relaxed, risk-sensitive objectives* such as entropic risk and value-at-risk, bridging connections to risk-aware MDPs and RLs (Howard & Matheson, 1972; Filar et al., 1989; 1995; Osogami, 2012; Chow & Ghavamzadeh, 2014; Chow et al., 2015; 2018; Ying et al., 2022).

This area has become an active focus of research, with recent surveys aiming to systematically organize and compare diverse formulations and assumptions in robust RL (Moos et al., 2022; Schott et al., 2024).

**Adversarial Disturbances**   To consider the pessimistic dynamics scenario for the complex model-free settings beyond the tabular or linear cases (Nilim & El Ghaoui, 2005; Iyengar, 2005), external forces are introduced to the dynamics model, modified from $\mathcal{F}(s' \mid s, a)$ to $\mathcal{F}(s' \mid s, a_1, a_2)$. Here $a_1$ is the agent action and $a_2$ is the adversarial action, which is assumed to be chosen by an adversary which decides the action from the same observation $s$.

In the early stages, Morimoto & Doya (2005) proposed a robust RL method that accounts for input disturbances and modeling errors by introducing an external adversary and applying a robust control method based on $H^{\infty}$ theory (RRL). Pinto et al. (2017) extended this approach by incorporating DRL approximators, specifically TRPO (Schulman et al., 2015), and proposed Robust Adversarial RL (RARL), which enhances robustness against dynamic perturbations from the training environment and external forces. Their approach consists of two RL components: an on-policy agent RL and an on-policy adversarial RL, resembling the alternating training on state observations (ATLA) (Zhang et al., 2021). Subsequent studies reported that RARL sometimes converges to suboptimal solutions and addressed this issue using gradient-based techniques (Kamalaruban et al., 2020). Reddi et al. (2024) proposed a novel method, Quantal Adversarial RL (QARL), which mitigates this suboptimality by relaxing the hardmin-hardmax formulation through entropy regularization. Although its formulation and approach resemble VALT-SOFT-SAC, a significant difference remains: QARL still relies on alternating training between the agent and the adversary because it assumes a two-player zero-sum Markov game with two action inputs for the dynamics, $\mathcal{F}(s' \mid s, a_1, a_2)$. More recently, Dong et al. (2025) proposed a method that considers an ensemble of adversarial disturbers (typically around 10) instead of a single worst-case adversary, and updates the distribution using divergence-constrained optimization or percentile-based criteria.

Interestingly, these methods improve not only robustness to adversarial disturbances but also generalization to dynamic variations, such as changes in friction or mass.

**Dynamics Parameter** As another form of dynamics modification, closely related to the previous paragraph, many studies consider modeling the environment as $\mathcal{F}(s' \mid s, a, \omega)$, where $\omega$ represents uncertain physical parameters such as mass or friction coefficients. These works assume the ability to control such parameters in simulation or collect diverse training data across different settings. By explicitly manipulating these parameters, the agent can learn to be robust to dynamics uncertainty.

One simple approach is *Dynamics Randomization* (Peng et al., 2018), which maximizes expected performance over a distribution of environment parameters. To avoid local minima, Vinitsky et al. (2020) propose using multiple concurrent adversaries to introduce diverse perturbations. Another line of work (Tanabe et al., 2022) incorporates the effect of uncertain parameters into the $Q$-function, and applies gradient-based optimization to find perturbations that degrade the value function.

Compared to the external-force-based methods described earlier, these parameter-based methods typically assume that the environment parameters (i.e., $\omega$) remain fixed within each episode. This assumption helps avoid overly conservative policies, as the agent can adapt across episodes without facing highly variable dynamics within a single rollout.

### A.3. Constrained MDP on State Observation.

There are similar problem settings that address observation robustness, considering them within the framework of safe Constrained Markov Decision Processes (CMDP) (Altman, 1999). In this setting, the agent is required to maximize the accumulated reward while simultaneously ensuring that a separately defined cost measure remains below a certain threshold.

Yang et al. (2021) proposed an off-policy method, Worst-Case Soft Actor-Critic (WCSAC), which efficiently solves the problem of maintaining the accumulated cost constraint below a predefined limit. Their approach utilizes the Lagrangian-constrained SAC framework (Ha et al., 2021) to enforce cost constraints effectively.

Furthermore, Liu et al. (2023) conducted additional analyses and introduced two novel types of adversarial attacks: one that *maximizes the cost* and another that *maximizes the reward while violating the safety constraints*. Notably, the latter attack is particularly stealthy and highly effective in compromising agent safety without significantly altering the reward signal.

While these studies are crucial contributions to the field, our problem setting specifically considers an adversary that *directly hinders the maximization of the accumulated reward sum* rather than enforcing cost constraints. Thus, we do not include them as direct points of comparison in our study.

## B. Details of Derivation and Proof

### B.1. Symmetric Property of Policy Evaluation

**Proof of Theorem 4.6.** Recall that Eq. (7) satisfies the $\gamma$-contraction property. Thus, we can express the following relationship:

$$
\begin{aligned}
\left\| (\mathcal{T}_{\nu^{\mathrm{soft}}}^{\pi} Q_1^{\pi})(s_t, \tilde{a}_t) - (\mathcal{T}_{\nu^{\mathrm{soft}}}^{\pi} Q_2^{\pi})(s_t, \tilde{a}_t) \right\|_{s_t, \tilde{a}_t} &= \gamma \left\| \mathbb{E}_{s_{t+1} \sim \mathcal{F}} \left[ \min_{\nu \in \mathcal{N}} V_{\nu^{\mathrm{soft}}, 1}^{\pi}(s_{t+1}) - \min_{\nu \in \mathcal{N}} V_{\nu^{\mathrm{soft}}, 2}^{\pi}(s_{t+1}) \right] \right\|_{s_t, \tilde{a}_t} \\
&\leq \gamma \left\| \min_{\nu \in \mathcal{N}} V_{\nu^{\mathrm{soft}}, 1}^{\pi}(s_{t+1}) - \min_{\nu \in \mathcal{N}} V_{\nu^{\mathrm{soft}}, 2}^{\pi}(s_{t+1}) \right\|_{s_{t+1}} \\
&\leq \gamma \left\| Q_1^{\pi}(s_{t+1}, \tilde{a}_{t+1}) - Q_2^{\pi}(s_{t+1}, \tilde{a}_{t+1}) \right\|_{s_{t+1}, \tilde{a}_{t+1}},
\end{aligned}
\tag{15}
$$

here $\| \dots \|_{s_t, \tilde{a}_t}$ denotes the max norm over $s_t, \tilde{a}_t$. Recall the definitions of $V_{\nu^{\mathrm{soft}}, .}^{\pi}(s_{t+1})$ and $V_{\pi, .}^{\nu^{\mathrm{soft}}}(s_{t+1})$:

$$
\begin{aligned}
V_{\nu^{\mathrm{soft}}, .}^{\pi}(s_{t+1}) &= \mathbb{E}_{\nu} \left[ \mathbb{E}_{\pi} \left[ Q^{\pi}(s_{t+1}, \tilde{a}_{t+1}) \right] \right] + \alpha_{ent} \mathcal{H}(\pi \circ \nu) + \alpha_{attk} D_f(\nu \| p), \\
V_{\pi, .}^{\nu^{\mathrm{soft}}}(s_{t+1}) &= \mathbb{E}_{\nu} \left[ Q^{\nu}(s_{t+1}, \tilde{s}_{t+1}) \right] - \alpha_{ent} \mathcal{H}(\pi \circ \nu) - \alpha_{attk} D_f(\nu \| p).
\end{aligned}
\tag{16}
$$

Here, we define two distinct action-value functions: $Q_1^{\nu}(s, \tilde{s}) = \mathbb{E}_{\pi}[-Q_{1'}^{\pi}(s, \tilde{a})]$ and $Q_2^{\nu}(s, \tilde{s}) = \mathbb{E}_{\pi}[-Q_{2'}^{\pi}(s, \tilde{a})]$. We consider the operation $\mathcal{T}_{\pi}^{\nu^{\mathrm{soft}}}$,

$$\left\| (\mathcal{T}_\pi^{\nu^{\text{soft}}} Q_1^\nu)(s_t, \tilde{s}_t) - (\mathcal{T}_\pi^{\nu^{\text{soft}}} Q_2^\nu)(s_t, \tilde{s}_t) \right\|_{s_t, \tilde{s}_t} = \gamma \left\| \mathbb{E}_{s_{t+1} \sim \mathcal{F} \circ \pi} \left[ \max_{\nu \in \mathcal{N}} V_{\pi,1}^{\nu^{\text{soft}}}(s_{t+1}) - \max_{\nu \in \mathcal{N}} V_{\pi,2}^{\nu^{\text{soft}}}(s_{t+1}) \right] \right\|_{s_t, \tilde{s}_t}$$

$$\underset{(1)}{=} \gamma \left\| \mathbb{E}_{s_{t+1} \sim \mathcal{F} \circ \pi} \left[ \min_{\nu \in \mathcal{N}} V_{\nu^{\text{soft}}, 1'}^{\pi}(s_{t+1}) - \min_{\nu \in \mathcal{N}} V_{\nu^{\text{soft}}, 2'}^{\pi}(s_{t+1}) \right] \right\|_{s_t, \tilde{s}_t} \tag{17}$$

$$\leq \gamma \left\| \min_{\nu \in \mathcal{N}} V_{\nu^{\text{soft}}, 1'}^{\pi}(s_{t+1}) - \min_{\nu \in \mathcal{N}} V_{\nu^{\text{soft}}, 2'}^{\pi}(s_{t+1}) \right\|_{s_{t+1}}$$

$$\leq \gamma \left\| Q_{1'}^{\pi}(s_{t+1}, \tilde{a}_{t+1}) - Q_{2'}^{\pi}(s_{t+1}, \tilde{a}_{t+1}) \right\|_{s_{t+1}, \tilde{a}_{t+1}}$$

We substitute $Q_1^\nu$ and $Q_2^\nu$, and apply the relationship $V_{\pi, \cdot}^{\nu^{\text{soft}}}(s_{t+1}) = -V_{\nu^{\text{soft}}, \cdot'}^{\pi}(s_{t+1})$ in step (1).

Or more simply, we can confirm the same result by using the result, $(\mathcal{T}_\pi^{\nu^{\text{soft}}} Q_\cdot^\nu) = \mathbb{E}_\pi \left[ -(\mathcal{T}_{\nu^{\text{soft}}}^{\pi} Q_{\cdot'}^{\pi}) \right]$, then the following relationship holds:

$$\left\| \left( \mathcal{T}_\pi^{\nu^{\text{soft}}} Q_1^\nu \right) - \left( \mathcal{T}_\pi^{\nu^{\text{soft}}} Q_2^\nu \right) \right\|_{s_t, \tilde{s}_t} = \left\| \mathbb{E}_\pi \left[ -\mathcal{T}_{\nu^{\text{soft}}}^{\pi} Q_{1'}^{\pi} \right] - \mathbb{E}_\pi \left[ -\mathcal{T}_{\nu^{\text{soft}}}^{\pi} Q_{2'}^{\pi} \right] \right\|_{s_t, \tilde{s}_t} \leq \gamma \left\| Q_{1'}^{\pi} - Q_{2'}^{\pi} \right\|_{s_{t+1}, \tilde{a}_{t+1}}. \tag{18}$$

By applying $\mathcal{T}_\pi^{\nu^{\text{soft}}}$ infinitely many times, the action-value functions $Q_{1'}^{\pi}(s, a)$ and $Q_{2'}^{\pi}(s, a)$ converge to the same fixed point at a rate governed by $\gamma \in (0.0, 1.0)$. Consequently, $Q_1^\nu(s, \tilde{s})$ and $Q_2^\nu(s, \tilde{s})$ also converge to the same fixed point under the same policy.

To prove the symmetry of the value function, we demonstrate the equivalence of the fixed points:

$$Q_\pi^{\nu^{\text{soft}}}(s, \tilde{s}) = \mathbb{E}_\pi \left[ -Q_{\nu^{\text{soft}}}^{\pi}(s, \tilde{a}) \right] \tag{19}$$

To verify this, we define the action-value function for the adversary as $Q_1^\nu(s, \tilde{s}) = \mathbb{E}_\pi \left[ -Q_{1'}^{\pi}(s, \tilde{a}) \right]$, and action-value for the agent, $Q_2^\pi(s, \tilde{a})$. We then compare the differences between them after applying each operation:

$$\left\| (\mathcal{T}_\pi^{\nu^{\text{soft}}} Q_1^\nu)(s_t, \tilde{s}_t) - \mathbb{E}_\pi \left[ -(\mathcal{T}_{\nu^{\text{soft}}}^{\pi} Q_2^\pi)(s_t, \tilde{a}_t) \right] \right\|_{s_t, \tilde{s}_t}$$

$$\underset{(1)}{=} \left\| c(s_t, \tilde{s}_t) + \gamma \mathbb{E}_{\mathcal{F} \circ \pi} \left[ \max_\nu V_{\pi,1}^{\nu^{\text{soft}}}(s_{t+1}) \right] + \mathbb{E}_\pi \left[ r(s_t, \tilde{a}_t) + \gamma \mathbb{E}_\mathcal{F} \left[ \min_\nu V_{\nu^{\text{soft}}, 2}^{\pi}(s_{t+1}) \right] \right] \right\|_{s_t, \tilde{s}_t}$$

$$= \gamma \left\| \mathbb{E}_{\mathcal{F} \circ \pi} \left[ \max_\nu V_{\pi,1}^{\nu^{\text{soft}}}(s_{t+1}) + \min_\nu V_{\nu^{\text{soft}}, 2}^{\pi}(s_{t+1}) \right] \right\|_{s_t, \tilde{s}_t}$$

$$\leq \gamma \left\| \max_\nu V_{\pi,1}^{\nu^{\text{soft}}}(s_{t+1}) + \min_\nu V_{\nu^{\text{soft}}, 2}^{\pi}(s_{t+1}) \right\|_{s_{t+1}}$$

$$\underset{Eq.(16)}{=} \gamma \left\| \max_\nu \mathbb{E}_\nu \left[ Q_1^\nu - \alpha_{attk} \mathcal{H} - \alpha_{attk} D_f \right] + \min_\nu V_{\nu^{\text{soft}}, 2}^{\pi}(s_{t+1}) \right\|_{s_{t+1}} \tag{20}$$

$$\underset{(2)}{=} \gamma \left\| \max_\nu \mathbb{E}_\nu \left[ \mathbb{E}_\pi \left[ -Q_{1'}^{\pi}(s_{t+1}, \tilde{a}_{t+1}) \right] - \alpha_{attk} \mathcal{H} - \alpha_{attk} D_f \right] + \min_\nu V_{\nu^{\text{soft}}, 2}^{\pi}(s_{t+1}) \right\|_{s_{t+1}}$$

$$\underset{Eq.(16)}{=} \gamma \left\| \min_\nu V_{\nu^{\text{soft}}, 1'}^{\pi}(s_{t+1}) - \min_\nu V_{\nu^{\text{soft}}, 2}^{\pi}(s_{t+1}) \right\|_{s_{t+1}}$$

$$= \left\| (\mathcal{T}_{\nu^{\text{soft}}}^{\pi} Q_{1'}^{\pi})(s_t, \tilde{a}_t) - (\mathcal{T}_{\nu^{\text{soft}}}^{\pi} Q_2^\pi)(s_t, \tilde{a}_t) \right\|_{s_t, \tilde{a}_t}$$

$$\underset{(3)}{\leq} \gamma \left\| Q_{1'}^{\pi}(s_{t+1}, \tilde{a}_{t+1}) - Q_2^\pi(s_{t+1}, \tilde{a}_{t+1}) \right\|_{s_{t+1}, \tilde{a}_{t+1}},$$

We use the definition of $c(s_t, \tilde{s}_t)$ in step (1), the existence condition in step (2), and the $\gamma$-contraction condition of the theorem in step (3). Thus, each stationary point must satisfy Eq. (19).

Finally, substituting Eq. (19) into Eq. (16), we obtain the result:

$$V_\pi^{\nu^{\star \text{soft}}}(s_t) = \max_{\nu \in \mathcal{N}} V_\pi^{\nu^{\text{soft}}}(s_t) = -\min_{\nu \in \mathcal{N}} V_{\nu^{\text{soft}}}^{\pi}(s_t) = -V_{\nu^{\star \text{soft}}}^{\pi}(s_t). \tag{21}$$

$\square$

As assumed in the theorem, the adversary's action-value function is expressed as the expectation of a certain action-value function with respect to the agent's policy: $Q^\nu(s, \tilde{s}) = \mathbb{E}_\pi[-Q'^\pi(s, \tilde{a})]$. While some readers might find this assumption restrictive, it is crucial to note that the desired soft optimal adversary always satisfies Eq. (19). This implies that, by deriving the soft optimal adversary's policy from the agent's action-value function, we are effectively learning within a constrained set of functions that include the desired solution, rather than searching over the entire arbitrary set of adversary action-value functions.

### B.2. VALT-SOFT: Derivation and Proofs

**Derivation of Lemma 4.7.** In this paragraph, we describe derivation of the analytical solution of Eq. (5) as in Eq. (9). Now we consider the case of the KL-divergence as follows:

$$
\begin{aligned}
\underset{\nu(\tilde{s}|s)}{\text{maximize}} \quad -V^\pi_{\nu^{\text{soft}}}(s) = \underset{\nu(\tilde{s}|s)}{\text{maximize}} \quad & \sum_{\tilde{s}\in\tilde{\mathcal{S}}} \underbrace{\nu(\tilde{s}|s)}_{\nu_{\tilde{s}}} \underbrace{\sum_{\tilde{a}\in\tilde{\mathcal{A}}} (-Q^\pi_\nu(s,\tilde{a}))}_{q_{\tilde{s}}} - \alpha_{ent} \sum_{\tilde{s}\in\tilde{\mathcal{S}}} \nu(\tilde{s}|s) \sum_{\tilde{a}\in\tilde{\mathcal{A}}} \underbrace{\pi(\tilde{a}|\tilde{s})}_{\pi_{\tilde{a},\tilde{s}}} (-\log\pi(\tilde{a}|\tilde{s})) \\
& -\alpha_{attk} \sum_{\tilde{s}\in\tilde{\mathcal{S}}} \nu(\tilde{s}|s)\left(\log\nu_{\tilde{s}} - \log\underbrace{p(\tilde{s}|s)}_{p_{\tilde{s}}}\right) \\
= \sum_{\tilde{s}\in\tilde{\mathcal{S}}} \nu_{\tilde{s}}q_{\tilde{s}} + \alpha_{ent} \sum_{\tilde{s}\in\tilde{\mathcal{S}}} \nu_{\tilde{s}} & \left(\sum_{\tilde{a}\in\tilde{\mathcal{A}}} \pi_{\tilde{a},\tilde{s}}\log\pi_{\tilde{a},\tilde{s}}\right) - \alpha_{attk} \sum_{\tilde{s}\in\tilde{\mathcal{S}}} \nu_{\tilde{s}}(\log\nu_{\tilde{s}} - \log p_{\tilde{s}}) := g(\boldsymbol{\nu}) \\
\text{subject to} \quad & \forall\tilde{s}\in\tilde{\mathcal{S}}, \quad \nu_{\tilde{s}} \geq 0, \\
& \sum_{\tilde{s}\in\tilde{\mathcal{S}}} \nu_{\tilde{s}} = 1, .
\end{aligned}
\tag{22}
$$

The Lagrangian function for this problem is given by [3]:

$$
L(\boldsymbol{\nu}, \boldsymbol{\lambda}) = g(\boldsymbol{\nu}) + \lambda(1 - \sum_{\tilde{s}\in\mathcal{S}} \nu_{\tilde{s}}),
\tag{23}
$$

where $\lambda$ is the Lagrange multiplier associated with the equality constraint. By rolling out the Karush-Kuhn-Tucker (KKT)'s stationary condition:

$$
\begin{aligned}
\nabla_{\boldsymbol{\nu}} L(\boldsymbol{\nu}, \boldsymbol{\lambda}) &= \mathbf{0} \\
\rightarrow q_{\tilde{s}} + \alpha_{ent} \underbrace{\sum_{\tilde{a}\in\tilde{\mathcal{A}}} \pi_{\tilde{a},\tilde{s}}\log\pi_{\tilde{a},\tilde{s}}}_{-\mathcal{H}_{\tilde{a},\tilde{s}}} - \alpha_{attk}(\log\nu_{\tilde{s}} - \log p_{\tilde{s}} + I_{\tilde{s}}) - \lambda &= \mathbf{0} \\
\rightarrow \log\nu_{\tilde{s}} &= \log p_{\tilde{s}} - I_{\tilde{s}} + \frac{q_{\tilde{s}} - \alpha_{ent}\mathcal{H}_{\tilde{a},\tilde{s}} - \lambda}{\alpha_{attk}} \\
\rightarrow \nu_{\tilde{s}} &= p_{\tilde{s}}\exp(\frac{q_{\tilde{s}} - \alpha_{ent}\mathcal{H}_{\tilde{a},\tilde{s}}}{\alpha_{attk}})\exp(-\lambda/\alpha_{attk} - I_{\tilde{s}}).
\end{aligned}
\tag{24}
$$

From the equality condition, we can decide $\lambda$, then:

$$
\nu_{\tilde{s}}^{\star\text{soft}} = \nu^{\star\text{soft}}(\tilde{s}\mid s) = \frac{p(\tilde{s}\mid s)\exp\left((\mathbb{E}_\pi[-Q^\pi_\nu(s,\tilde{a})] - \alpha_{ent}\mathcal{H}(\pi))/\alpha_{attk}\right)}{Z},
\tag{25}
$$

where $Z$ is the partition function. This is the result of the Lemma 4.7. $\qquad\square$

To prepare the next paragraph, give this back into the objective (Eq. (22)), then we derive:

$$
V^\pi_{\nu^{\star\text{soft}}}(s) = -\alpha_{attk}\log\int_{\tilde{s}\in\tilde{\mathcal{S}}} p(\tilde{s}\mid s)\exp\left(\frac{\mathbb{E}_\pi[-Q^\pi_\nu(s,\tilde{a})] - \alpha_{ent}\mathcal{H}(\pi)}{\alpha_{attk}}\right)d\tilde{s}.
\tag{26}
$$

---

[3]We abbreviate the inequality constraint because it can be vanished.

**Proof of Contraction Theorem 4.8.** Recall the definition of the soft worst policy evaluation in Eq. (7), substitute the results Eq. (26) in the previous paragraph,

$$(\mathcal{T}^\pi_{\nu^{\text{soft}}} Q^\pi_\nu)(s_t, \tilde{a}_t) = r(s_t, \tilde{a}_t) + \gamma \mathbb{E}_{\mathcal{F}} \left[ \min_{\nu \in \mathcal{N}} V^\pi_{\nu^{\text{soft}}}(s_{t+1}) \right],$$

$$\min_{\nu \in \mathcal{N}} V^\pi_{\nu^{\text{soft}}}(s_{t+1}) = V^\pi_{\nu^{\star\text{soft}}}(s_{t+1}) = \underbrace{-\alpha_{attk} \log \int_{\tilde{s}_{t+1} \in \tilde{\mathcal{S}}} p(\tilde{s}_{t+1} \mid s_{t+1}) \exp \left( \frac{\mathbb{E}_\pi \left[ -Q^\pi_\nu(s_{t+1}, \tilde{a}_{t+1}) \right] - \alpha_{ent} \mathcal{H}(\pi)}{\alpha_{attk}} \right) d\tilde{s}_{t+1}}_{f(Q^\pi_\nu, s_{t+1})}.$$

$$(27)$$

To represent the dependency of $V^\pi_{\nu^{\text{soft}}}$ on $Q^\pi_\nu$, we define $f(Q^\pi_\nu, s_{t+1})$ for simplifying notation.

We assume there are two different action-value functions, $Q^\pi_{\nu,1}(s_t, \tilde{a}_t)$ and $Q^\pi_{\nu,2}(s_t, \tilde{a}_t)$. We abbreviate the small character and simply show $Q_1, Q_2$ as the following.

We use the same metric as G-learning (Fox et al., 2016) and Soft Q-learning (Haarnoja et al., 2017), we define $\epsilon = \|Q_1(s_t, \tilde{a}_t) - Q_2(s_t, \tilde{a}_t)\|_{s_t, a_t}$.

Since $f(Q, s_t)$ is a monotonically increasing function for Q, then we can say:

$$\begin{aligned}
f(Q_1, s_t) &\leq f(Q_2 + \epsilon, s_t) \\
&= \epsilon + f(Q_2, s_t) \\
&= \|Q_1 - Q_2\|_{s_t, \tilde{a}_t} + f(Q_2, s_t) \\
&\leftrightarrow f(Q_1, s_t) - f(Q_2, s_t) \leq \|Q_1 - Q_2\|_{s_t, \tilde{a}_t}.
\end{aligned} \tag{28}$$

In the same way:

$$\begin{aligned}
f(Q_1, s_t) &\geq f(Q_2 - \epsilon, s_t) = f(Q_2(t)) - \epsilon \\
&\leftrightarrow f(Q_1, s_t) - f(Q_2, s_t) \geq -\|Q_1 - Q_2\|_{s_t, \tilde{a}_t}.
\end{aligned} \tag{29}$$

Therefore, from the both inequalities, we derive:

$$\|f(Q_1, s_t) - f(Q_2, s_t)\|_{s_t} \leq \|Q_1(s_t, \tilde{a}_t) - Q_2(s_t, \tilde{a}_t)\|_{s_t, \tilde{a}_t}. \tag{30}$$

Then, we consider difference between the two action-value functions after our Bellman operator:

$$\begin{aligned}
\| (\mathcal{T}^\pi_{\nu^{\text{soft}}} Q_1)(s_t, \tilde{a}_t) - (\mathcal{T}^\pi_{\nu^{\text{soft}}} Q_2)(s_t, \tilde{a}_t)\|_{s_t, \tilde{a}_t} &= \|\gamma \mathbb{E}_{\mathcal{F}} [f(Q_1, s_{t+1}) - f(Q_2, s_{t+1})]\|_{s_t, \tilde{a}_t} \\
&\leq \gamma \|f(Q_1, s_{t+1}) - f(Q_2, s_{t+1})\|_{s_{t+1}} \\
&\underset{(30)}{\leq} \gamma \|Q_1(s_{t+1}, \tilde{a}_{t+1}) - Q_2(s_{t+1}, \tilde{a}_{t+1})\|_{s_{t+1}, \tilde{a}_{t+1}}.
\end{aligned} \tag{31}$$

Therefore, we can say $\mathcal{T}^\pi_{\nu^{\text{soft}}}$ is a contraction operator, because we perform this operation an infinite number of times, $Q_1$ and $Q_2$ converge to a fixed point. $\square$

### B.3. VALT-EPS: Derivation and Proofs

**Derivation and Approximation of VALT-EPS.** If we continue to keep the f-divergence during the derivation, we can obtain a broader perspective about the relationship between choice of the divergence and the corresponding soft worst-case attack. This approach is inspired by (Belousov & Peters, 2017; 2019), which provides more detail about the derivation and related perspectives on policy improvement.

An $f$-divergence is a measurements between two distribution and defined as:

$$D_f(\nu \parallel p) \triangleq \sum_{\tilde{s} \in \mathcal{S}} p(\tilde{s}|s) f\left(\frac{\nu(\tilde{s}|s)}{p(\tilde{s}|s)}\right) \quad (s \text{ is given}). \tag{32}$$

Here, $f(\cdot) : \Omega \to \mathbb{R}$ is a convex function with the properties, $\text{Range}(f) = (0, \infty)$ and $f(x') = 0 \leftrightarrow x' = 1$. An $\alpha$-divergence is a sub-family of the $f$-divergence that is defined as:

$$f_\alpha(x) \triangleq \frac{(x^\alpha - 1) - \alpha(x - 1)}{\alpha(\alpha - 1)} \tag{33}$$

The $\alpha$-divergence includes many popular divergence: for example, the case, $\alpha \to 0$, results in Reverse-KL-divergence, $\alpha \to 1$ derive KL-divergence, and $\alpha = 2$ occurs (Pearson's) $\chi^2$-divergence. To solve dual problems, the convex conjugate function $f^*(y)$ for $f(x)$ is known to be useful. This is defined as:

$$f^*(y) = \sup_{x \in \text{dom}(f)} \{\langle y, x \rangle - f(x)\}, \tag{34}$$

where $\langle \cdot, \cdot \rangle$ denotes the inner dot product. If $f$ and $f^*$ are differentiable (Assumption 4.2), by taking the derivative of Eq. (34), $\nabla_y f^*(y) = x^\star$ and $\nabla_x \{\langle y, x \rangle - f(x)\}|_{x=x^\star} = \mathbf{0}$, we obtain the property $(f^*)' = (f')^{-1}$.

For the $\alpha$-divergence case, its conjugate function and the derivative function of the conjugate are:

$$f_\alpha^*(y) = \frac{1}{\alpha}(1 + (\alpha - 1)y)^{\frac{\alpha}{\alpha-1}} - \frac{1}{\alpha}, \tag{35}$$

$$(f_\alpha^*)'(y) = (1 + (\alpha - 1)y)^{\frac{1}{\alpha-1}}, \text{ for } (1 - \alpha)y < 1. \tag{36}$$

To consider the case described in Eq. (7), we can rewrite the problem as:

$$
\begin{aligned}
\underset{\nu(\tilde{s}|s)}{\text{maximize}} \quad -V_{\nu^{\text{soft}}}^\pi(s) &= \underset{\nu(\tilde{s}|s)}{\text{maximize}} \quad \sum_{\tilde{s} \in \tilde{\mathcal{S}}} \underbrace{\nu(\tilde{s}|s)}_{\nu_{\tilde{s}}} \underbrace{\sum_{\tilde{a} \in \tilde{\mathcal{A}}} \pi(\tilde{a}|\tilde{s})(-Q_\nu^\pi(s,\tilde{a}))}_{q_{\tilde{s}}} - \alpha_{ent} \sum_{\tilde{s} \in \tilde{\mathcal{S}}} \nu(\tilde{s}|s) \sum_{\tilde{a} \in \tilde{\mathcal{A}}} \underbrace{\pi(\tilde{a}|\tilde{s})}_{\pi_{\tilde{a},\tilde{s}}} (-\log \pi(\tilde{a}|\tilde{s})) \\
&\quad - \alpha_{attk} \sum_{\tilde{s} \in \mathcal{S}} p(\tilde{s}|s) f_\alpha \Big( \underbrace{\frac{\nu(\tilde{s}|s)}{p(\tilde{s}|s)}}_{\frac{\nu_{\tilde{s}}}{p_{\tilde{s}}}} \Big) \\
&= \sum_{\tilde{s} \in \tilde{\mathcal{S}}} \nu_{\tilde{s}} q_{\tilde{s}} + \alpha_{ent} \sum_{\tilde{s} \in \tilde{\mathcal{S}}} \nu_{\tilde{s}} \underbrace{\Big( \sum_{\tilde{a} \in \tilde{\mathcal{A}}} \pi_{\tilde{a},\tilde{s}} \log \pi_{\tilde{a},\tilde{s}} \Big)}_{\mathcal{H}(\pi)} - \alpha_{attk} \sum_{\tilde{s} \in \mathcal{S}} p_{\tilde{s}} f_\alpha \Big( \frac{\nu_{\tilde{s}}}{p_{\tilde{s}}} \Big) := g(\boldsymbol{\nu}) \\
&\text{subject to} \quad \forall \tilde{s} \in \tilde{\mathcal{S}}, \quad \nu_{\tilde{s}} \geq \mathbf{0}, \\
&\qquad\qquad\quad \sum_{\tilde{s} \in \tilde{\mathcal{S}}} \nu_{\tilde{s}} = 1, .
\end{aligned}
\tag{37}
$$

The Lagrangian for this problem is given by:

$$
L(\boldsymbol{\nu}, \boldsymbol{\lambda}, \boldsymbol{\kappa}) = g(\boldsymbol{\nu}) + \lambda \Big(1 - \sum_{\tilde{s} \in \mathcal{S}} \nu_{\tilde{s}}\Big) + \sum_{\tilde{s} \in \mathcal{S}} \kappa_{\tilde{s}} \nu_{\tilde{s}},
$$

$$
\begin{aligned}
&\text{with the KKT's condition:} \\
&\nabla_{\boldsymbol{\nu}} g(\boldsymbol{\nu}) - \lambda I_{\tilde{s}} + \kappa_{\tilde{s}} = \mathbf{0}, \\
&\nu_{\tilde{s}} \geq \mathbf{0}, \\
&\kappa_{\tilde{s}} \nu_{\tilde{s}} = \mathbf{0}, \\
&\kappa_{\tilde{s}} \geq \mathbf{0}.
\end{aligned}
\tag{38}
$$

Here, $\kappa_{\tilde{s}} := \kappa(\tilde{s}|s)$ represents the complementary slackness for the inequality constraint. From the stationarity condition:

$$
\begin{aligned}
&q_{\tilde{s}} - \alpha_{ent} \mathcal{H}(\pi) - \alpha_{attk} f_\alpha' \Big(\frac{\nu_{\tilde{s}}}{p_{\tilde{s}}}\Big) - \lambda I_{\tilde{s}} + \kappa_{\tilde{s}} = 0 \\
&\to f_\alpha' \Big(\frac{\nu_{\tilde{s}}}{p_{\tilde{s}}}\Big) = \frac{q_{\tilde{s}} - \alpha_{ent} \mathcal{H} - \lambda I_{\tilde{s}} + \kappa_{\tilde{s}}}{\alpha_{attk}} \\
&\to \nu_{\tilde{s}} = p_{\tilde{s}} (f_\alpha')^{-1} \left( \frac{q_{\tilde{s}} - \alpha_{ent} \mathcal{H} - \lambda I_{\tilde{s}} + \kappa_{\tilde{s}}}{\alpha_{attk}} \right).
\end{aligned}
\tag{39}
$$

Then, using the property, where $(f^*)' = (f')^{-1}$, we derive the optimal $\nu_{\tilde{s}}$ as:

$$\nu^\star(\tilde{s}|s) = p(\tilde{s}|s)(f_\alpha^*)' \left( \frac{\overbrace{\mathbb{E}_\pi\left[-Q_\nu^\pi(s,\tilde{a})\right] - \alpha_{ent}\mathcal{H}(\pi) - \lambda^\star I_{\tilde{s}} + \kappa(\tilde{s}|s)}^{-V^\pi(s,\tilde{s})}}{\alpha_{attk}} \right), \tag{40}$$

This result represents Lemma 4.9.

This solution can be regarded as how to asign probability mass on the prior distribution $p(\tilde{s}|s)$ and it depends on $\pi, Q_\nu^\pi, \alpha_{ent}, \alpha_{attk}$, and $\alpha$. If we assume only $\alpha < 1$ case, from Eq. (36), the term of $(f_\alpha^*)'$ must hold $> 0$ (due to its exponantial term), then $p(\tilde{s}|s) > 0 \leftrightarrow \nu^\star(\tilde{s}|s) > 0$ holds. From the complementary condition in Eq. (38), in this case, the slackness parameter $\kappa(s|\tilde{s})$ must be 0.

From the constraint in Eq. (36), we get the condition for $\lambda$:

$$\forall \tilde{s} \in \mathcal{S}, \lambda > -V^\pi(s,\tilde{s}) - \alpha_{attk}\frac{1}{1-\alpha}. \tag{41}$$

This inequality must hold for all $\tilde{s}$, thus in the case of maximum of the right term. Then, we define:

$$\lambda^\star := \max_{\tilde{s}}(-V^\pi(s,\tilde{s})) - \alpha_{attk}\frac{1}{1-\alpha} + \xi_\alpha, \tag{42}$$

where $\xi_\alpha$ is a residual that satisfies $\xi_\alpha > 0$. Considering the case if $\alpha \to -\infty$, the optimal value of constraint term approach the bound, $\lambda^\star \to \max_{\tilde{s}\in supp(p)}(-V^\pi(s,\tilde{s}))$. Therefore, $\xi_\alpha$ approaches to zero as $\alpha \to -\infty$. Substituting this back into Eq. (40) and setting $\xi(V^\pi;\tilde{s}) := \max_{\tilde{s}\in supp(p)}(-V^\pi(s,\tilde{s})) - (-V^\pi(s,\tilde{s})) \geq 0$, then we can represent as: $-V^\pi - \lambda^\star I_{\tilde{s}} = -\xi(V^\pi;\tilde{s}) + \alpha_{attk}\frac{1}{1-\alpha} - \xi_\alpha$. We can consider the cases where $\alpha \ll 0$ as follows:

$$\begin{aligned}\nu^\star(\tilde{s}|s) &= p(\tilde{s}|s)(f_\alpha^*)' \left( \frac{-\xi(V^\pi;\tilde{s}) + \alpha_{attk}\frac{1}{1-\alpha} - \xi_\alpha}{\alpha_{attk}} \right) \\ &= p(\tilde{s}|s)\left(\frac{1-\alpha}{\alpha_{attk}}(\xi_\alpha + \xi(V^\pi;\tilde{s}))\right)^{\frac{1}{\alpha-1}} \\ &= p(\tilde{s}|s)\left(\frac{\alpha_{attk}}{1-\alpha}\frac{1}{\xi_\alpha + \xi(V^\pi;\tilde{s})}\right)^{\frac{1}{1-\alpha}}. \end{aligned} \tag{43}$$

From this equation, we can determine that $\nu^\star(\tilde{s}|s)$ exhibits a strong peak at $\xi(V^\pi;\tilde{s}) = 0$, which corresponds to $\tilde{s}^\star := \arg\max_{\tilde{s}\in supp(p)}(-V^\pi(s,\tilde{s})) = \arg\min_{\tilde{s}\in supp(p)} V^\pi(s,\tilde{s})$. It also shows a mild probability mass for the other perturbation states due to the term of exponential, $0 < \frac{1}{1-\alpha} \ll 1$.

Now, we assume the prior $p(\tilde{s}|s)$ is the uniform distribution over $L_\infty$-norm constrained range. Then, we approximate the peak of the probability by a constant multiple of Dirac's delta function as $\kappa_{worst}\delta(\tilde{s}^\star)$ and distribute the remaining probability equally as $1 - \kappa_{worst}$. We represent this approximation as:

$$\nu^{\star soft}(\tilde{s}|s) \simeq \begin{cases} \kappa_{worst} + \frac{1-\kappa_{worst}}{|\tilde{\mathcal{S}}_\epsilon|}, & \text{if } \tilde{s} = \arg\min_{\tilde{s}'\in\mathcal{B}_\epsilon} V^\pi(s,\tilde{s}) \\ \frac{1-\kappa_{worst}}{|\tilde{\mathcal{S}}_\epsilon|}, & \text{others} \end{cases}. \tag{44}$$

**Proof of Contraction Theorem 4.10.** First, We confirm the soft worst-case Bellman operator in Eq. (7) for the case of Eq. (44).

$$\begin{aligned}(\mathcal{T}_{\nu^{soft}}^\pi Q)(s_t,\tilde{a}_t) &\triangleq r(s_t,\tilde{a}_t) + \gamma\mathbb{E}_\mathcal{F}\left[\mathbb{E}_{\nu^{soft}}V^\pi(s_{t+1},\tilde{s}_{t+1})\right] \\ &= r(s_t,\tilde{a}_t) + \gamma\mathbb{E}_\mathcal{F}\left[\kappa_{worst}\underbrace{V^\pi(s_{t+1},\tilde{s}_{t+1}^\star)}_{V_{worst}^\pi} + (1-\kappa_{worst})\underbrace{\mathbb{E}_{\tilde{s}_{t+1}\sim p(\cdot|s_{t+1})}\left[V^\pi(s_{t+1},\tilde{s}_{t+1})\right]}_{V_p^\pi}\right], \end{aligned} \tag{45}$$

where we set

$$\tilde{s}^{\star} = \arg\min_{\tilde{s}\in\mathcal{B}_{\epsilon}} V^{\pi}(s,\tilde{s}) = \arg\min_{\tilde{s}\in\mathcal{B}_{\epsilon}} \mathbb{E}_{\pi}[Q(s,\tilde{a})] + \alpha_{ent}\mathcal{H}(\pi). \tag{46}$$

Then, we consider difference between two action-value functions, $Q_1, Q_2$, after using the Bellman operator:

$$
\begin{aligned}
||(\mathcal{T}_{\nu^{\text{soft}}}^{\pi}Q_1)(s_t,a_t) - (\mathcal{T}_{\nu^{\text{soft}}}^{\pi}Q_2)(s_t,\tilde{a}_t)||_{s_t,\tilde{a}_t} &= ||\gamma\mathbb{E}_{\mathcal{F}}\left[\kappa V_{worst,1}^{\pi} + (1-\kappa)V_{p,1}^{\pi} - \kappa V_{worst,2}^{\pi} - (1-\kappa)V_{p,2}^{\pi}\right]||_{s_t,\tilde{a}_t} \\
&\leq \gamma||\kappa V_{worst,1}^{\pi} + (1-\kappa)V_{p,1}^{\pi} - \kappa V_{worst,2}^{\pi} - (1-\kappa)V_{p,2}^{\pi}||_{s_{t+1}} \\
&\leq \gamma\kappa||V_{worst,1}^{\pi} - V_{worst,2}^{\pi}||_{s_{t+1}} + \gamma(1-\kappa)||V_{p,1}^{\pi} - V_{p,2}^{\pi}||_{s_{t+1}} \\
&\underset{(1)}{\leq} \gamma\kappa||V_{worst,1}^{\pi} - V_{worst,2}^{\pi}||_{s_{t+1}} + \gamma(1-\kappa)||Q_1 - Q_2||_{s_{t+1},\tilde{a}_{t+1}} \\
&\underset{(2)}{\leq} \gamma\kappa||Q_1 - Q_2||_{s_{t+1},\tilde{a}_{t+1}} + \gamma(1-\kappa)||Q_1 - Q_2||_{s_{t+1},\tilde{a}_{t+1}} \\
&= \gamma||Q_1(s_{t+1},\tilde{a}_{t+1}) - Q_2(s_{t+1},\tilde{a}_{t+1})||_{s_{t+1},\tilde{a}_{t+1}}.
\end{aligned}
\tag{47}
$$

For the inequality (1), we cancel entropy terms under the same policy and perturbation, then use inequality for $\tilde{a}_{t+1}$.

For the inequality (2), similarly to the WocaR case (Liang et al., 2022), we utilized:

$$|\min_{\tilde{s}\in\mathcal{B}} A - \min_{\tilde{s}\in\mathcal{B}} B| \leq \max_{\tilde{s}\in\mathcal{B}} |A - B| \tag{48}$$

and canceling entropy terms due to the same fixed agent policy.

Therefore, we can say $\mathcal{T}_{\nu^{\text{soft}}}^{\pi}$, for the case of Eq. (12), is a contraction operator, because we perform this operation an infinite number of times, $Q_1$ and $Q_2$ converge to a fixed point. $\qquad\square$

## B.4. Proof of Policy Improvement with a Fixed Adversary

Here, we present the policy improvement theorem (Sutton & Barto, 1998) under a fixed adversary $\nu$. While we focus on the case of the maximum entropy scheme (Haarnoja et al., 2018a;b) in this study, other cases, such as deterministic policy gradient (DDPG) (Lillicrap et al., 2016; Fujimoto et al., 2018), can be similarly applied as $\pi_{\text{new}} \leftarrow \arg\max_{\pi} \mathbb{E}_{\nu}[Q_{\nu}^{\pi}(s,\pi(\tilde{s}))]$.

For simplicity, we omit the entropy coefficient $\alpha_{\text{ent}}$ in this discussion.

*Proof.*

Given a fixed adversary $\nu$ and a policy $\pi$, define a new policy $\hat{\pi}$ such that for all states $s$,

If we extract a new policy $\hat{\pi}$ by using $\pi$ and the corresponding action-value function $Q_{\nu}^{\pi}$, a following equation holds:

$$\mathbb{E}_{\tilde{s}\sim\nu}[\mathcal{H}(\pi(\cdot|\tilde{s})) + \mathbb{E}_{\tilde{a}\sim\pi}[Q_{\nu}^{\pi}(s,\tilde{a})]] \leq \mathbb{E}_{\tilde{s}\sim\nu}[\mathcal{H}(\hat{\pi}(\cdot|\tilde{s})) + \mathbb{E}_{\tilde{a}\sim\hat{\pi}}[Q_{\nu}^{\pi}(s,\tilde{a})]] \tag{49}$$

$$\leftrightarrow \mathcal{H}(\pi\circ\nu(\cdot|s)) + \mathbb{E}_{\tilde{a}\sim\pi\circ\nu}[Q_{\nu}^{\pi}(s,\tilde{a})] \leq \mathcal{H}(\hat{\pi}\circ\nu(\cdot|s)) + \mathbb{E}_{\tilde{a}\sim\hat{\pi}\circ\nu}[Q_{\nu}^{\pi}(s,\tilde{a})]. \tag{50}$$

This is proven by:

$$
\begin{aligned}
D_{KL}(\pi\circ\nu(\cdot|s) \,\|\, \hat{\pi}\circ\nu(\cdot|s)) &= \sum_{\tilde{a}\in\mathcal{A}} \pi\circ\nu(\tilde{a}|s)\left(\log\pi\circ\nu(\tilde{a}|s) - \log\hat{\pi}\circ\nu(\tilde{a}|s)\right) \\
&= -\mathcal{H}(\pi\circ\nu(\cdot|s)) - \sum_{\tilde{a}\in\mathcal{A}} \pi\circ\nu(\tilde{a}|s)\left(\log\hat{\pi}\circ\nu(\tilde{a}|s)\right) \\
&= -\mathcal{H}(\pi\circ\nu(\cdot|s)) - \sum_{\tilde{a}\in\mathcal{A}} \pi\circ\nu(\tilde{a}|s)\left(Q_{\nu}^{\pi}(s,\tilde{a}) - \log\sum_{\tilde{a}'\in\mathcal{A}}\exp\{Q_{\nu}^{\pi}(s,\tilde{a}')\}\right) \\
&= -\mathcal{H}(\pi\circ\nu(\cdot|s)) - \mathbb{E}_{\tilde{a}\sim\pi\circ\nu(\cdot|s)}[Q_{\nu}^{\pi}(s,\tilde{a})] + \log\sum_{\tilde{a}'\in\mathcal{A}}\exp\left(Q_{\nu}^{\pi}(s,\tilde{a}')\right) \\
&= -[\text{LHS of (50)}] + [\text{RHS of (50)}] \geq 0.
\end{aligned}
\tag{51}
$$

In the third equality, we used the result defined in Theorem 4.12:

$$\hat{\pi}\circ\nu(\tilde{a}|s) = \frac{\exp Q_{\nu}^{\pi}(s,\tilde{a})}{\sum_{\tilde{a}'\in\mathcal{A}}\exp Q_{\nu}^{\pi}(s,\tilde{a}')}. \tag{52}$$

By using this result, we can confirm:

$$
\begin{aligned}
Q_\nu^\pi(s_t, \tilde{a}_t) &= r(s_t, \tilde{a}_t) + \gamma \mathbb{E}_{s_{t+1}\sim\mathcal{F}}[\mathbb{E}_{\tilde{s}_{t+1}\sim\nu}[\mathbb{E}_{\tilde{a}_{t+1}\sim\pi}[Q_\nu^\pi(s_{t+1}, \tilde{a}_{t+1})] + \mathcal{H}(\pi(\cdot|\tilde{s}_{t+1}))]] \\
&\leq r(s_t, \tilde{a}_t) + \gamma \mathbb{E}_{s_{t+1}\sim\mathcal{F}}[\mathbb{E}_{\tilde{s}_{t+1}\sim\nu}[\mathbb{E}_{\tilde{a}_{t+1}\sim\hat{\pi}}\underbrace{[Q_\nu^\pi(s_{t+1}, \tilde{a}_{t+1})]}_{\text{rollout}} + \mathcal{H}(\hat{\pi}(\cdot|\tilde{s}_{t+1}))]] \\
&\vdots \\
&\leq Q_\nu^{\hat{\pi}}(s_t, \tilde{a}_t).
\end{aligned}
\tag{53}
$$

Then, under a fixed adversary $\nu$, we can improve our policy $\pi \circ \nu$ by targeting Eq. (52). $\qquad \square$

In summary, we have shown that the agent's soft worst-case action-value function can be learned. By leveraging the symmetry properties of the adversary with respect to the agent, the adversary's (soft) optimal policy can be obtained without explicitly conducting a separate RL process for the adversary. By utilizing such stabilized and fixed soft optimal policies, robust policy improvement for the agent can be achieved.

## C. Practical Implementation for VALT and SAC-PPO

---
**Algorithm 1** Framework of SAC Training with Adversary
---
1: Initialize agent critic $Q_{\theta_{1,2}}(s, a)$ and actor $\pi_\phi(s)$
2: Initialize agent target networks $Q_{\theta'_{1,2}}(s, a)$ by setting $\theta'_{1,2} \leftarrow \theta_{1,2}$
3: Initialize a replay buffer $\mathcal{R} \leftarrow \emptyset$ and entropy coefficient $\alpha_{ent} \leftarrow 1.0$
4: (1) Initialize the parameters for adversary, $\nu^{\text{soft}}$, if needed
5: **for** $t = 1$ to $T$ **do**
6:     (2) Execute action $a_t \sim \pi^{\text{behavior}}(\cdot \mid s_t)$ , observe $(s_t, a_t, r_t, s_{t+1}, done_t)$, and store in $\mathcal{R}$
7:     Sample a mini-batch of $M$ transitions $(s_t^i, a_t^i, r_t^i, s_{t+1}^i, done_t^i) \sim \mathcal{R}$
8:     (3) With $\nu^{\text{soft}}$, update the critic parameter $\theta_{1,2}$ by minimizing the Huber-Loss in Eq. (54)
9:     (4) With $\nu^{\text{soft}}$, update the actor parameter $\phi$ by maximizing the policy objective in Eq. (56)
10:     Update Entropy Coefficient $\alpha_{ent}$ by minimizing the loss:
        $L(\alpha_{ent}) = -\frac{1}{M}\sum_{i=1}^{M}\alpha_{ent}(\mathcal{H}_{target} - \mathcal{H}_{current}^i)$
        s.t.   $\mathcal{H}_{current}^i = -\log\pi_\phi(\tilde{a}_t^i|\tilde{s}_t^i)$
11:     (5) Update the parameters for adversary, $\nu^{\text{soft}}$, if needed
      Post Processing:
12:     Soft update the target networks: $\theta'_{1,2} \leftarrow (1-\tau)\theta'_{1,2} + \tau\theta_{1,2}$
13:     Update the scheduled parameters
14: **end for**
---

In this section, we provide additional details about the implementation of our VALT-EPS-SAC, VALT-SOFT-SAC, and the comparison method, SAC-PPO. All three methods incorporate the adversary during the Actor-Critic update processes. Thus, we first describe the common notation for policy evaluation, policy improvement in the following two paragraphs, and basical procedure by the pseudo-code. Subsequently, we explain the specific processes unique to each method.

**Policy Evaluation with Adversary.** As discussed in Section 4.2, we replace the original SAC policy evaluation process with the soft worst-case Bellman update. The practical procedure for policy evaluation is described as follows:

$$
s_t^i, a_t^i, r_t^i, s_{t+1}^i, done_t^i \sim \mathcal{R}, \quad i = 1, 2, \ldots, M,
$$

$$
L(\theta_{1,2}) = \frac{1}{M}\sum_{i=1}^{M} L_{\text{Hu}}\left(Q_{\theta_{1,2}}(s_t^i, a_t^i), \min\left(y_1(s_{t+1}^i), y_2(s_{t+1}^i)\right)\right),
\tag{54}
$$

$$
y_{1,2}(s_{t+1}^i) = r_t^i + \gamma(1 - done_t^i)V_{\pi,1,2}^{\nu^{\text{soft}}}(s_{t+1}^i),
$$

$$
V_{\pi,1,2}^{\nu^{\text{soft}}}(s_{t+1}^i) = \mathbb{E}_{\nu^{\text{soft}}}\left[\mathbb{E}_\pi\left[Q_{\theta'_{1,2}}(s_{t+1}^i, \tilde{a}_{t+1}^i) - \alpha_{ent}\log\pi_\phi(\tilde{a}_{t+1}^i|\tilde{s}_{t+1}^i)\right]\right],
$$

where $\sim \mathcal{R}$ indicates sampling $M$ trajectory tuples from the replay buffer $\mathcal{R}$. *done* represents a binary flag indicating the end of the episode. $L_{Hu}$ denotes the Huber loss function, $Q_{\theta_{1,2}}$ and $Q_{\theta'_{1,2}}$ represent the action-value function parameterized

by $\theta_1$ or $\theta_2$, and their target networks. $\pi_\phi$ represents the agent policy function parameterized by $\phi$. $y_j^i$ represents the function for the target value, and $V_{\pi,1,2}^{\nu^{\text{soft}}}$ denotes the value estimation for the agent under the corresponding (soft) optimal adversary $\nu^{\text{soft}}$.

Note that $\nu^{\text{soft}}$ is defined as the approximated solution to the following term:

$$\min_{\nu \in \mathcal{B}_\epsilon} \mathbb{E}_\nu \left[ \cdot \right] + \alpha_{ent} \mathcal{H}(\pi \circ \nu) + \alpha_{attk} D_f(\nu \parallel p). \tag{55}$$

**Policy Improvement with Adversary.** As discussed in Section 4.4, we replace the original SAC policy improvement with the policy improvement using a soft-worst but fixed adversary. The practical procedure are as follows:

$$s^i \sim \mathcal{R}, \quad i = 1, 2, \ldots, M,$$
$$\tilde{s}^i \sim \nu^{\text{soft}}(\cdot \mid s^i),$$
$$\tilde{a}^i \sim \pi_\phi(\cdot \mid \tilde{s}^i),$$
$$J_{\text{MDP}}(\phi) = \frac{1}{M} \sum_{i=1}^{M} \left( \min_{1,2} Q_{\theta_{1,2}}(s^i, \tilde{a}^i) - \alpha_{ent} \log \pi_\phi(\tilde{a}^i \mid \tilde{s}^i) \right), \tag{56}$$
$$J_{\text{reg}}(\phi) = -\frac{1}{M} \sum_{i=1}^{M} \left( \max_{\tilde{s}'^i \in \mathcal{B}_\epsilon} D_{KL}(\pi_\phi(\cdot \mid s^i) \parallel \pi_\phi(\cdot \mid \tilde{s}'^i)) \right),$$
$$J_{\text{total}}(\phi) = J_{\text{MDP}}(\phi) + \kappa_{\text{reg}} J_{\text{reg}}(\phi),$$

where $J_{\text{total}}(\phi)$ denotes the total objective for the agent's policy, $J_{\text{MDP}}(\phi)$ refers to the objective derived from the (SA-)MDP, $J_{\text{reg}}(\phi)$ represents the policy regularization term proposed in Zhang et al. (2020b), and $\kappa_{\text{reg}}$ is the coefficient that controls the strength of the regularization.

Note that $\nu^{\text{soft}}$ is defined as in Eq. (55).

**Procedure of SAC Training with Adversary.** We summarize Algorithm 1 to illustrate the common framework for SAC training with an adversary, as proposed. The processes that differ from the original SAC are colored as follows: Violet colors indicate modifications required for all three methods, and brown colors indicate modifications specific to adversary learning methods (VALT-SOFT and SAC-PPO).

In the following subsections, we provide details on the highlighted processes unique to each method, with a particular focus on the implementation of $\nu^{\text{soft}}$.

## C.1. Implementation Details for VALT-EPS-SAC

VALT-EPS-SAC approximates $\nu^{\text{soft}}$ using a distribution that combines uniform random samples and worst-case samples for the agent, weighted by the worst-case rate $\kappa_{worst}$. We use PGD to generate the worst-case samples, eliminating the need for additional learning parameters or corresponding procedures (brown parts in Algorithm 1).

We illustrate Algorithm 2 for this sampling process. Since SAC employs two critic networks, we use the mean of their outputs, denoted as $\overline{Q}_{\theta_{1,2}}^\pi$, for PGD. We confirm that two gradient steps with a randomized initial step achieve a good balance between computational efficiency and robustness performance. However, as this process is computationally intensive and requires GPU calculations, we use a uniform distribution at (2) in Algorithm 1 and apply (3) and (4) to incorporate the PGD outcomes.

## C.2. Implementation Details for VALT-SOFT-SAC

VALT-SOFT-SAC approximates the corresponding adversary, $\nu^{\text{soft}}$, using a parameterized policy network. Thus, we define "Initialize adversary actor $\nu_\psi^{\text{model}}(\tilde{s} \mid s)$" in (1) of Algorithm 1. By utilizing this network, we can efficiently calculate the violet parts (2), (3), and (4) in Algorithm 1 with a computationally lightweight process.

For learning the adversary in (5), we apply the policy improvement defined in Eq. (10). The detailed implementation is as follows:

---

**Algorithm 2** VALT-EPS-SAC Adversary Distribution Procedure

---

**Input:** State $s$, agent policy $\pi(a|s)$, action-value function $Q^\pi_{\theta_{1,2}}(s, a)$, worst-case rate $\kappa_{\text{worst}}$, attack scale $\epsilon$
**Output:** Epsilon-worst distribution $p^\epsilon(\tilde{s} \mid s)$
1: **Function** VALT-EPS-SAC-DISTRIBUTION$(s, \pi, Q^\pi_{\theta_{1,2}}, \kappa_{\text{worst}}, \epsilon)$
2: Compute the worst-case state using PGD (minQ attack):
3: **for** $i = 1$ to $iter\_steps$ **do**
4: $\quad \tilde{s}^\star_{new} \leftarrow \text{proj}\left(\tilde{s}^\star_{old} - \eta \nabla_{\tilde{s}^\star_{old}} \overline{Q}^\pi_{\theta_{1,2}}(s, \mu(\tilde{s}^\star_{old}))\right)$
5: **end for**
6: $p^\epsilon(\tilde{s} \mid s) \coloneqq \kappa_{\text{worst}}\delta(\tilde{s}^\star) + (1 - \kappa_{\text{worst}})\mathcal{U}(\tilde{s} \mid s - \epsilon, s + \epsilon)$
7: *(where $\eta$ is the step size, $\mu$ is the mean action output of policy $\pi$, and $\delta$ denotes the Dirac delta function)*
8: **return** $p^\epsilon(\tilde{s} \mid s)$

---

$$
\begin{aligned}
s^i &\sim \mathcal{R}, \quad i = 1, 2, \ldots, M, \\
\tilde{s}^i &\sim \nu^{\text{model}}_\psi(\cdot \mid s^i), \\
\tilde{a}^i &\sim \pi_\phi(\cdot \mid \tilde{s}^i),
\end{aligned}
\tag{57}
$$

$$
L(\psi) = \frac{1}{M} \sum_{i=1}^{M} \left( \alpha_{attk} \log \nu^{\text{model}}_\psi(\tilde{s}^i \mid s^i) + \min_{1,2} Q_{\theta_{1,2}}(s^i, \tilde{a}^i) - \alpha_{ent} \log \pi_\phi(\tilde{a}^i \mid \tilde{s}^i) \right).
$$

There is some arbitrariness in the frequency of updates and the number of iterations. We found that updating once per environmental interaction step works well, although slightly more stable results can be achieved by resetting every few environment steps and re-learning from the initial parameters.

We hypothesize that the adversary policy explores the agent's action-value function, and periodic resetting helps prevent the adversarial policy from converging to a local minimum. This sub-process for (5) in Algorithm 1 is detailed in Algorithm 3.

---

**Algorithm 3** VALT-SOFT-SAC Adversary Learning Procedure

---

**Require:** Environmental step $t$, number of learning steps per environment step: $num\_steps$
1: **if** $(t \bmod num\_steps) == 0$ **then**
2: $\quad$ Reset the adversary policy $\nu^{\text{model}}_\psi$
3: $\quad$ **for** $j = 1$ to $num\_steps$ **do**
4: $\quad\quad$ Optimize the adversary policy by minimizing the loss in Eq. (57)
5: $\quad$ **end for**
6: **else**
7: $\quad$ Do nothing (**pass**)
8: **end if**

---

### C.3. Implementation Details for SAC-PPO

This method is based on adversarial learning and resembles VALT-SOFT-SAC described in the previous subsection. At step (1), the parameters of the PPO adersary used for RL are initialized. Steps (2), (3), and (4) involve sampling from this adversarial model. However, the key difference lies in step (5), where the adversary is trained through on-policy learning using PPO with interactions from the environment.

To balance the training of the adversary and the agent, we adopted the original adversary settings as proposed in Zhang et al. (2021). Since PPO trains on a mini-batch basis, the SAC training steps for the agent were evenly divided to alternate training between the PPO adversary and the SAC agent.

For the adversarial training of PPO, please refer to Algorithm 1 in ATLA-PPO (Zhang et al., 2021), as it is implemented exactly as described in the original paper.

*Table 2.* Common SAC Hyperparameter Settings

| Hyperparameter | Value |
|---|---|
| Optimizer | Adam (Kingma & Ba, 2015) |
| Learning Rate(Actor, Critic, $\alpha_{ent}$) | 0.0003 |
| Discount Factor ($\gamma$) | 0.99 |
| Target Smoothing Coefficient ($\tau$) | 0.005 |
| Optimization Interval per Steps | 1 |
| Number of Nodes | 256 |
| Number of Hidden Layers | 2 |
| Activation Function | ReLU |
| Prioritized Experience Replay ($\beta_{per}$) | start from 0.4 to 1.0 at the end |
| Replay Buffer Size | 1,000,000 |
| Batch Size | 256 |
| Temperature Parameter ($\alpha_{ent}$) | Auto-tuning |
| Target Entropy ($\mathcal{H}_{target}$) | -dim$|\mathcal{A}|$, only for Hopper-v2: $\mathcal{H}_{target} = 0.2$ |
| Normalizer | state normalizer only |

# D. Settings and Hyperparameters

In this section, we describe the experimental settings and hyperparameters used in Section 5. The contents are divided into three subsections: training for off-policy methods, training for on-policy methods, and evaluation (attackers).

### D.1. Settings for On-Policy Methods

We use the code and settings provided by the original authors. For a fair comparison of robustness with SAC variants, we adopt the LSTM setting if it is available in the implementation.

**PPO Settings.**    We use the same settings and parameters as those provided in Zhang et al. (2021).

**Robust-PPO Settings.**    We use the settings and parameters of PPO (+LSTM) provided in Zhang et al. (2021) and tune the regularization terms. We assume that our evaluation results for Robust-PPO appear more robust than those reported in (Zhang et al., 2020b; 2021; Liang et al., 2022), likely due to the inclusion of LSTM.

**ATLA-PPO Settings.**    We use the same settings and parameters as those provided in Zhang et al. (2021).

**PA-ATLA-PPO Settings.**    We use the code provided in Sun et al. (2022) and tune the entropy coefficient and the adversary's hyperparameters as described in their paper.

**WocaR-PPO Settings.**    We use the code provided in Liang et al. (2022) and set the hyperparameters as described in their paper. We used the non-LSTM settings.

### D.2. Settings for Off-Policy Methods

We use SAC (Haarnoja et al., 2018a;b) as the base off-policy algorithm due to its good sample efficiency, high performance, and theoritical soundness. All variants, including original SAC (SAC), Robust-SAC, SAC-PPO, WocaR-SAC, and our VALT-EPS-SAC and VALT-SOFT-SAC generally adhere to the settings and parameters outlined in the next paragraph, except for their specific adjustments.

**Common SAC Settings: A Unified Basis.**    The following settings form the common foundation for all SAC variants in our experiments. While most configurations follow the second SAC paper (Haarnoja et al., 2018b), we made several adjustments to ensure stability across tasks:

- **Training steps**: 1M for HalfCheetah, 0.5M for Hopper, 1.5M for Walker2d, and 3M for Ant. For SAC-PPO, the SAC agent's training steps remain consistent, but additional interaction steps are required for the PPO adversary.

- **Modifications for stability**: Specific hyperparameter adjustments were made to improve stability (see Table 2 for details).

Table 2 lists all hyperparameters, with underlined parts highlighting the modifications. Detailed explanations for these adjustments are provided below.

- **Target Entropy for Hopper-v2**: We found that the target entropy $\mathcal{H}_{\text{ent}} = -\dim|\mathcal{A}| = -3$, as used for Hopper-v2 in the original SAC (Haarnoja et al., 2018b), is too low to get stable final results. Consequently, scores during training oscillated drastically, a phenomenon observed not only in our SAC implementation but also in a well-known open-source implementation (Raffin et al., 2021). Though a fixed temperature parameter $\alpha_{ent} = 0.2$, as in the first SAC paper (Haarnoja et al., 2018a), or decreasing learning rate for $\alpha_{ent}$ also work well, we decide to tune the target entropy only for Hopper-v2 so as to keep stable final results.

- **Prioritized Experience Replay** We use Prioritized Experience Replay (PER) (Schaul et al., 2016) to accelerate learning outcomes. While its effect is only slightly better in some tasks, we do not observe any disadvantages to using PER. Therefore, we decide to continue to use this method.

- **Normalizer** We normalize state inputs by recording running statistics as the agent receives information from environments. Subsequently, when the SAC agent samples a mini-batch for learning or calculating output from observation inputs, we use the state normalizer to set the mean and standard deviation to 0.0 and 1.0, respectively. We attempt to normalize rewards by episodic returns as PPO (Zhang et al., 2020b) and other recent on-policy methods (Zhang et al., 2021; Oikarinen et al., 2021; Sun et al., 2022; Liang et al., 2022) did. However, we observe that the agent's learning became critically slow in HalfCheetah due to delays in updating the running statistics. Therefore, we decided to abandon this normalizer.

In the following paragraphs, we describe the specific settings required for each method.

**Robust-SAC Settings.** Originally, the robust regularizer method (Zhang et al., 2020b) was implemented for PPO, A2C, DDPG, and DQN. We extended this approach to SAC, resulting in the Robust-SAC variant, by carefully integrating key elements from both PPO and DDPG implementations.

Robust-SAC incorporates action consistency terms into the policy update to address scenarios where attackers manipulate observations to induce actions that deviate from the original ones. The loss function for the policy is given as:

$$L(\pi) = \mathbb{E}_{s\sim D(\cdot)}\left[\mathbb{E}_{a\sim\pi}\left[\alpha_{ent}\log\pi(a|s) - Q^\pi(s,a)\right] + \kappa_{reg}\max_{\tilde{s}\in\mathcal{B}_{\epsilon_p}}D_{KL}\left(\pi(\cdot|s)\parallel\pi(\cdot|\tilde{s})\right)\right], \tag{58}$$

where $\kappa_{reg}$ is a coefficient balancing the original SAC loss and action consistency.

To solve the maximization term in Eq. (58), we evaluated two approaches: convex relaxation and Stochastic Gradient Langevin Dynamics (SGLD). We opted for SGLD due to its superior performance as observed in Robust-PPO (Zhang et al., 2020b).

Configurations were adapted by extracting the core ideas from DDPG and PPO variants. The attack scale increases linearly from 0 to the final target value, starting at $t = 1.0 \cdot 10^5$ and reaching the target at $t =$ training-steps $-$ remain-steps. The values for *remain-steps* were set as $2.0 \cdot 10^5$, $1.0 \cdot 10^5$, $2.0 \cdot 10^5$, and $3.0 \cdot 10^5$ for Hopper, HalfCheetah, Walker2d, and Ant, respectively. Additionally, five gradient steps were used in each calculation as in Robust-DDPG implementation (Zhang et al., 2020b).

We conducted a comprehensive search for $\kappa_{\text{reg}}$ values among $\{1, 3, 10, 30, 100, 300\}$. Except for HalfCheetah, we identified effective parameters that achieve a balance between performance in both unperturbed and adversarial settings. However, in HalfCheetah, a trade-off between robustness and performance in attack-free evaluations was observed, as reported in (Liang et al., 2022). Even in this case, stronger regularization beyond our selected settings resulted in training instability and

*Table 3.* Hyperparameters for Robust-SAC.

| Environment | Env. steps | Agent lr | Ent. target | Reg. coeff. | SGLD iter. | Other settings |
|---|---|---|---|---|---|---|
| **HalfCheetah** | 1.0M | 3e-4 | $-\dim|A|$ | 300.0 | 5 | Linear schedule: attack scale for reg. term from 0.0 to 0.15 (0.1M-0.8M) |
| **Hopper** | 0.5M | 3e-4 | 0.2 | 300.0 | 5 | Linear schedule: attack scale for reg. term from 0.0 to 0.075 (0.1M-0.4M) |
| **Walker2d** | 1.5M | 3e-4 | $-\dim|A|$ | 10.0 | 5 | Linear schedule: attack scale for reg. term 0.0 to 0.05 (0.1M-1.3M) |
| **Ant** | 3.0M | 3e-4 | $-\dim|A|$ | 30.0 | 5 | Linear schedule: attack scale for reg. term 0.0 to 0.15 (0.1M-2.7M) |

*Table 4.* Hyperparameters for SAC-PPO. The second column from the left indicates the number of environment interaction steps used for training the SAC agent. The column 'PPO t×steps' shows the number of additional environment interaction steps used for training the PPO adversary, on top of those used for the SAC agent. We applied the PA-AD technique (Sun et al., 2022) only in the Ant task.

| Environment | Env. steps | Agent lr | Ent. target | Reg. coeff. | SGLD iter. | PPO t×steps | Adv. ratio | Adv. lr | Adv. Val. lr | Adv. Ent. coeff. | Other settings |
|---|---|---|---|---|---|---|---|---|---|---|---|
| **HalfCheetah** | 1.0M | 3e-4 | $-\dim|A|$ | 30.0 | 2 | 2048 × 2441 | 0.5 | 3e-5 | 1e-5 | 1e-3 | Linear schedule: attack scale for reg. term from 0.0 to 0.15 (0.1M-0.8M) |
| **Hopper** | 0.5M | 3e-4 | 0.2 | 300.0 | 2 | 2048 × 2441 | 0.5 | 3e-5 | 3e-6 | 1e-3 | Linear schedule: attack scale for reg. term from 0.0 to 0.075 (0.1M-0.4M) |
| **Walker2d** | 1.5M | 3e-4 | $-\dim|A|$ | 10.0 | 2 | 2048 × 2441 | 0.5 | 3e-3 | 1e-3 | 3e-3 | Linear schedule: attack scale for reg. term 0.0 to 0.05 (0.1M-1.3M) |
| **Ant** | 3.0M | 3e-4 | $-\dim|A|$ | 30.0 | 2 | 2048 × 4882 | 0.5 | 1e-4 | 1e-5 | 1e-3 | Linear schedule: attack scale for reg. term 0.0 to 0.15 (0.1M-2.7M) |

degraded performance in noise-free environments. Furthermore, we found that relying solely on regularization does not improve performance against strong adversarial attacks.

Table 3 summarizes the hyperparameter settings.

**SAC-PPO Settings.** We introduce a novel off-policy alternative training framework, SAC-PPO, by incorporating the ATLA (Zhang et al., 2021) and PA-ATLA (Sun et al., 2022) implementations into our SAC framework. The SAC (agent) component retains the same settings as the base SAC, while the PPO (adversary) component follows the adversary settings of ATLA-PPO (Zhang et al., 2021). The SAC training steps (i.e., environmental interactions) are evenly divided by the PPO learning iterations to balance agent training and adversary training.

To enhance robustness, we integrate robust regularization methods (Zhang et al., 2020b), as reported in other studies (Zhang et al., 2021; Sun et al., 2022; Liang et al., 2022). Regularization settings are identical to those described in the previous paragraph, except for the use of two gradient steps to improve computational efficiency.

Therefore, the tunable parameters include those for the PPO adversary (policy learning rate, value-function learning rate, and entropy coefficient) and the coefficient of the regularizer. We perform a grid search for these parameters and adopt the best configuration.

The behavior policy for collecting training samples was set such that the adversary is always present, as defined in ATLA (Zhang et al., 2021). However, this setting caused overly pessimistic behavior, hindering learning progress (especially in Ant). On the other hand, completely excluding the adversary led to overly optimistic behavior. Therefore, we adopted a 50:50 ratio to incorporate information from both cases, ensuring stable learning and robustness.

Table 4 summarizes the hyperparameter settings.

**WocaR-SAC Settings.** WocaR (Liang et al., 2022) is originally implemented only for PPO, A2C, and DQN. We have adapted the essence of this method into our SAC implementation. In WocaR, the agent learns an additional action-value function and its target network, which estimate the worst-case scenario induced by the policy's action. This action-value function for the worst-case scenario is represented as the Bellman equation associated with the Worst-Case Bellman operator $\mathcal{T}$ as:

$$(\mathcal{T}Q)(s_t, a_t) = r(s_t, a_t) + \gamma \mathbb{E}_{s_{t+1} \sim \mathcal{F}} \left[ \min_{\tilde{s}_{t+1} \in \mathcal{B}_\epsilon} Q(s_{t+1}, \mu(\tilde{s}_{t+1})) \right]. \tag{59}$$

To calculate the next worst-case action value during updates practically, they calculate action bounds using convex relaxation (IBP) (Gowal et al., 2018) and determine the worst action using PGD within these bounds. We employ *auto_LiRPA* (Xu et al., 2020) for the convex relaxation and perform five iterations of PGD to ensure efficient computation within realistic time constraints.

During policy improvement, they enhance the gradient ascent objective of the policy by incorporating the worst-case action value, aiming to balance performance under attack-free evaluation with robustness in the worst-case attack scenario. For our

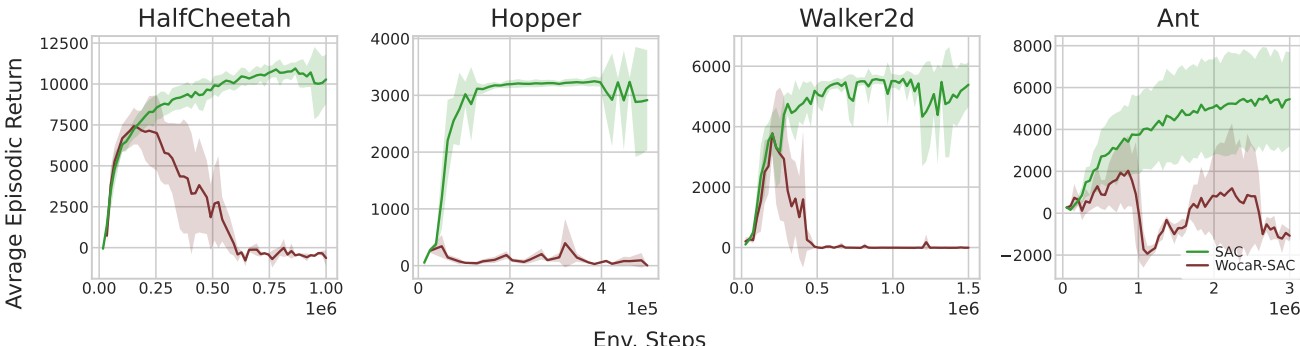

*Figure 3.* Learning curves of WocaR-SAC for four MuJoCo control tasks under no-attack settings. The solid lines represent the average evaluation scores, and the shaded areas indicate standard deviations across different seeds. While it performs well during the initial training phase, as the attack scale increases, training deteriorates across all four tasks. During training, we utilize the WocaR action-value function for deterministic gradients, set $\kappa_{\text{worst}} = 0.4$, and apply tighter relaxation methods (Zhang et al., 2020a) than IBP.

SAC implementation, we implement two methods: mixing the worst action value with the analytical target value (soft-max of the Q) as:

$$L(\pi) = \mathbb{E}_{s \sim D(\cdot)} \left[ \mathbb{E}_{a \sim \pi} \left[ \alpha_{ent} \log \pi(a|s) - \left( (1 - \kappa_{worst}) Q^\pi(s,a) + \kappa_{worst} \underline{Q}^\pi(s,a) \right) \right] \right], \tag{60}$$

combining DDPG's policy loss with the worst action-value function by using the mean of the action, $\mu(s)$, as:

$$L(\pi_\phi) = \mathbb{E}_{s \sim D(\cdot)} \left[ (1 - \kappa_{worst}) \mathbb{E}_{a \sim \pi} \left[ \alpha_{ent} \log \pi(a|s) - Q^\pi(s,a) \right] - \kappa_{worst} \underline{Q}^\pi(s, \mu(s)) \right]. \tag{61}$$

To mitigate poor exploration and pessimistic learning behaviors, they schedule both the attack scale and the mixture rate of the worst-case objective throughout the policy optimization process. We initiate the attack scale at 0.0 and linearly increase it to the final scale by $t = \text{training-steps} - \text{remain-steps}$, maintaining this value until the end of the training. The values for *remain-steps* were set as $2.0 \cdot 10^5$, $1.0 \cdot 10^5$, $2.0 \cdot 10^5$, and $3.0 \cdot 10^5$ for HalfCheetah, Hopper, Walker2d, and Ant, respectively. Similarly, we schedule $\kappa_{worst}$ to increase from 0.0 to the final target value, testing among $\{0.2, 0.4, 0.6\}$.

Both methods in Eq. (60) and Eq. (61) learn well in smaller tasks (Pendulum, InvertedPendulum) and at the initial stages of the tasks. However, as the attack scale increases, the performance drastically declines, leading to the collapse of training, even when we use more tighter bound method (Zhang et al., 2020a) than IBP used in WocaR (See Fig. 3). We hypothesize that this is because the policy fails to learn from the worst-case scenarios due to the absence of adversaries during policy improvement. Consequently, the policy cannot produce valid actions for the next target action value, causing the worst action value to estimate increasingly poorer values as the attack scale enlarges.

A similar phenomenon is observed in the ablation experiment of VALT-EPS-SAC in Ant-v2, which lacks an adversary during policy improvement.

Therefore, our findings indicate that WocaR is an effective technique for on-policy methods, as it utilizes a gradient ascent target combining normal PPO's value and WocaR's action-value. However, a different approach is required to achieve policy robustness in off-policy settings.

Considering the opposite case, if we were to adapt the VALT technique to on-policy methods, we would face the following challenge: since the VALT technique can only estimate off-policy (soft) worst-case value estimations, we would need to either abandon multi-step rollouts for the agent and perform optimization in a single step, or combine PPO's multi-step estimations with VALT's (soft) worst-case value estimation. In the latter case, its implementation may resemble WocaR-PPO. Thus, WocaR's worst-case ratio $\kappa_{\text{worst}}$ can be interpreted as analogous to VALT-EPS's coefficient, which represents the strength of the soft-constrained term.

**VALT-EPS-SAC Settings.** VALT-EPS-SAC combines uniform perturbations with an approximation of the worst attack using the mixture coefficient $\kappa_{\text{worst}}$. We employ PGD to approximate the worst attack, similarly to the approach used in Pattanaik et al. (2018); Zhang et al. (2020b). Since estimating the worst-case states that consider the entropy term of SAC is complicated and has concerns of unstable learning, we employ two approximations: (1) using the mean of the action

*Table 5.* Hyperparameters for VALT-EPS-SAC. Our method requires fewer hyperparameters than ATLA, as we do not explicitly train an adversary policy or value function. Additional parameters, such as the adversarial ratio and $\kappa_{worst}$ scheduling, are less sensitive to tuning.

| Environment | Env. steps | Agent lr | Ent. target | Reg. coeff. | SGLD iter. | Adv. ratio | Other settings |
|---|---|---|---|---|---|---|---|
| **HalfCheetah** | 1.0M | 3e-4 | $-\dim\lvert A\rvert$ | 30.0 | 2 | 0.5 | Linear schedule: attack scale for reg. term from 0.0 to 0.15 (0.1M-0.8M), $\kappa_{worst}$: from 0.0 to 1.0 (0M-0.8M) |
| **Hopper** | 0.5M | 3e-4 | 0.2 | 3.0 | 2 | 0.5 | Linear schedule: attack scale for reg. term from 0.0 to 0.075 (0.1M-0.4M), $\kappa_{worst}$: from 0.0 to 1.0 (0M-0.4M) |
| **Walker2d** | 1.5M | 3e-4 | $-\dim\lvert A\rvert$ | - | - | 0.5 | ~~Linear schedule: attack scale for reg. term 0.0 to 0.05 (0.1M-1.3M)~~, $\kappa_{worst}$: from 0.0 to 1.0 (0M-1.3M) |
| **Ant** | 3.0M | 3e-4 | $-\dim\lvert A\rvert$ | 30.0 | 2 | 1.0 | Linear schedule: attack scale for reg. term 0.0 to 0.15 (0.1M-2.7M), $\kappa_{worst}$: from 0.0 to 1.0 (0M-2.7M) |

outputs, and (2) ignoring the entropy term. As the SAC-based algorithm's Critic comprises two networks, we average their outputs for gradient calculations.

To balance computational resources and performance, we determine that two gradient steps with a random start during PGD suffice for our methods. To mitigate poor exploration during the initial training phase, we start $\kappa_{\text{worst}} = 0$ and linearly increase it to 1.0. by $t = \text{training-steps} - \text{remain-steps}$, maintaining this value until the end of the training. The values for *remain-steps* were set as $2.0 \cdot 10^5$, $1.0 \cdot 10^5$, $2.0 \cdot 10^5$, and $3.0 \cdot 10^5$ for HalfCheetah, Hopper, Walker2d, and Ant, respectively.

Regarding the behavior policy for collecting training samples, we found that calculating PGD at every step for each action with the environment is computationally too heavy. Instead, we utilize a uniform distribution in place of the epsilon-worst distribution. A 50:50 ratio of incorporating information from perturbed and non-perturbed settings works well, while for high-dimensional tasks like Ant, fully perturbed settings enhance robustness. We hypothesize that the uniform perturbation setting is not overly pessimistic and increases the likelihood of encountering state-action pairs that are challenging for the agent.

To enhance robustness, we apply the robust regularization technique in the same manner as described for SAC-PPO.

Table 5 summarizes the hyperparameter settings.

**VALT-SOFT-SAC Settings.** VALT-SOFT-SAC uses an inference model to represent the (soft) optimal adversary policy. The network architecture and learning rate are identical to the policy network in SAC, but the output layer is modified to match the dimensionality of the state space to generate noise.

The update frequency for the adversary policy is somewhat arbitrary. Although updating the adversary policy at every step yielded good results, resetting and retraining it every 1,000-2,000 steps slightly improved training stability. Thus, we chose periodic resets and retraining, with resets every 2,000 steps for HalfCheetah, Hopper, and Walker2d, and every 1,000 steps for Ant.

The tuning parameter $\alpha_{attk}$, which determines the strength of the adversary's constraint, was set to constant values of 8.0, 2.0, 4.0, and 4.0 for HalfCheetah, Hopper, Walker2d, and Ant, respectively. Although these values may appear large, they act as an entropy term to prevent the model from converging to local minima. Additionally, unlike in the PPO setting, rewards were not normalized. Given the typical scale of discounted cumulative rewards (approximately 300-500), these values are reasonable.

For the behavior policy, we used 100% adversary influence for HalfCheetah and a 50:50 ratio of adversary and non-adversary policies for the other tasks. To avoid overly pessimistic training while acquiring challenging data, we observed that linearly adjusting the proportion during training, particularly for Ant, led to stronger robustness. However, as this introduces task-specific complexity in experience collection, further exploration of behavior policies and experience memory is left for future work.

To enhance robustness, we applied the robust regularization technique in the same manner as described for SAC-PPO and VALT-EPS-SAC.

Table 6 summarizes the hyperparameter settings.

### D.3. Evaluation Settings

Our trainings and evaluations require much computational resources (CPUs/GPUs) therefore, we extract median seed from from eight different training seeds, then evaluate 20 episodic returns.

*Table 6.* Hyperparameters for VALT-SOFT-SAC. Compared to VALT-EPS-SAC, VALT-SOFT-SAC introduces more hyperparameters due to adversary policy training. However, the adversary learning rate (Adv. lr) was fixed across tasks without tuning, and others (e.g., $\alpha_{attk}$, reset interval) showed low sensitivity.

| Environment | Env. steps | Agent lr | Ent. target | Reg. coeff. | SGLD iter. | Adv. lr | Adv. ratio | Reset Adv. per Steps | $\alpha_{attk}$ | Other settings |
|---|---|---|---|---|---|---|---|---|---|---|
| HalfCheetah | 1.0M | 3e-4 | $-\dim\|A\|$ | 1.0 | 2 | 3e-4 | 1.0 | 2000 | 8.0 | Linear schedule: attack scale for reg. term from 0.0 to 0.15 (0.1M-0.8M) |
| Hopper | 0.5M | 3e-4 | 0.2 | 3.0 | 2 | 3e-4 | 0.5 | 2000 | 2.0 | Linear schedule: attack scale for reg. term from 0.0 to 0.075 (0.1M-0.4M) |
| Walker2d | 1.5M | 3e-4 | $-\dim\|A\|$ | - | - | 3e-4 | 0.5 | 2000 | 4.0 | ~~Linear schedule: attack scale for reg. term from 0.0 to 0.05 (0.1M-1.3M)~~ |
| Ant | 3.0M | 3e-4 | $-\dim\|A\|$ | 30.0 | 2 | 3e-4 | 0.5 | 1000 | 4.0 | Linear schedule: attack scale for reg. term from 0.0 to 0.15 (0.1M-2.7M) |

During evaluation and adversarial training (RS/SA-RL/PA-AD), we employ the same state normalizer learned by the agent to scale attacker strength, Although this approach is somewhat akin to white-box attack settings, we ensure that these settings are treated consistently, thus guaranteeing that the results are evaluated in a uniform manner.

**Max Action Deiffference (MAD) Attack.**  We follow as the original implementation (Zhang et al., 2020b). We utilize SGLD and confirm that ten gradient iterations after random initial start are enough for the evaluations.

**PGD (minV/minQ) Attack.**  We follow the original implementation (Zhang et al., 2020b), rather than the method originally proposed in Pattanaik et al. (2018).

For the PPO variants, we use the implementation provided in (Zhang et al., 2021). For the SAC variants, we implement them ourselves by extracting the essence of the PPO implementation. As mentioned in Appendix D.2, the SAC critic comprises two networks, and we use the mean of these two action-value estimates to compute the gradient descent.

We utilize Projected Gradient Descent (PGD) and confirm that ten gradient iterations with a random initial start are sufficient for the evaluations.

**RobustSarsa (RS) Attack.**  We apply the implementation provided in (Zhang et al., 2021) for both PPO and SAC variants. We perform multiple training runs to tune robust critic parameters (perturbation scale and regression term) around the benchmark's attack scale. Subsequently, we conduct multiple evaluations using all trained models and adopt the worst evaluation results. For fair comparisons, we use the same number of critic trainings and evaluations across all methods in a uniform manner.

**SA-RL and PA-AD Attacks Implemented with PPO and SAC.**  For PPO (SA-RL/PA-AD) attackers, we apply the implementations provided in (Zhang et al., 2021; Sun et al., 2022). We perform approximately seventy training runs with different seeds and parameters to identify the strongest attacker for the agent. Subsequently, we conduct evaluations with these attackers and adopt the worst evaluation scores.

For SAC (SA-RL/PA-AD) attackers, we use the same hyperparameters and settings as detailed in Appendix D.2, with two exceptions: flipping the reward term $r \rightarrow -r$ and setting the target entropy as $\mathcal{H}_{\text{target}} = -\dim|\mathcal{S}|$ for SA-RL. To mitigate variance, we perform four training runs with different seeds for both SA-RL and PA-AD. Subsequently, we conduct evaluations with these attackers and adopt the worst evaluation scores.

For fair comparisons, we ensure the same number of attacker training runs and evaluations across all methods in a consistent manner.

# E. Additional Experiments

## E.1. Vertifying without robust regularizer term

To examine the robustness characteristics of SAC-based methods and the tendencies of our proposed approaches, we conducted evaluations with and without regularization. The results are summarized in Table 7.

The colored settings indicate the more robust configurations, which were adopted in our main results. Specifically, yellow highlights settings where regularization was applied, while lime denotes settings without regularization.

Our method demonstrated strong robustness against adversarial attacks (SA-RL/PA-AD) in HalfCheetah, even without regularization. However, in the Ant task, VALT-SOFT-SAC suffered a significant performance drop even in evaluations

*Table 7.* Comparison of performance with and without regularization terms. Average episodic rewards (± standard deviation) for median-seed models of our proposed methods (VALT-EPS, VALT-SOFT) and other SAC baselines across four MuJoCo tasks. All evaluations were conducted over twenty episodes with different seeds. Yellow indicates a setting where regularization improves robustness, while lime denotes a non-regularized setting. The most robust method in the off-policy category is highlighted in bold.

| Environment | Method | Env. Steps | Natural Reward | Uniform | MAD | PGD (minV) | PGD (minQ) | RS | SA-RL (SAC) | PA-AD (SAC) | SA-RL (PPO) | PA-AD (PPO) | Worst score across attacks |
|---|---|---|---|---|---|---|---|---|---|---|---|---|---|
| **HalfCheetah** state-dim: 17 action-dim: 6 $\epsilon = 0.15$ | SAC | 1M | 10985±117 | 6411±273 | 3288±154 | - | 3424±171 | 4215±247 | -51±267 | 1757±409 | 936±465 | 1171±214 | -51±267 |
| | Robust-SAC | 1M | 6350±68 | 6127±84 | 5837±112 | - | 2601±311 | 3183±375 | 2026±417 | 3481±383 | 3197±532 | 3367±468 | 2026±417 |
| | SAC-PPO | 6M | 6385±1930 | 5533±870 | 3778±475 | - | 1716±117 | 3140±310 | 397±68 | 1479±188 | 1578±149 | 1832±581 | 397±68 |
| | SAC-PPO +reg. | 6M | 7487±147 | 7154±141 | 6030±1084 | - | 1823±871 | 5836±138 | 3206±112 | 2052±835 | 2849±726 | 3008±303 | 1823±871 |
| | **VALT-EPS-SAC** | 1M | 7313±967 | 7345±156 | 6183±887 | - | 2690±137 | 6652±161 | 2416±27 | 4181±300 | 3027±264 | 4442±219 | 2416±27 |
| | **VALT-EPS-SAC** +reg. | 1M | 6027±112 | 5900±84 | 5850±98 | - | 4647±180 | 5834±107 | 5216±159 | 5396±113 | 5707±100 | 5695±114 | **4647±180** |
| | **VALT-SOFT-SAC** | 1M | 6100±76 | 5769±78 | 4867±310 | - | 4533±80 | 5067±66 | 3329±88 | 4385±81 | 4796±135 | 4697±82 | 3329±88 |
| | **VALT-SOFT-SAC** +reg. | 1M | 5881±45 | 5798±55 | 5448±93 | - | 4480±83 | 5434±80 | 3584±61 | 4449±69 | 4722±83 | 4868±87 | 3584±61 |
| **Hopper** state-dim: 11 action-dim: 3 $\epsilon = 0.075$ | SAC | 0.5M | 3199±1 | 3213±6 | 3194±391 | - | 3281±12 | 3337±5 | 2979±1 | 3166±11 | 3034±1 | 3070±2 | 2979±1 |
| | Robust-SAC | 0.5M | 3237±5 | 3234±8 | 3237±9 | - | 2947±6 | 3290±2 | 2990±10 | 3114±26 | 2952±4 | 3015±8 | 2947±6 |
| | SAC-PPO | 5.5M | 3359±12 | 3368±4 | 3479±17 | - | 3528±12 | 3527±6 | 3203±1 | 3286±12 | 3198±2 | 3236±6 | 3198±2 |
| | SAC-PPO +reg. | 5.5M | 3176±2 | 3174±6 | 3185±10 | - | 3036±19 | 3104±2 | 3178±4 | 3244±19 | 3157±2 | 3140±4 | 3036±19 |
| | **VALT-EPS-SAC** | 0.5M | 3200±2 | 3195±4 | 3196±6 | - | 2507±779 | 2675±812 | 3095±13 | 3109±23 | 2974±2 | 1391±112 | 1391±112 |
| | **VALT-EPS-SAC** +reg. | 0.5M | 3297±2 | 3297±3 | 3293±7 | - | 3264±17 | 3358±2 | 3267±2 | 3255±9 | 3212±4 | 3264±6 | **3212±4** |
| | **VALT-SOFT-SAC** | 0.5M | 3310±3 | 3319±6 | 3407±14 | - | 3478±5 | 3465±4 | 3244±1 | 3219±14 | 650±118 | 3300±3 | 650±118 |
| | **VALT-SOFT-SAC** +reg. | 0.5M | 3228±2 | 3230±10 | 3241±15 | - | 2986±3 | 3462±3 | 3034±3 | 3102±9 | 3059±7 | 3031±4 | 2986±3 |
| **Walker2d** state-dim: 17 action-dim: 6 $\epsilon = 0.05$ | SAC | 1.5M | 5866±47 | 5601±1086 | 4288±2280 | - | 3550±2427 | 2732±2710 | 2143±2452 | 337±567 | 1911±2525 | 3252±2773 | 337±567 |
| | Robust-SAC | 1.5M | 5840±21 | 5824±41 | 5784±63 | - | 5180±808 | 5766±20 | 5432±842 | 5552±26 | 5645±50 | 5644±67 | **5180±808** |
| | SAC-PPO | 6.5M | 3493±1473 | 4004±1423 | 4365±1501 | - | 2423±1536 | 3266±1145 | 1812±153 | 1343±178 | 1609±528 | 1695±1666 | 1343±178 |
| | SAC-PPO +reg. | 6.5M | 5182±36 | 5009±847 | 5005±889 | - | 3420±1237 | 4232±1640 | 821±11 | 2287±795 | 1830±1398 | 2936±2090 | 821±11 |
| | **VALT-EPS-SAC** | 1.5M | 5901±56 | 5904±50 | 5587±1158 | - | 4777±1932 | 5486±1146 | 5820±93 | 3442±2428 | 4883±2066 | 3993±2371 | 3442±2428 |
| | **VALT-EPS-SAC** +reg. | 1.5M | 5757±58 | 5740±128 | 4979±1783 | - | 5791±97 | 4972±1834 | 3134±2458 | 323±99 | 2824±2568 | 3597±2452 | 323±99 |
| | **VALT-SOFT-SAC** | 1.5M | 5604±24 | 5609±27 | 5579±55 | - | 5327±925 | 5490±71 | 5249±18 | 4981±1567 | 5485±19 | 5468±20 | 4981±1567 |
| | **VALT-SOFT-SAC** +reg. | 1.5M | 5655±16 | 5658±17 | 5679±28 | - | 2838±2035 | 5451±1129 | 5372±887 | 5568±223 | 5525±38 | 4487±2035 | 2838±2035 |
| **Ant** state-dim: 111 action-dim: 8 $\epsilon = 0.15$ | SAC | 3M | 6583±1852 | 996±1076 | -27±74 | - | -23±53 | -106±370 | -1953±1213 | -1047±700 | -725±322 | -7±102 | -1953±1213 |
| | Robust-SAC | 3M | 6790±1548 | 6117±2226 | 4847±2578 | - | 1687±1125 | 937±899 | 3171±2529 | 68±1016 | 1045±1135 | 3351±2085 | 68±1016 |
| | PA-SAC-PPO | 13M | 7093±1297 | 5854±1932 | 336±302 | - | 184±272 | 60±94 | 1431±1738 | -144±237 | 336±351 | 837±866 | -144±237 |
| | PA-SAC-PPO +reg. | 13M | 6351±1730 | 6308±1518 | 5680±1404 | - | 1778±693 | 2760±1798 | 4169±2144 | -354±468 | 3754±691 | 4077±2110 | -354±468 |
| | **VALT-EPS-SAC** | 3M | 6251±1802 | 4643±2557 | 3026±2283 | - | 1601±1188 | 1252±1419 | 3351±2208 | 4481±2327 | 741±874 | 1425±1374 | 741±874 |
| | **VALT-EPS-SAC** +reg. | 3M | 6332±748 | 5129±2397 | 5019±1923 | - | 1487±1176 | 3356±2408 | 4710±2319 | 4944±1891 | 1738±1466 | 4585±2022 | 1487±1176 |
| | **VALT-SOFT-SAC** | 3M | 3293±1703 | 2458±1524 | 873±588 | - | 1077±731 | 1009±1010 | 392±411 | 74±71 | 413±441 | 520±320 | 74±71 |
| | **VALT-SOFT-SAC** +reg. | 3M | 6719±66 | 6552±260 | 5287±2035 | - | 2757±1297 | 3088±1521 | 3808±1718 | 5152±1653 | 4281±167 | 3684±2590 | **2757±1297** |

without noise.

At first glance, this result may seem unusual. However, since our method generates adversarial outputs in the state space, Ant, with its 111-dimensional state space, presents a particularly large and complex setting. We propose two hypotheses: first, without a certain degree of robust adjustment (in this case, regularization), the agent cannot effectively counter the adversary; second, since the adversary policy is trained through value estimation via the agent's policy, successful adversary training requires the agent's policy to be sufficiently smooth.

As noted in the main section, the impact of regularization is relatively small for tasks with smaller noise scales, such as Hopper and Walker. Eysenbach & Levine (2022) noted that SAC inherently possesses robustness properties, allowing it to handle relatively small perturbations without explicit regularization, and to perform comparably to the standard baseline in such settings. However, larger perturbations require deeper insights into the MDP structure, as addressed by methods like ATLA and our proposed VALT. In environments with high-dimensional input spaces, regularization becomes essential for stabilizing adversary learning. Analyzing the behavior of non-max-ent DRL methods, such as TD3 (Fujimoto et al., 2018), in this context is also an interesting direction, which we leave for future work.

### E.2. Ablation Study for VALT-EPS and VALT-SOFT

**Effects of Policy Evaluation and Policy Improvement with Soft-Worst Adversary.** We conducted an investigation into the importance of Policy Evaluation (PE) and Policy Improvement (PI) among the update methods we introduced in Section 4, which assume an adversary during training. This analysis focused on the Ant task, where particularly distinctive trends were observed for VALT-EPS-SAC and VALT-SOFT-SAC. The training curves are shown in Fig. 4, and the evaluation results are summarized in Table 8.

In the ablation setting where the adversary is not assumed during policy evaluation (w/o PE), there is little change in evaluation scores. However, in the ablation setting where the adversary is not assumed during policy improvement (w/o PI), training collapses in both methods, resulting in catastrophic performance even in noise-free evaluations.

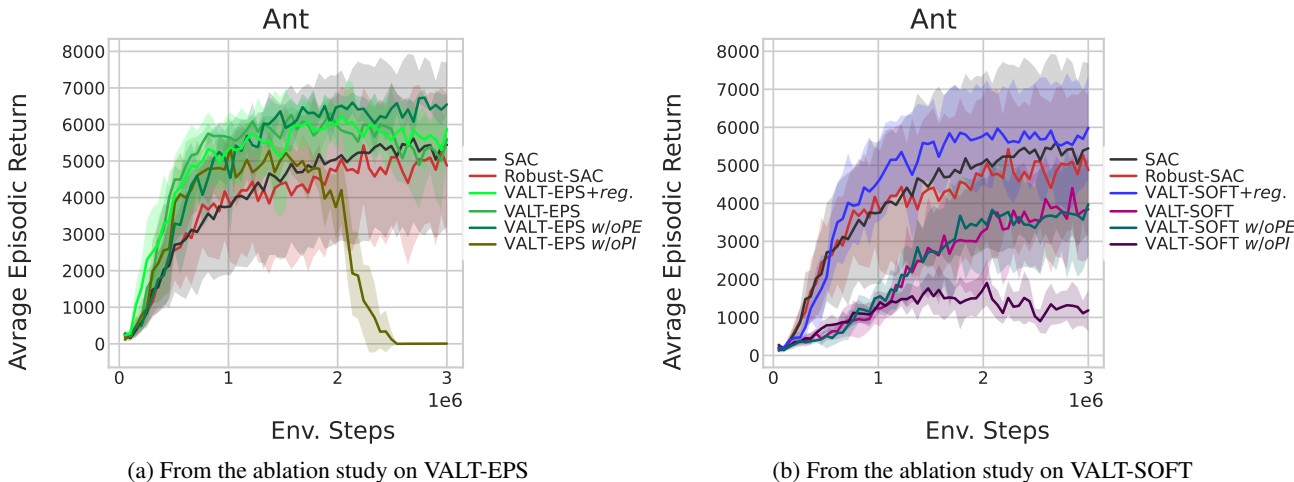

(a) From the ablation study on VALT-EPS      (b) From the ablation study on VALT-SOFT

*Figure 4.* Learning curves from the ablation study on VALT-EPS and VALT-SOFT for the Ant task under no-attack settings. We ablate the policy regularization term (*+reg.*), the adversary during policy evaluation (*w/o PE*), and the adversary during policy improvement (*w/o PI*). For both methods, *w/o PI* is crucial for training stability.

*Table 8.* Comparison of performance with and without the soft worst-case adversary during Policy Evaluation (PE) and Policy Improvement (PI). We denote these settings as *w/o PE* and *w/o PI*. Average episodic rewards (± standard deviation) for median-seed models. All evaluations were conducted over twenty episodes with different seeds. Yellow indicates the setting used in the main results (with regularization).

| Environment | Method | Env. Steps | Natural Reward | Uniform | MAD | PGD (minV) | PGD (minQ) | RS | SA-RL (SAC) | PA-AD (SAC) | SA-RL (PPO) | PA-AD (PPO) | Worst score across attacks |
|---|---|---|---|---|---|---|---|---|---|---|---|---|---|
| | SAC | 3M | 6583±1852 | 996±1076 | -27±74 | - | -23±53 | -106±370 | -1953±1213 | -1047±700 | -725±322 | -7±102 | -1953±1213 |
| Ant state-dim: 111 action-dim: 8 $\epsilon = 0.15$ | **VALT-EPS-SAC** *+reg.* | 3M | 6332±748 | 5129±2397 | 5019±1923 | - | 1487±1176 | 3356±2408 | 4710±2319 | 4944±1891 | 1738±1466 | 4585±2022 | 1487±1176 |
| | **VALT-EPS-SAC** | 3M | 6251±1802 | 4643±2557 | 3026±2283 | - | 1601±1188 | 1252±1419 | 3351±2208 | 4481±2327 | 741±874 | 1425±1374 | 741±874 |
| | **VALT-EPS-SAC** *w/o PE* | 3M | 6318±1681 | 6418±1283 | 2779±2218 | - | 1144±1044 | 3048±2155 | 2269±1890 | 3249±2137 | 1128±1102 | 2952±2029 | 1128±1102 |
| | **VALT-EPS-SAC** *w/o PI* | 3M | 3±3 | 2±2 | -9±8 | - | -2±16 | -8±4 | 2±3 | -6±4 | -10±5 | -3±3 | -10±5 |
| | **VALT-SOFT-SAC** *+reg.* | 3M | 6719±66 | 6552±260 | 5287±2035 | - | 2757±1297 | 3088±1521 | 3808±1718 | 5152±1653 | 4281±167 | 3684±2590 | 2757±1297 |
| | **VALT-SOFT-SAC** | 3M | 3293±1703 | 2458±1524 | 873±588 | - | 1077±731 | 1009±1010 | 392±411 | 74±71 | 413±441 | 520±320 | 74±71 |
| | **VALT-SOFT-SAC** *w/o PE* | 3M | 4917±1495 | 3316±2081 | 1855±1201 | - | 1420±880 | 424±440 | 3472±1882 | 157±110 | 205±176 | 864±666 | 157±110 |
| | **VALT-SOFT-SAC** *w/o PI* | 3M | 993±773 | 554±601 | 78±190 | - | 12±52 | 380±388 | 344±457 | -704±84 | -795±614 | 342±374 | -795±614 |

We hypothesize that this occurs because the adversary learns to generate difficult noise through the agent's value function, whereas the agent's policy does not adapt to such noise during training. As a result, the value estimation may gradually become excessively pessimistic.

This trend resembles the learning collapse observed in WocaR-SAC, as discussed in Section D.2 and illustrated in Fig. 3, further suggesting that assuming the presence of an adversary during policy improvement is crucial for off-policy methods.

Some readers may wonder why the *w/o PE* settings perform better than the standard settings (especially in the case of VALT-SOFT-SAC), as the Natural Reward and Worst Score are not worse—and in some cases even better—than those of the standard settings without regularization. To investigate this phenomenon, we conducted an additional evaluation described in the next paragraph.

**Additional Evaluation of VALT-EPS and VALT-SOFT with Regularization under the *w/o PE* Condition.** To confirm the robustness performance under the *w/o PE* condition, we conducted additional evaluations of VALT-EPS-SAC and VALT-SOFT-SAC with regularization. The results are summarized in Table 9.

Although the *w/o PE* condition shows better natural performance than the unregularized settings, its worst-case robustness is lower than that of the standard settings (i.e., with policy evaluation adversaries). Interestingly, the strongest adversary against the standard settings is PGD (minQ), while for the *w/o PE* settings, it is SA-RL PPO.

These results suggest that omitting adversaries during policy evaluation stabilizes training and leads to better performance under clean conditions. However, the agent's action-value function lacks sufficient pessimism, which limits its ability to capture worst-case robustness.

Our VALT framework is designed to learn robust policies by embedding a soft worst-case estimation directly into a unified action-value function. Consequently, this value function not only guides policy learning under adversarial perturbations, but also serves as a robustness-aware evaluation metric under white-box settings. This dual role highlights that the learned value function functions both as a target for adversarial exploitation and as a reliable indicator of worst-case performance.

*Table 9.* Additional comparison of performance with and without the soft worst-case adversary during policy evaluation (PE). We denote the ablation setting as *w/o PE*, and regularization is applied in all settings, including the ablation. Values indicate average episodic rewards ($\pm$ standard deviation) for models trained with the median performance seed. All evaluations were conducted over twenty episodes using different evaluation seeds. Yellow highlights the setting used in the main results (i.e., without the *w/o PE* ablation).

| Environment | Method | Env. Steps | Natural Reward | Uniform | MAD | PGD (minV) | PGD (minQ) | RS | SA-RL (SAC) | PA-AD (SAC) | SA-RL (PPO) | PA-AD (PPO) | Worst score across attacks |
|---|---|---|---|---|---|---|---|---|---|---|---|---|---|
| **Ant** state-dim: 111 action-dim: 8 $\epsilon = 0.15$ | SAC | 3M | 6583±1852 | 996±1076 | -27±74 | - | -23±53 | -106±370 | -1953±1213 | -1047±700 | -725±322 | -7±102 | -1953±1213 |
| | **VALT-EPS-SAC** *+reg.* | 3M | 6332±748 | 5129±2397 | 5019±1923 | - | 1487±1176 | 3356±2408 | 4710±2319 | 4944±1891 | 1738±1466 | 4585±2022 | 1487±1176 |
| | **VALT-EPS-SAC** *w/o PE +reg.* | 3M | 6204±1417 | 6327±1345 | 6296±143 | - | 3462±1537 | 1538±1649 | 3466±2700 | 4724±2208 | 1155±1129 | 2332±1929 | 1155±1129 |
| | **VALT-SOFT-SAC** *+reg.* | 3M | 6719±66 | 6552±260 | 5287±2035 | - | 2757±1297 | 3088±1521 | 3808±1718 | 5152±1653 | 4281±167 | 3684±2590 | **2757±1297** |
| | **VALT-SOFT-SAC** *w/o PE +reg.* | 3M | 6590±1018 | 6309±1387 | 5215±1977 | - | 1902±1226 | 1719±1603 | 4156±2183 | 3030±2236 | 1398±1518 | 3125±2595 | 1398±1518 |

## E.3. Considerations for Behavior Policy

*Table 10.* Comparison of performance under different adversary rates in the behavior policy. We denote these settings as **Adv\***, where * indicates the adversary rate. Values represent average episodic rewards ($\pm$ standard deviation) for models trained with the median seed. All evaluations were conducted over twenty episodes using different evaluation seeds. Yellow highlights the setting used in the main results (with a 100% adversary rate and regularization).

| Environment | Method | Env. Steps | Natural Reward | Uniform | MAD | PGD (minV) | PGD (minQ) | RS | SA-RL (SAC) | PA-AD (SAC) | SA-RL (PPO) | PA-AD (PPO) | Worst score across attacks |
|---|---|---|---|---|---|---|---|---|---|---|---|---|---|
| **HalfCheetah** state-dim: 17 action-dim: 6 $\epsilon = 0.15$ | SAC | 1M | 10985±117 | 6411±273 | 3288±154 | - | 3424±171 | 4215±247 | -51±267 | 1757±409 | 936±465 | 1171±214 | -51±267 |
| | **VALT-SOFT Adv1.0** *+reg.* | 1M | 5881±45 | 5798±55 | 5448±93 | - | 4480±83 | 5434±80 | 3584±61 | 4449±69 | 4722±83 | 4868±87 | 3584±61 |
| | **VALT-SOFT Adv1.0** | 1M | 6100±76 | 5769±78 | 4867±310 | - | 4533±80 | 5067±66 | 3329±88 | 4385±81 | 4796±135 | 4697±82 | 3329±88 |
| | **VALT-SOFT Adv0.5** | 1M | 7782±99 | 6736±923 | 4552±276 | - | 4496±178 | 4905±1036 | 5644±1011 | 3439±192 | 2660±197 | 3829±212 | 2660±197 |
| | **VALT-SOFT Adv0.0** | 1M | 9083±80 | 6502±1545 | 3728±333 | - | 2471±203 | 2261±269 | 1290±103 | 1706±150 | 3332±87 | 3224±450 | 1290±103 |

**HalfCheetah Case**   We evaluated the impact of different adversary assumptions in the behavior policy, as shown in the training curves in Fig. 5 and summarized in Table 10 for robustness evaluation. The **Adv1.0+***reg.* and **Adv1.0** settings exhibit similar training curves, likely due to the small regularization coefficient.

In the Adv0.0 setting, performance is high in a noise-free environment, but it significantly deteriorates under strong adversarial attacks. This may occur because, if the replay buffer contains only favorable samples, policy evaluation assumes an adversary only for one-step predictions from stable states, and policy improvement further enhances actions from these favorable states. As a result, HalfCheetah achieves high scores in the noise-free setting, but at the cost of reduced robustness.

Since HalfCheetah is less prone to training collapse, we adopted a 100% adversary influence. However, future research should explore adaptive balancing mechanisms to mitigate task dependency and correct distributional shifts in the replay buffer.

*Table 11.* Comparison of performance under different adversary rates in the behavior policy. We denote these settings as **Adv\***, where * indicates the adversary rate. Values represent average episodic rewards ($\pm$ standard deviation) for models trained with the median seed. All evaluations were conducted over twenty episodes using different evaluation seeds. Yellow highlights the setting used in the main results (with a 100% adversary rate and regularization).

| Environment | Method | Env. Steps | Natural Reward | Uniform | MAD | PGD (minV) | PGD (minQ) | RS | SA-RL (SAC) | PA-AD (SAC) | SA-RL (PPO) | PA-AD (PPO) | Worst score across attacks |
|---|---|---|---|---|---|---|---|---|---|---|---|---|---|
| **Ant** state-dim: 111 action-dim: 8 $\epsilon = 0.15$ | SAC | 1M | 6583±1852 | 996±1076 | -27±74 | - | -23±53 | -106±370 | -1953±1213 | -1047±700 | -725±322 | -7±102 | -1953±1213 |
| | **VALT-SOFT Adv0.0** *+reg.($\alpha_{attk} = 4$ const.)* | 3M | 6062±1450 | 3808±2290 | 402±722 | - | 63±171 | 217±167 | 135±110 | 55±38 | 40±108 | 111±56 | 40±108 |
| | **VALT-SOFT Adv0.5** *+reg.($\alpha_{attk} = 4$ const.)* | 3M | 6719±66 | 6552±260 | 5287±2035 | - | 2757±1297 | 3088±1521 | 3808±1718 | 5152±1653 | 4281±167 | 3684±2590 | **2757±1297** |
| | **VALT-SOFT Adv1.0** *+reg.($\alpha_{attk} = 4$ const.)* | 3M | 6069±408 | 5955±532 | 5436±1054 | - | 2908±1374 | 4723±1056 | 4651±1814 | 5121±495 | 3307±2066 | 4433±2099 | **2908±1374** |
| | **VALT-SOFT Adv1.0** *+reg.($\alpha_{attk} = 1000 \rightarrow 4$)* | 3M | 6420±91 | 5839±1478 | 5445±1475 | - | 3598±990 | 4833±1720 | 4802±2086 | 5301±1033 | 3022±2289 | 4193±1879 | **3022±2289** |

**Ant Case**   We further evaluated the impact of different adversarial influence ratios during agent behavior on the Ant task. In addition to fixed adversarial strengths, we introduced a configuration that anneals the adversary temperature parameter $\alpha_{attk}$ from a high value (1000.0) to a low value (4.0), aiming to gradually increase the adversarial impact over the course of training, while keeping the adversarial influence ratio constant (**Adv** = 1.0).

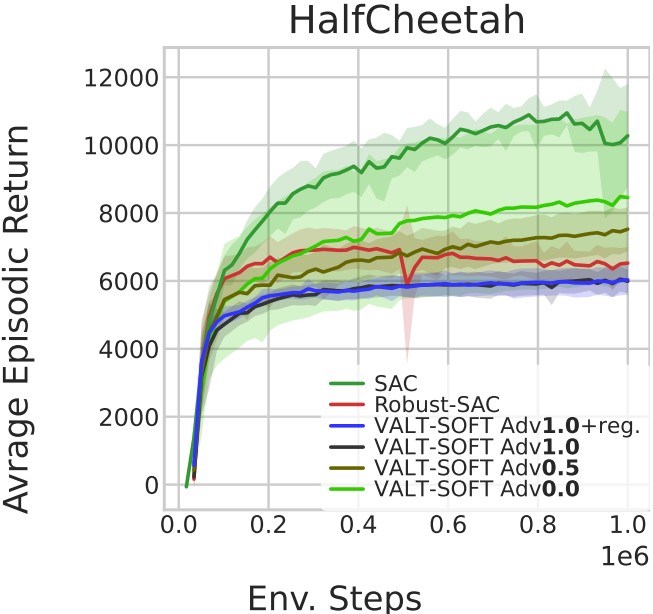

*Figure 5.* Impact of different adversary assumptions in the behavior policy. Training curves for various adversary influence rates in VALT-SOFT-SAC on the HalfCheetah task under no-attack settings. **Adv1.0** represents a setting where the behavior policy is fully influenced by the adversary (i.e., $\tilde{a}_t \sim \pi \circ \nu^{\text{soft}}(\cdot \mid s_t)$), while **Adv0.0** assumes no adversary influence (i.e., $a_t \sim \pi(\cdot \mid s_t)$).

Figure 6 shows the clean evaluation curves averaged over eight different seeds. Most configurations achieved successful learning; however, the setting with full adversarial influence during behavior (**Adv** $= 1.0$) and fixed temperature ($\alpha_{\text{attk}} = 4.0$) showed large variance across seeds and an increased chance of converging to suboptimal policies (local minima), occurring in 3 out of 8 seeds compared to the typical 2 out of 8. This degradation likely stems from the fact that, under this setting, training trajectories are collected assuming strong adversaries even in the early stages—before the agent has learned sufficiently good behaviors—thus increasing the chance of getting stuck in poor local solutions.

To further assess robustness, we report adversarial evaluation scores in Table 11. Similar to the HalfCheetah case, models trained without adversarial consideration in the behavior policy suffered severe performance drops under attack—even when regularization was applied. This suggests that in high-dimensional input spaces, failing to account for adversarial perturbations during data collection leads to insufficient coverage of pessimistic scenarios, degrading robustness.

Interestingly, the setting with full adversarial influence (**Adv** $= 1.0$) and regularization produced the robust policies in terms of median performance, and further improvements were observed when the adversary temperature, $\alpha_{attk}$, was annealed from 1000 to 4. This schedule consistently resulted in stable learning across all seeds and yielded robust behavior even under worst-case evaluations.

These findings imply that introducing adversarial signals gradually—i.e., allowing the agent to collect sufficient data for reasonable behavior early on and emphasizing stronger perturbations later—can improve both optimization and data collection strategies. Although further fine-grained scheduling of adversarial influence or temperature might enhance learning efficiency and robustness, off-policy methods inherently face the challenge of policy-data mismatch. Therefore, we focused on systematic experiments with straightforward and consistent configurations in this study.

## F. Possible Framework for Discrete Action Domains in the Case of VALT

In this work, we focused on continuous action domains, especially in the case of SAC. Though we believe that our proposed methods can be extended to discrete action environments, here we discuss the possible framework for discrete action environments in the case of VALT and the possible challenges for the implementation. For the simplicity, we discuss the DQN base framework (deterministic policy), but the same idea can be applied to the other cases like SAC-Discrete (Christodoulou, 2019).

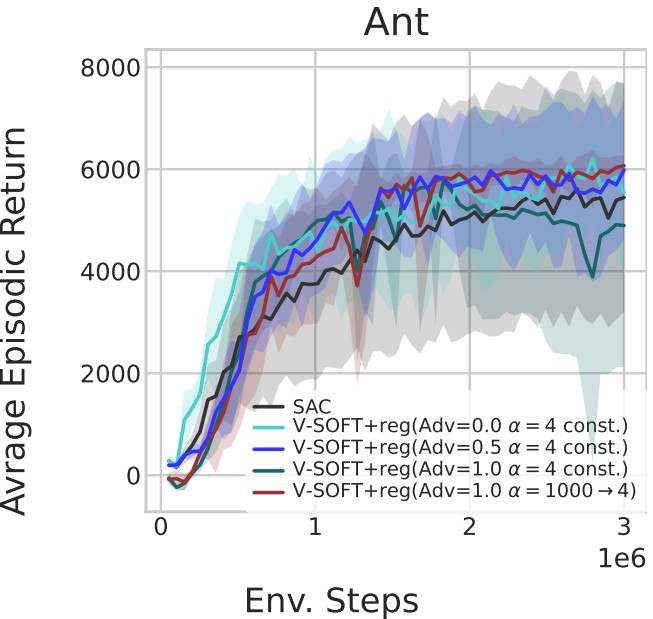

*Figure 6.* Training performance of VALT-SOFT-SAC on the Ant task under no-attack evaluation, comparing different behavior policy assumptions and $\alpha_{\text{attk}}$ schedules. **Adv1.0** assumes full adversarial influence ($\tilde{a}_t \sim \pi \circ \nu^{\text{soft}}$), while **Adv0.0** uses the agent policy alone ($a_t \sim \pi$). $\alpha = 4$ **const.** denotes a fixed attack temperature, and $\alpha = 1000 \to 4$ indicates a scheduled decay from 1000 to 4 over **0–2.7M steps**. The main result uses **Adv0.5** with $\alpha_{\text{attk}} = 4$ (blue line).

To consider VALT framework, we assume the agent action as:

$$\tilde{a}_{\text{agent}} = \pi(\tilde{s}) = \text{argmax}_{\tilde{a} \in \mathcal{A}} Q_{\text{agent}}(\tilde{s}, \tilde{a}), \tag{62}$$

where $Q_{\text{agent}}(\tilde{s}, \tilde{a})$ is the Q-function for the agent. Then, we can define the (soft) optimal adversary for the agent as:

$$\nu^{\star\text{soft}}(\tilde{s} \mid s) = \text{argmin}_{\nu \in \mathcal{N}} \mathbb{E}_{\nu} \left[ \mathbb{E}_{\text{agent}} \left[ Q_{\text{agent}}(s, \tilde{a}) \right] \right] + \alpha_{attk} D_f(\nu \parallel p). \tag{63}$$

Lastly, the action-value function for the agent is updated under the fixed adversary, Eq. (63), by minimizing the following loss:

$$Loss(Q_{\text{agent}}) = \text{Loss\_func}(Q_{\text{agent}}(s, a), y(s, a, s')) \tag{64}$$
$$y(s, a, s') = r(s, a) + \gamma \mathbb{E}_{\nu^{\star\text{soft}}} \left[ \mathbb{E}_{\pi} \left[ Q'_{\text{agent}}(s', \tilde{a}') \right] \right]$$

where $Q'_{\text{agent}}$ is the target Q-function for the agent, Loss\_func represents the loss function, e.g., L2-mean and Huber loss, under the distribution, $s, a, s' \sim D(\cdot)$, and $y(s, a, s')$ represents the one-step target value for the agent.

The main challenge in practical implementation is the development of an algorithm to compute the following expectation term in Eq. (64):

$$\mathbb{E}_{\nu^{\star\text{soft}}} \left[ \mathbb{E}_{\pi} \left[ Q_{\text{agent}}(s', \tilde{a}') \right] \right], \tag{65}$$

which is handled differently in the VALT-SOFT and VALT-EPS cases, as described below.

**VALT-SOFT for Discrete Actions** In the VALT-SOFT case, we can derive the soft worst-case adversary distribution (as in VALT-SOFT-SAC, Eq. (9)) as:

$$\nu^{\star\text{soft}}(\tilde{s} \mid s) = \frac{p(\tilde{s} \mid s) \exp\left(-V_{\text{agent}}(s, \tilde{s})/\alpha_{\text{attk}}\right)}{Z}, \tag{66}$$

where $Z$ is the partition function, and $V_{\text{agent}}(s, \tilde{s}) = \mathbb{E}_{\pi} \left[ Q_{\text{agent}}(s, \tilde{a}) \right]$ is the agent's value function. The inner expectation $\mathbb{E}_{\pi}$ can be computed as in Eq. (62).

Similar to VALT-SOFT-SAC, this soft worst-case adversary distribution can be approximated using a surrogate model such as a neural network. However, in high-dimensional environments like Atari (e.g., 80×80 inputs), training the adversary model becomes challenging due to the curse of dimensionality in the adversarial action space.

Although we have not yet experimented with this, we assume that regularization plays an important role in stabilizing adversary training, as observed in the ablation study of VALT-SOFT-SAC on Ant (with state dimension 111). Additional techniques, such as PA-AD (Sun et al., 2022), may also be required.

**VALT-EPS for Discrete Actions**  In the VALT-EPS case, we can expand Eq. (65) into:

$$\mathbb{E}_{\nu^{\star \text{soft}}}\left[\mathbb{E}_\pi\left[Q_{\text{agent}}(s', \tilde{a}')\right]\right] = (1.0 - \kappa_{\text{worst}})\mathbb{E}_p\left[\mathbb{E}_\pi\left[Q_{\text{agent}}(s', \tilde{a}')\right]\right] + \kappa_{\text{worst}}\min_{\tilde{s}' \in \mathcal{B}_\epsilon}\mathbb{E}_\pi\left[Q_{\text{agent}}(s', \tilde{a}')\right]. \tag{67}$$

Here, the first term—the expected value under the prior distribution (uniform in this study)—can be easily computed, while the second term—the worst-case value within the $\epsilon$-ball—requires approximation techniques to compute the minimum value, since $Q_{\text{agent}}$ is represented by a neural network.

In WocaR-DQN, Liang et al. (2022) proposed two methods to estimate the worst-case value: (1) deriving the available action set via convex relaxation bounds (upper/lower) of the $\epsilon$-ball, from which the worst-case action is selected; or (2) using the Projected Gradient Descent (PGD) method to approximate the worst-case perturbation that degrades the agent's best action.

If we adopt these techniques, VALT-EPS for DQN becomes similar to WocaR-DQN, although some differences remain. WocaR-DQN introduces an additional Q-function to model the worst-case, whereas in our approach, the worst-case is unified into the agent's Q-function as a soft worst-case while preserving the contraction property.

In the case of VALT-EPS, high-dimensional state spaces may not pose a significant challenge, as the worst-case values are computed through the Q-function using convex relaxation or gradient descent techniques.

**Actor-Critic Cases**  Although the above discussion assumes a value-iteration-style method (e.g., DQN), where Eq. (64) can be interpreted as a form of value iteration, a policy-iteration-style framework (e.g., Actor-Critic) may offer advantages in computing Eq. (65). Instead of requiring an additional procedure to estimate the $\arg\max$ in Eq. (62), we can directly compute the expectation term $\mathbb{E}_\pi[\cdot]$ that appears in Eqs. (65), (66), and (67), under a fixed policy assumption. Then, both the adversary's optimization in VALT-SOFT (Eq. (66)) and the worst-case value estimation in VALT-EPS (Eq. (67)) can be approximated as demonstrated in this study: the former directly utilizes the fixed policy for the expectation, while the latter is approximated by applying PGD either to the mean action of a deterministic policy or to the value prediction under the softmax policy (as in SAC-Discrete).

**Summary**  In summary, we have presented a possible framework for VALT in discrete action domains. While the proposed approaches show promise, further progress is expected in integrating advances in areas such as the vulnerability of DNN in high-dimensional input spaces, computational efficiency, and the practical application to modern discrete-action domains.

## G. Computational Resources

**Wall-clock Time to Train**  To clarify the time and resources required for our algorithms and other baselines, we present Table 12, which summarizes the total training time for each algorithm. For a fair comparison, all training runs were conducted on *Tesla V100 32GB* GPUs.

Notably, our implementation can also run on a CPU with as little as 1GB of memory (3GB for Ant), albeit at approximately 1.5 times the training duration compared to GPU-based execution. However, we observed that VALT-EPS-SAC fails to proceed with learning on CPUs, potentially due to the computational overhead of PGD.

PPO variants (e.g., PPO, ATLA-PPO) have been reported in prior work (Liang et al., 2022), and we confirm that our results are consistent with theirs, as we use the same publicly available codebase.

**Analysis of Computation Time**  To analyze the computational efficiency of our methods and guide future improvements, we provide a breakdown of computation time in Table 13, based on training various SAC variants on the HalfCheetah task. Here, cur_q, next_q, q_back, and q_opt respectively represent the computation of current Q-values during policy evaluation, value estimation for the next state, backpropagation for action-value function optimization, and parameter

*Table 12.* Computation times for each method

| Model | HalfCheetah | Hopper | Walker2d | Ant |
|---|---|---|---|---|
| | Time (hour) | | | |
| SAC | 4.2 | 2.1 | 6.2 | 12.8 |
| Robust-SAC | 7.7 | 3.7 | 11.2 | 23.4 |
| SAC-PPO | 13.5 | 10.3 | 17.8 | 47.9 |
| VALT-EPS-SAC | 9.7 | 4.8 | 14.1 | 30.1 |
| VALT-SOFT-SAC | 10.1 | 5.0 | 14.8 | 31.0 |

*Table 13.* Computation time (seconds) per 10k steps on HalfCheetah.

| Model | act | Policy Evaluation | | | | Policy Improvement | | | | Other Components | |
|---|---|---|---|---|---|---|---|---|---|---|---|
| | | cur_q | next_q | q_back | q_opt | p_fwd | reg | p_back | p_opt | $\alpha$_learn | adv_learn |
| SAC | 18 | 6 | 21 | 17 | 22 | 24 | – | 23 | 13 | 7 | – |
| Robust-SAC | 19 | 6 | 22 | 17 | 22 | 24 | 125 | 32 | 13 | 7 | – |
| SAC-PPO | 25 | 6 | 27 | 18 | 27 | 31 | 71 | 35 | 16 | 9 | 229 |
| VALT-EPS-SAC | 20 | 6 | 150 | 18 | 24 | 156 | 69 | 53 | 14 | 8 | – |
| VALT-SOFT-SAC | 35 | 6 | 38 | 19 | 24 | 43 | 70 | 43 | 14 | 8 | 90 |

updates for the action-value network. Similarly, p_fwd, reg, p_back, and p_opt correspond to action computation during policy improvement, regularization term computation, backpropagation of the policy loss, and policy parameter updates. alpha_learn and adv_learn denote the time spent on temperature parameter adjustment and adversary training, respectively.

For SAC-PPO, we normalize the computation time based on the number of learning steps equivalent to SAC's 10k steps, since it consists of PPO adversary training, 2048 steps over 2441 iterations in total.

We omit simulation time in MuJoCo, replay buffer operations (e.g., storing and sampling), and post-processing such as logging, as these components did not significantly vary across methods and were not major contributors to overall computation.

Note that the reported values reflect peak computation time, and are subject to fluctuation due to scheduling (e.g., of regularization terms) and variations in overall server load.

Among the most notable factors:

- **SAC-PPO:** As the forward passes of the adversary (PPO) are computationally light, the agent-side optimization is also relatively inexpensive. However, the large number of adversary updates required by the alternating training scheme results in a significant overall time cost. While reducing the number of adversary updates is a possible approach, achieving a balance between SAC's training efficiency and proper tuning of the adversary proves to be highly challenging and likely impractical.

- **VALT-EPS:** This method incurs substantial overhead during both policy evaluation and improvement due to the requirement of PGD computations at each iteration. As a result, next_q and p_fwd show markedly high values. Since reducing the number of PGD iterations (currently two) would severely compromise robustness, further efficiency gains would require parameter tuning or alternative methods.

- **VALT-SOFT:** The adversary is trained at fixed intervals (every 1000–2000 steps) using trajectories sampled from the replay buffer. We observed that robustness performance was relatively insensitive to the choice of interval size. While reducing the number of update steps per interval could further improve efficiency, adversary training was not a major computational bottleneck and thus remained unchanged.

- **Other Observations:** Methods incorporating adversarial signals into the behavior policy (e.g., SAC-PPO, VALT-SOFT) show minor additional costs proportional to computational time. Robust-SAC's regularization takes 70–125 seconds due to 5 SGLD iterations (following SA-DDPG (Zhang et al., 2020b)), while other methods use only 2. For better efficiency, Robust-SAC could also adopt fewer iterations.

**Summary**    As discussed in the main results and computational analysis, our framework achieves observation robustness through a formulation naturally integrated into the MDP—effectively addressing a challenge for existing off-policy methods. Despite its effectiveness, our approach entails additional computational overhead, as described in this section. These aspects leave room for further improvement and serve as a promising direction for future research.

One possible direction for broader adoption by general off-policy actor-critic users— while keeping additional cost minimal— might be to introduce assumed environment noise during behavior policy execution, policy evaluation, and policy updates. This idea is analogous to how TD3 adds noise to actions during policy evaluation.

## H. Reproducing Statement

We provide detailed procedures, experimental settings, and hyperparameters in Appendix C and Appendix D. In addition, we release our code—including implementations of SAC variants and evaluation settings—for reproducibility and further research.

## I. Limitations

As discussed in Section 4, our proposed methods are grounded in a solid theoretical foundation and can be applied across a variety of scenarios. However, there are two primary limitations to be noted.

First, although we outline a possible extension of the framework to discrete action domains in Appendix F, a detailed algorithm and implementation are not yet provided.

Second, the proposed methods are less compatible with on-policy algorithms, as the VALT framework fundamentally relies on off-policy value estimation, which is not well aligned with the core principles of on-policy learning. (See the discussion on "WocaR-SAC Settings" in Appendix D.2 for further details.)

