# OpenReview forum: "Off-Policy Actor-Critic for Adversarial Observation Robustness: Virtual Alternative Training via Symmetric Policy Evaluation"
_ICML.cc/2025/Conference — ICML 2025 poster_

### Official Review · Reviewer_Vgnr · 2025-03-10

**Overall Recommendation:** 3

**Summary:**

The paper presents a novel off-policy reinforcement learning approach that addresses adversarial input observations without requiring additional environmental interactions, thus enhancing sample efficiency and avoiding inefficiencies in agent-environment interactions. By reformulating adversarial learning as a soft-constrained optimization problem, the method eliminates mutual dependencies between the agent and adversary. The approach is theoretically supported by the symmetric property of policy evaluation and shows consistent success and strong sample efficiency in evaluations, making it a promising contribution to the field.

**Claims And Evidence:**

The claims of this work is supported by both theories and empirical results.

**Essential References Not Discussed:**

I think the authors cover most of the references.

**Experimental Designs Or Analyses:**

Please refer to "Questions for authors"

**Methods And Evaluation Criteria:**

Yes, I think the proposed method and evaluation are reasonable. However, this paper also mentions multiple of [1], this baseline should be included.





#### [1] Reddi, A., T¨ olle, M., Peters, J., Chalvatzaki, G., and D’Eramo, C. Robust adversarial reinforcement learning via bounded rationality curricula. ICLR 2024

**Other Comments Or Suggestions:**

Please avoid citing paper in arXiv version if it was accepted. For example, the paper below.





#### Reddi, A., T¨ olle, M., Peters, J., Chalvatzaki, G., and D’Eramo, C. Robust adversarial reinforcement learning via bounded rationality curricula. ICLR 2024

**Other Strengths And Weaknesses:**

Strengths
* well-written
* comprehensive evaluation on different attacks
* proposed method is impactful and concise
* algorithms are supported with theories

Weaknesses
* In 5.2. Attacker Settings, it is mentioned that the common attack scales are used in previous studies. Please cite those works.
* Although, this works briefly shows how to choose the mix in line 283, I still have the concern about the over-optimism/pessimism, which is pointed out from a lots of robust optimization works, e.g., [2]. Further defense or mentioning explicit limitations can enhance this work.

#### [2] Juncheng Dong et al. Variational Adversarial Training Towards Policies with Improved Robustness. AISTATS 2025

**Questions For Authors:**

* How can we tell the most robust sample-efficient method from the values shown in table 1?
* I thought that different heuristic attacks is only for evaluation. Then what is the reason that the the most robust sample-efficient method under different attacks are different training methods?
* Can you elaborate more why both SAC and VALT-EPS has higher variance only at the end for hopper task in figure 2?
* Could you discuss when people should use between VALT-EPS and VALT-SOFT from both theoretical and empirical perspectives?
* Table 6 records the computation time among different off-policy methods. In the experiments, SAC-PPO took more training steps from previous tables. Then how will be a fair comparison regarding computation time in table 6?
* I have the concern if there increases a lot more hyper-parameters for the two variants of the proposed methods compared with existing robust RL methods.
* It seems that each proposed method has been evaluated under different attacks. While during training, it requires hyper-parameter tuning for each method, including baselines, how do you decide what kind of hyper-parameters should be adopted and when the training can be stopped? In other words, there may happen that the learning curves perform better for hyper-parameter A while it may be more robust to some attacks in evaluation for hyper-parameter B, and maybe be more robust to the other attacks in evaluation for hyper-parameter C.

**Relation To Broader Scientific Literature:**

* This work opens an alternative way to implicitly learn adversarial training without two-player game. It will be better for contributing to community with the open-source code.

**Theoretical Claims:**

I went through the theories and proofs. I did not go into details of the proofs in the appendix, but overall it sounds reasonable.

---

> ### Author Rebuttal · Authors · 2025-03-31
>
> We thank the reviewer for the constructive and detailed feedback. Below, we respond to each point.
>
> ---
> ## Methods and Evaluation Criteria
> ### About Baseline [1]
> > [1] Reddi et al., *Robust adversarial reinforcement learning via bounded rationality curricula*, ICLR 2024
>
> Thank you for the suggestion.
> We considered adapting [1]—originally proposed for dynamics robustness—to the observation robustness setting. However, we found key theoretical mismatches. [1] trains the adversary using a separate replay buffer under stationary dynamics. Under observation perturbations, the effective dynamics become $\mathcal{F} \circ \pi(s' \mid s, \tilde{s})$, leading to inconsistency between the adversary's buffer and the trajectory induced by the current policy, thereby violating convergence assumptions.
>
> This reveals a structural gap: [1] assumes dynamics that receive two actions in parallel, whereas our setting involves a cascade structure (perturb, then act). Bridging this gap is an interesting direction, but it is beyond the scope of this work.
>
> ---
>
> ## Relation to Broader Literature
>
> > **Code Release**
>
> Thank you. Given the multiple baselines and training procedures, additional time is needed for cleanup and documentation. We are committed to releasing the code with the camera-ready version.
>
> ---
>
> ## Weaknesses
>
> - **[W1] Missing Citations in Section 5.2:**
>
> Thank you. We will revise the manuscript to include relevant citations.
>
> - **[W2] Mixture Coefficient (line 283):**
>
> Our heuristic choice was made for simplicity, but we agree it may cause miscalibration. We will add a discussion and cite works on distributionally robust optimization.
>
> ---
>
> ## Other Comments
>
> - **arXiv vs. Published Citations:**
>
> We will update arXiv references to their published versions where applicable.
>
> ---
>
> ## Questions for Authors
>
> - **[Q1] Identifying Robust and Sample-Efficient Methods**
>
> In Table 1, the highest scores within each on-policy or off-policy category are highlighted in bold, and the best-performing methods among the most sample-efficient are shaded in gray. Our methods consistently maintain strong robustness, especially in complex tasks (HalfCheetah, Ant), emphasizing the value of MDP-aware modeling. Following Reviewer AMCX's suggestion, we will use worst-case metrics to better capture robustness.
>
> - **[Q2] Why Robust Methods Vary Across Attacks**
>
> We attribute this to equilibrium differences across training methods. Robust RL does not guarantee global optimality, and policies may settle into different robustness profiles depending on the training dynamics.
>
> - **[Q3] Variance in Hopper**
>
> This is addressed in our response to Reviewer wL61.
>
> - **[Q4] When to Use VALT-EPS vs. VALT-SOFT**
>
> Two key factors:
>
> 1. **Computation**: VALT-EPS uses PGD-based attacks across two networks, requiring GPU acceleration (line 1689).
> 2. **Task Sensitivity**: In failure-sensitive environments, VALT-SOFT better explores around risky observations. In tolerant domains like HalfCheetah, VALT-EPS is more effective.
>
> - **[Q5] Computation Time in Table 6**
>
> Table 6 is intended as reference, not comparison. We agree that step count affects time. In the revision, we’ll include per-update time and breakdowns to improve fairness.
>
> - **[Q6] Hyperparameter Complexity**
>
> Our method introduces fewer tunable parameters than ATLA. For example, we avoid learning adversary networks. Most hyperparameters follow SAC; new ones (e.g., mixture rate) are easier to tune. We will include hyperparameter tables for clarity as:
> https://gofile.io/d/8Eq728
>
> - **[Q7] Hyperparameter Tuning and Early Stopping**
>
> Thank you for the question. We set the number of training steps based on when the policies achieve sufficiently high scores under non-attacked conditions. While some environments may require fewer steps, we adopted a consistent training schedule to ensure fair comparisons across methods.
>
> In off-policy settings, longer training can help refresh the replay buffer with newer data, mitigating distribution shift and potentially improving robustness—albeit at the cost of sample efficiency. However, we did not adopt this strategy in the current work and instead consider addressing distribution shift an important direction for future research.
>
> Tuning hyperparameters across multiple attacks is challenging. During training, we apply simple heuristic attacks to identify promising candidates, and later refine them under various attacks.
>
> While trade-offs exist, our experiments indicate that robustness tends to be more influenced by the choice of method than by hyperparameter settings. We view robust hyperparameter tuning under various attacks as an important and open challenge for future work.
>
> ---
> We hope our responses help clarify your concerns, or will be addressed in the revision. If you find the direction and potential contributions meaningful—especially as a baseline for off-policy adversarial RL—we would sincerely appreciate your consideration in revisiting the score.

---

### Official Review · Reviewer_AMCX · 2025-03-10

**Overall Recommendation:** 3

**Summary:**

This paper proposes a method to address the observation robustness, which does not rely on interacting with the environment and making the algorithm off-policy.

**Claims And Evidence:**

**General:**

By looking at the formulation in sec. 3 and related work, it seems this work aims only at state adversarial robustness. Then the scope should be explicitly written early in the title/abstract/introduction.



**Introduction:**

Why is ATLA framework mentioned that frequently? It seems less novel to me. As pointed out by the authors, ATLA framework by Zhang et al. and Sun et al. is done in 2021. But then, the authors claim this framework is adopted by Pinto 2017. It is not sensible to claim a later-proposed framework is adopted by an algorithm proposed 4 years ago. RARL (Pinto et al.) directly use such adversarial training framework (similar to GAN as well) without theory, thus this so-called ATLA framework, is just normal adversarial training protocol. Back to this work, the authors should not introduce such method by citing Zhang et al. but better citing RARL/GAN first and then, say sth like formally concluded by Zhang et al.

"The mutual dependency between the victim agent and the adversary doubles the required sample size for training, leading to inefficiencies and increased computational costs." Not sure about this, isn't every work studying adversarial robust needs such additional cost?

**Essential References Not Discussed:**

The main issue is related work seems only include "Adversarial Attack and Defense on State Observations." There are lots of other type of attack such as action attack:

Pinto, Lerrel, et al. "Robust adversarial reinforcement learning." *International conference on machine learning*. PMLR, 2017.

Tessler, Chen, Yonathan Efroni, and Shie Mannor. "Action robust reinforcement learning and applications in continuous control." *International Conference on Machine Learning*. PMLR, 2019.





The minor one is that authors claim about the novelty of leveraging off-policy notion to address robustness. But off-policy seems often required in offline robustness problems. Would be good to include some of them in the related work such as but not limited to:

Panaganti, Kishan, et al. "Robust reinforcement learning using offline data." *Advances in neural information processing systems* 35 (2022): 32211-32224.

Rigter, Marc, Bruno Lacerda, and Nick Hawes. "Rambo-rl: Robust adversarial model-based offline reinforcement learning." *Advances in neural information processing systems* 35 (2022): 16082-16097.

Tang, Xiaohang, et al. "Adversarially Robust Decision Transformer." *arXiv preprint arXiv:2407.18414* (2024).

**Experimental Designs Or Analyses:**

The experiments are comprehensive, but there are concerns.

The first one is Fig 2, where the caption is not clearly demonstrating the experimental tasks are under attack or not, and it is under which attack. Besides, SAC is significantly better than others, which might also be a concern.

The issue of Table 1 is that it is hard to conclude the proposed method is superior or not. By looking at the average score, the proposed method only works in Halfcheetah, but this way of concluding is definitely biased. Considering the defense effect against all attacks are important. My proposal will be, to add an additional column/plot of worst-case performance under these attacks. I think it's a way better metric to conclude the defense effect than average performance.

**Methods And Evaluation Criteria:**

The graphical demonstration in Fig 1 is clear and nice-looking!

**Other Comments Or Suggestions:**

For now, I will offer a weak reject. But will consider increasing the score if the concerns and questions are addressed, especially the experiments, specifically the results are hard to draw conclusion.

Typos:
Appendix C: basical procedure
Bottom of page 21: pratical procedure
Algorithm 1: minimiging

**Other Strengths And Weaknesses:**

Paper is well-written. However, it is hard to obtain conclusion from the experiments. Need better metric.

**Questions For Authors:**

The paper tries to motivate by making the algorithm for robustness off-policy. Will this be beneficial if you are doing online RL?

**Relation To Broader Scientific Literature:**

The paper is based on a general adversarial training framework for robustness, namely VALT [Zhang et al, 2021]. In this work, the authors remove the interactions with environment by providing analytical solution to the optimal adversary.

**Theoretical Claims:**

NA

---

> ### Author Rebuttal · Authors · 2025-03-31
>
> We sincerely thank the reviewer for the thoughtful and detailed feedback. Below, we respond to each of the concerns and suggestions.
>
> ---
>
> ## Claims and Evidence
>
> ### **Scope Clarification (Title/Abstract/Introduction)**
>
> Thank you for pointing this out. We agree that our work specifically addresses **adversarial robustness in state observations**, and we will explicitly clarify this scope in the **title**, **abstract**, and **introduction** of the revised manuscript.
>
> In particular, we plan to revise the title to:
> **"Off-Policy Actor-Critic for Adversarial Observation Robustness: ..."**
> to more accurately reflect the focus of our study.
>
> ---
>
> ### **ATLA in Introduction**
>
> Thank you for your comment regarding the historical positioning of the ATLA framework.
>
> Our intention in referring to ATLA was to highlight its influence in the **observation robustness** setting, where it has seen wide empirical use in this domain. We did not mean to suggest that ATLA introduced adversarial training prior to earlier works such as **GANs** or **RARL (Pinto et al., 2017)**. We sincerely apologize for any confusion this may have caused.
>
> In the revised manuscript, we will:
> - Properly cite **GANs** and **RARL** as foundational works in adversarial training.
> - Position **ATLA** as a more recent **formalization** of this framework, specifically within observation perturbation settings.
>
> ---
>
> ### **On Sample Inefficiency of Adversarial Training**
>
> > "The mutual dependency between the victim agent and the adversary doubles the required sample size for training..."
>
> We appreciate this concern.
>
> We agree that not all robust RL approaches incur this cost—especially **offline** or **model-based** methods, as noted. Our original statement specifically referred to **model-free online RL**, where the agent and adversary must interact with the environment during training. We will revise the manuscript to clarify this scope and explicitly note the exceptions.
>
> ---
>
> ## Experimental Clarity
>
> ### **Figure 2 Clarification**
>
> Thank you for the helpful comment. Due to space constraints, the caption of Figure 2 lacked sufficient detail. We will revise it to explicitly state that the evaluations are conducted under nominal (non-adversarial) conditions, as also mentioned in the main text. To further avoid confusion, we will update the caption to direct readers to Table 1 for robustness metrics under attack.
>
> ---
>
> ### **Table 1 Interpretation**
>
> Thank you for the valuable suggestion. We agree that average performance under attacks may obscure important aspects of robustness. In the revision, we will use the worst-case scores instead of average scores. Please see the revised table at the following anonymous link:
>
> https://gofile.io/d/G0l2mW
>
> Along with the revised table, we will clarify that VALT-EPS and VALT-SOFT consistently maintain high robustness across all tasks. For complex tasks such as HalfCheetah and Ant, incorporating MDP-level considerations—as in our methods—is essential for high robustness, especially off-policy. SAC-based methods generally outperform PPO-based methods on Hopper. While PPO variants achieve the highest score in Ant, they require over three times more environment interactions, highlighting our methods' potential for robust RL with significantly improved sample efficiency.
>
> Thank you again for your insightful feedback.
>
> ---
>
> ## Essential References Not Discussed
>
> We greatly appreciate the additional references. In the revised manuscript, we will address topics that were not sufficiently covered, such as **attacks on actions and rewards**.
>
> We also recognize that **offline robust RL** has become increasingly important. We plan to incorporate the literature you suggested, along with recent developments in this area.
>
> Due to space constraints, this discussion may be included in **Appendix A**.
>
> ---
>
> ## Other Comments or Suggestions
>
> ### **Typos**
>
> Thank you for pointing these out. We will correct them in the revised manuscript.
>
> ---
>
> ## Questions for Authors
>
> > **Does your off-policy approach provide benefits in online RL?**
>
> Thank you for this insightful question.
>
> Yes, we believe our off-policy approach can offer significant benefits in **online RL settings**. In real-world applications such as **robotic learning**, improved sample efficiency leads to **reduced interaction time with the physical environment**, which lowers the need for human monitoring and enables **faster, safer deployment**.
>
> That said, if the reviewer is referring to **non-stationary or rapidly changing environments**, we agree that **on-policy methods** or **adaptive formulations** may be more appropriate. Exploring hybrid or continual learning strategies is a promising direction for future work.
>
> ---
>
> Thank you again for your constructive and thoughtful feedback.

---

### Official Review · Reviewer_wL61 · 2025-03-13

**Overall Recommendation:** 3

**Summary:**

This paper proposed an off-policy VALT framework for SA-MDP based on Symmetric Property and Soft Optimization. Compared with the existing ATLA framework, this framework improves sample utilization efficiency as it does not require additional training for the Adversary.

**Claims And Evidence:**

+ In Line 50, it is mentioned that "there is currently no off-policy actor-critic method". Then, what are the advantages of adopting the off-policy approach in SA-MDPs? In the more sensitive setting of state adversarial, off-policy seems to cause severe impacts due to distribution shift (the authors seem to have also mentioned this issue in Line 285). Is such a sacrifice worthwhile for the sake of the advantages of off-policy?
+ What does "enhance generalization" mentioned in Line 67 refer to?

**Essential References Not Discussed:**

There are no necessary references known that this paper fails to discuss.

**Experimental Designs Or Analyses:**

+ Why does the phenomenon of a sudden increase in standard deviation occur in the VALT-EPS algorithm in Hopper in Figure 2(b)? Do the authors have any understanding of this?

**Methods And Evaluation Criteria:**

+ How is the update criterion for policy improvement in Proposition 4.11 obtained?
What are the differences between the SAC-PPO method mentioned in the appendix and the series of PPO methods in the baseline? Why is this additional algorithm introduced?

**Other Comments Or Suggestions:**

For the main comments and suggestions, please refer to the previous sections. Additionally, there are no further comments or suggestions.

**Other Strengths And Weaknesses:**

## Strengths
+ The Symmetric Property proposed in this paper effectively reduces the amount of environmental interaction required by the existing SA-MDPs framework.
+ The VALT framework proposed in this paper fills the gap of the missing off-policy algorithms for solving SA-MDPs.
## Weaknesses
+ Due to the lack of derivations or proofs for Proposition 4.4 and Proposition 4.5, I am not entirely certain about the correctness of the Symmetric Property proposed in this paper.
+ I'm not quite sure whether there are SA-MDPs environments in practice that are highly aggressive, making robust methods insufficiently robust.

**Questions For Authors:**

Refer to the previous sections.

**Relation To Broader Scientific Literature:**

+ The authors mention in Line 59 that "the proposed robust methods were not robust enough against stronger attacks". Are there any practical examples of such SA-MDPs with "stronger attacks"? Providing such examples would help readers better understand the motivation of this paper.

**Theoretical Claims:**

+ How does the general $f$-divergence term relax Equation (3) with constraint (1)? What is the gap between the final Nash equilibrium obtained after relaxation and the Nash equilibrium condition (2)?
+ It seems that the derivations or proofs of Proposition 4.4 and Proposition 4.5 are missing. Is the $Q$ here the same as the $Q$ in Section 3.1? How are $\mathcal{H}$ and $D_f$ introduced?

---

> ### Author Rebuttal · Authors · 2025-03-31
>
> We appreciate your review and interest in our work.
> Below, we address your main concerns. We also acknowledge that some notational details may have been omitted due to space limitations.
>
> ---
>
> ## Claims and Evidence
>
> ### **Motivation and Advantage of this work**
>
> We believe investigating off-policy adversarial learning in SA-MDPs is valuable due to its sample efficiency and performance across diverse tasks. This aligns with the broader trend in RL since the 2010s, where off-policy methods like DQN and DDPG are widely adopted despite distribution shift.
>
> While off-policy methods do suffer from distribution mismatch, their efficiency has justified continued use—alongside many efforts to mitigate such issues. We apply the same reasoning in the adversarial setting. VALT inherits distribution shift challenges like any off-policy method, but provides clear theoretical guarantees and achieves strong empirical results—surpassing Robust-SAC in robustness on HalfCheetah and Ant (especially under learned attacks such as SA-RL and PA-AD) while maintaining sample efficiency.
>
> We emphasize that our method grounds robustness within the MDP framework rather than relying on local smoothness. Our formulation supports both theoretical clarity and practical effectiveness, aligning with ICML’s emphasis on principled contributions.
>
>
>
> ### **What does "enhance generalization" refer to**
>
> “enhance generalization” refers to the agent's improved robustness under previously unseen or perturbed observations.
>
> ---
>
> ## Methods and Evaluation Criteria
>
> ### **Update criterion (Proposition 4.11):**
> We consider a fixed adversary and maximize the value function in Eq. (7). Since $D_f$ does not depend on the policy, it is omitted in the optimization:
>
> $\mathbb{E}\_{\nu} [ \mathbb{E}_{\pi}[Q] + \mathcal{H}(\pi) ]$,
>
> whose analytical solution (as in SAC) is $\pi^{\star}\_{\text{old}}$ in Eq. (13).
> To approximate $argmax\_{\pi} V^{\pi}\_{\nu^{\text{soft}}}(s_t)$, we minimize the KL divergence between $\pi$ and $\pi^{\star}_{\text{old}}$, using the joint distribution $\pi \circ \nu$. This serves as a soft policy improvement criterion.
>
> ### **SAC-PPO vs PPO variants:**
> SAC-PPO uses SAC as the agent, while the PPO variants use PPO.
> We introduce SAC-PPO to provide a fair min-max adversarial baseline for SAC agents, as such comparisons have not been explored in prior work.
>
> ---
>
> ## Theoretical Claims
>
> ### **Relaxation via $f$-divergence:**
> The $f$-divergence allows optimization over a broader perturbation set $\mathcal{N}$, while softly constraining the adversary to stay near a prior distribution $p$. Under Assumption 4.2, the choice of $p$ implicitly bounds the effective perturbations.
>
> ### **Gap to Nash Equilibrium (Eq. 2):**
> As $\alpha_{\text{attk}} \to 0$, the relaxed soft max-min game converges to the exact Nash equilibrium. Similar to entropy regularization in PPO, a small positive $\alpha_{\text{attk}}$ often improves learning stability in practice.
>
> ### **Missing proofs (Propositions 4.4 & 4.5):**
> Due to space constraints, we omitted full derivations. Similar to SAC, we define a modified reward:
>
> $\hat{r} = r + \mathbb{E}_{\mathcal{F}}[\mathcal{H} + D_f]$,
>
> which leads to the soft value update rule under an assumed soft-worst adversary at the next state, yielding Eq. (7) and (8).
> Here, the $Q$-function differs from that in Section 3.1 as it incorporates the adversary’s influence.
>
> ---
>
> ## Experimental Analysis
>
> ### **Variance in Hopper (Fig. 2b):**
> This is due to one unstable seed among the eight used. Similar fluctuations are reported in SAC's paper(Fig. 1 in [1]) for Hopper. As noted in line 1316, we apply adjustments, but Hopper's sensitivity makes some variance unavoidable.
>
> [1] Haarnoja et al., "Soft Actor-Critic Algorithms and Applications", arXiv 2018.
>
> ---
>
> ## Supplementary Material
>
> 1. Line 8 updates the critic; Line 12 updates the target network.
> 2. Learning curves show mean scores over 8 seeds; tables report median seed performance. Lower-scoring outliers affect averages, hence the difference.
>
> In the Ant ablation, the lack of regularization in VALT-SOFT likely leads to a weak agent policy, which in turn results in an inconsistent adversary. In such cases, excluding adversary effects from value estimation (w/oPE) may prevent overly pessimistic updates and yield better training performance. However, we observe that under regularized conditions (+reg.), w/oPE fails to provide correct policy evaluation and leads to lower robustness. We plan to include this analysis in the revised manuscript.
>
> 3. PPO’s training times match those in [2]; we will add our timing results in the revised version.
>
> [2] Liang et al., “Efficient Adversarial Training without Attacking”, NeurIPS 2022.
>
> ---
>
> ## Broader Scientific Context and Weaknesses
>
> - Concerns on theoretical gaps are addressed in the Theoretical Claims section.
> - The motivation and theoretical justification for off-policy SA-MDPs are detailed above (Claims and Evidence).

---

### Official Review · Reviewer_weVU · 2025-03-21

**Overall Recommendation:** 3

**Summary:**

The paper "Robust Off-Policy Actor-Critic: Virtual Alternative Training via Symmetric Policy Evaluation" addresses the challenge of training reinforcement learning (RL) agents that are robust to adversarial perturbations in their input observations. Existing methods often rely on alternating training between the agent and an explicitly learned adversary, which can be sample-inefficient and difficult to integrate with off-policy algorithms. This paper proposes a novel off-policy framework called Virtual ALternative Training (VALT). The key idea of VALT is to reformulate adversarial learning as a soft-constrained optimization problem that eliminates the need for additional environmental interactions to train the adversary. Instead, the adversary's policy and value function are implicitly derived by leveraging the agent's value estimation and a symmetric property of policy evaluation between the agent and the adversary.

The paper presents a way to construct an alternative adversarial training framework without explicitly learning an RL policy for the adversary. This is achieved by exploiting the symmetry in policy evaluation. The authors present two concrete algorithms based on the Soft Actor-Critic (SAC) that implement the VALT framework. These algorithms demonstrate both sample efficiency and robustness against various adversarial attacks. VALT-EPS-SAC uses an epsilon-worst case approach to approximate the adversary and VALT-SOFT-SAC employs a parameterized policy network to model the (soft) optimal adversary.

The paper presents experiments on challenging MuJoCo continuous control tasks (HalfCheetah, Hopper, Walker2d, and Ant) show that the proposed VALT-based algorithms achieve significantly better sample efficiency compared to on-policy adversarial training methods (like ATLA-PPO) and demonstrate superior robustness against a wide range of heuristic and learning-based adversaries, often outperforming existing robust off-policy baselines (like Robust-SAC).

**Claims And Evidence:**

The first claim is that it is theoretically possible to construct an adversarial framework without requiring an explicit RL process for the adversary. This is achieved by a theoretical proof showing that the adversary's optimal value function has a simple relationship to the agents optimal value function.

The second claim is that this new framework is viable and robust experimentally. This is supported by empirical results in figure 2 and table 1. Experiments are run on some OpenAi gym control problems.

**Essential References Not Discussed:**

N/A

**Experimental Designs Or Analyses:**

I focused on the main results from sec. 5. The main results seem sound and valid. The authors compare against a wide range of both on-policy and off-policy methods, including state-of-the-art robust RL algorithms, allows for a strong assessment of VALT's relative performance.

They evaluated robustness against various attackers:
* Heuristic Attacks: Random (Uniform), Max-ActionDiff (MAD), PGD (minQ for SAC and minV for PPO), and RobustSarsa (RS). This provides a range of attack strategies, from simple random noise to gradient-based and more sophisticated attacks.
* Learning-Based Adversary Attacks: SA-RL (PPO and SAC) and PA-AD (PPO and SAC). These are more challenging attackers, as they are learned RL agents themselves.

This provides a good picture of the performance of performance.

**Methods And Evaluation Criteria:**

The method is tested on OpenAI gym which has relatively simple state representation. The observations are visually very simple which somewhat limits the potential impact.

Mentioning computational cost is good, but a more thorough analysis of wall-clock time or FLOPs compared to baselines would be valuable, especially considering VALT claims to improve sample efficiency without increased computation.

**Other Comments Or Suggestions:**

n/a

**Other Strengths And Weaknesses:**

* The paper acknowledges that VALT currently lacks support for off-policy algorithms in discrete action environments. The experimental evaluation is therefore limited to continuous action domains. Showing results in discrete action domains, or at least discussing potential adaptations for discrete action spaces more thoroughly, would be beneficial for the broader applicability of VALT.

* Mentioning computational cost is good, but a more thorough analysis of wall-clock time or FLOPs compared to baselines would be valuable, especially considering VALT claims to improve sample efficiency without increased computation.

* In some of the experiments without any perturbation e.g. half cheetah the proposed method gets much worse results than SAC. This is not desirable in general.

**Questions For Authors:**

* Have you tried other more visually challenging environments. do the results still hold ?

* why do you think the results without noise are so much worse on half cheetah for VALT than the SAC baseline.

**Relation To Broader Scientific Literature:**

I think it's novel enough as a contribution to the robust RL literature.

**Theoretical Claims:**

I only had a look at the proof of theorem 4.6 because it is the most important piece. It seems correct.

---

> ### Author Rebuttal · Authors · 2025-03-31
>
> **We appreciate your detailed review and insightful suggestions.**
> We understand that your main concerns center around:
> (1) adaptation to discrete action domains,
> (2) analysis of computational cost, and
> (3) performance degradation of VALT on HalfCheetah in the absence of noise.
> Below, we address each point in detail.
>
> ---
>
> ### (1) Adaptation to Discrete Action Domains and Visually Challenging Environments
>
> > _"The paper acknowledges that VALT currently lacks support for off-policy algorithms in discrete action environments... Have you tried other more visually challenging environments?"_
>
> Thank you for highlighting this important direction.
> We agree that both discrete action settings and visually challenging observations are valuable for broadening the applicability of robust RL methods.
> Although we have not yet explored this direction, we plan to further develop our current approach to pursue significant improvements in this area.
>
> As briefly noted in line 1441, we observe a conceptual connection between VALT-EPS and WocaR. In the case of discrete action spaces, we believe that a discrete variant of VALT-EPS would resemble WocaR-DQN. Specifically, our variant would rely on a **single soft-worst Q-function** (compared to the two used in WocaR-DQN: vanilla and worst), updated using **both uniformly random attacked Q-values and PGD/convex-relaxed worst-case Q-values**.
>
> While this variation offers interesting theoretical properties—such as **contraction** and **policy improvement**—we chose not to further discuss it due to the lack of supporting experiments at this stage. However, if the reviewer finds it appropriate, we would be happy to include this theoretical discussion and point to it.
>
> That said, even without experiments on discrete action tasks with visual inputs, we believe the contributions of this work are significant enough to justify publication:
>
> - A demonstration that adversarial robustness can be achieved without explicitly training an adversary.
> - Theoretical guarantees (e.g., contraction, policy improvement under fixed adversary assumptions).
> - The first extensive benchmark for off-policy robust RL with SAC variants, evaluated under a PPO-compatible framework.
> - We are committed to releasing our code and evaluation environments to support reproducibility and facilitate future research on observation-robust off-policy methods.
>
> ---
>
> ### (2) Computational Cost Analysis
>
> > _"Mentioning computational cost is good, but a more thorough analysis of wall-clock time or FLOPs compared to baselines would be valuable."_
>
> We agree and appreciate this suggestion. While we report wall-clock time in Appendix F, we acknowledge that a more detailed breakdown (e.g., time per processes or FLOPs per step) would be valuable. We plan to include this analysis in a future revision and will provide code and scripts (currently being refactored and documented) that enable monitoring of computation times to support full reproducibility.
>
> ---
>
> ### (3) Performance on HalfCheetah without Noise
>
> > _"Why do you think the results without noise are so much worse on HalfCheetah for VALT than the SAC baseline?"_
>
> This is a great point. As noted in prior robust RL studies (e.g., [1]), there exists a fundamental **trade-off between clean-environment performance and robustness** under adversarial perturbations. This trade-off applies not only to VALT but also to methods like Robust-SAC (see line 1359), where reducing regularization restores SAC-like performance (\~10,000) but sacrifices robustness.
>
> For VALT, similar trade-offs are governed by:
>
> - The adversarial ratio in the behavior policy (e.g., assuming no attack vs. assuming adversarial conditions), and
> - The strength of the soft constraint coefficient (e.g., considering near-random vs. near-worst-case attacks).
>
> The effect of the adversarial ratio in the behavior policy is illustrated in Table 5 and Figure 5.
>
> Although WocaR-PPO [1] also reports such a trade-off, it appears less drastic due to PPO's lower clean performance (\~5,000-6,000), whereas SAC achieves higher scores(\~8,000-10,000), thus making the drop more visually noticeable.
>
> [1] Liang, Y., Sun, Y., Zheng, R., and Huang, F. Efficient adversarial training without attacking: Worst-case-aware robust reinforcement learning. *Advances in Neural Information Processing Systems*, 35:22547–22561, 2022.
>
> ---
>
> Once again, thank you for your constructive feedback. We believe your suggestions will help us further improve the clarity, applicability, and impact of our work.

---

### Decision · Program_Chairs · 2025-05-01

**Decision:**

Accept (poster)

**Comment:**

This paper proposes an interesting off-policy framework for robust reinforcement learning, addressing a key limitation of existing methods that require additional environment interactions to train an adversary. The reviewers generally appreciate the paper’s technical contributions, including the formulation of adversarial learning as a soft-constrained optimization problem supported by a symmetric property of policy evaluation. The proposed algorithms also show strong empirical performance across multiple challenging environments and attack settings, achieving better sample efficiency and robustness compared to prior baselines.

While some concerns were raised about the performance in clean environments, computational cost analysis, and applicability to discrete action spaces, the authors have responded thoroughly and clarified the limitations and trade-offs involved. Overall, the paper provides a valuable contribution to the robust RL literature with sound theoretical foundations, comprehensive experiments, and promising future directions.

Meanwhile, the review team would like to encourage the authors to carefully incorporate the reviewers’ constructive feedback and suggestions to further strengthen the clarity, experimental analysis, and broader applicability of their work. In particular, the authors may provide a more detailed analysis of the wall-clock time or FLOPs compared to baselines to better support their claims on computational efficiency. It would also be valuable to expand the discussion on the potential extension of the proposed framework to discrete action spaces, addressing both theoretical and practical challenges. Additionally, covering more related works suggested by the reviewers, particularly those on offline robust RL and action perturbation attacks, would help position the paper more comprehensively within the literature. Finally, the review team recommends discussing the importance of exploring hybrid or continual learning strategies as a promising direction for enhancing robustness and adaptability in dynamic or non-stationary environments.